# A Distributional View for Visual Mechanistic Interpretability: KL-Minimal Soft-Constraint Principle

Guancheng Zhou [1 2 3]   Yisi Luo [1]   Zhengfu He [2 3 4]   Zhenyu Jin [1]   Xuyang Ge [2 3 4]   Wentao Shu [2 3 4]   Deyu Meng [1 5]
Xipeng Qiu [2 3 4]

## Abstract

Most current paradigms in visual mechanistic interpretability (MI) remain confined to interpreting internal units of the vision model via heuristic methods (e.g., top-$K$ activation retrieval or optimization with regularization). In this work, we establish a theoretical distributional view for visual MI, which models the influence of a feature activation on the natural image distribution, thereby formulating a Kullback-Leibler (KL)-minimal optimization problem to model the MI task. Under this framework, statistical biases are identified within previous MI paradigms, which reveal that they may either be perceptually uninterpretable to humans (i.e., deviate from the natural image distribution), or mechanistically unfaithful to the vision models (i.e., unable to activate model features). To resolve the biases under the distributional view, we propose a model with a KL-minimal soft-constraint principle for visual MI that theoretically balances interpretability and faithfulness. We realize this principle via energy-guided diffusion posterior sampling. Extensive experiments validate the theoretical soundness of the proposed distributional view and demonstrate the practical effectiveness of our paradigm on the DINOv3 vision model. The code is available at https://github.com/SII-ZhouGC/EnergyDPS.

## 1. Introduction

While neural networks have achieved remarkable success in numerous fields (Redmon et al., 2016; Luo et al., 2023),

[1] School of Mathematics and Statistics, Xi'an Jiaotong University [2] Shanghai Innovation Institute [3] OpenMOSS Team [4] Fudan University [5] Ministry of Education Key Lab of Intelligent Networks and Network Security, Xi'an Jiaotong University. Correspondence to: Yisi Luo <yisiluo1221@foxmail.com>, Xipeng Qiu <xpqiu@fudan.edu.cn>.

*Proceedings of the 43rd International Conference on Machine Learning*, Seoul, South Korea. PMLR 306, 2026. Copyright 2026 by the author(s).

their internal mechanisms execute as an elusive "black box", creating hidden dangers in critical fields (Du et al., 2025; Wolf et al., 2024). The study of mechanistic interpretability (MI) aims to reverse-engineer models to explain their behaviors and internal mechanisms. However, when applied to vision models, these methods face a fundamental challenge compared to language models. Unlike language models, vision models process high-dimensional continuous signals (i.e. images), whereas language models process discrete artificial data (i.e. text tokens). The semantics in images are often entangled and lack canonical discrete anchors. As a result, mapping internal units to human-interpretable concepts and validating such mappings can be substantially more challenging in the vision models.

Neurons are the most direct intrinsic units of a model, and abundant research (Fel et al., 2023a; Hesse et al., 2025) has focused on them to reveal the internal mechanisms of vision models. Superposition makes single neurons encode multiple concepts simultaneously (Elhage et al., 2022), which complicates concept-level attribution and interpretation. Sparse autoencoders (SAEs) provide a promising approach to mitigate this issue by learning a monosemantic concept dictionary that can reduce polysemanticity and often yields more interpretable units (Thasarathan et al., 2025; Karvonen et al., 2025; Lim et al., 2025). We refer to these monosemantic concepts as "features".

Although SAEs have been successful in LLMs (Karvonen et al., 2025; He et al., 2025), their application to vision models is limited. Previous works typically rely on heuristic, sample-wise paradigms, such as retrieving high-activation examples from a dataset (Zeiler & Fergus, 2014) or synthesizing inputs via optimization-based feature visualization (Olah et al., 2017). These approaches face a recurring trade-off: retrieval is constrained by finite data and may yield unstable or weakly activating evidence, whereas optimization can produce highly activating but out-of-distribution artifacts. Consequently, the resulting explanations can be either *perceptually uninterpretable* (i.e., deviating from the natural image distribution) or *mechanistically unfaithful* (failing to reflect how the model behaves).

To address these limitations, we propose a fundamental shift

*Table 1.* **Unified Taxonomy of Visual MI Paradigms.** Under the distributional view, we categorize existing methods based on three orthogonal dimensions: the **Object of study** ($\mathcal{O}$), the **Constraint formulation** ($\mathcal{C}$), and the **Sampling methodology** ($\mathcal{S}$). Our analysis reveals that prior biases (highlighted in gray) stem from specific structural designs in these dimensions, while our approach aligns them to achieve the balance between faithfulness and interpretability.

| Paradigm | Representative works | Object of study $\mathcal{O}$ | Constraint formulation $\mathcal{C}$ | Sampling methodology $\mathcal{S}$ | Identified bias |
|---|---|---|---|---|---|
| Activation maximization | Feature vis. (Olah et al., 2017) MACO (Fel et al., 2023a) VITAL (Gorgun et al., 2025) | Neuron w/ task (e.g., classification) | Extreme hard-constraint (point-wise) (e.g., label confidence) | Optimization with regularization | Typical set mismatch (Appendix F.3) Regularization artifact |
| Dataset search | PatchSAE (Lim et al., 2025) USAE (Thasarathan et al., 2025) | SAE feature | Hard-constraint ($s(x) > m$) | Retrieval (e.g., Top-$K$ Sample) | Boundary bias (Lemma D.7) Threshold instability (Lemma D.8) Interpretability gap (Theorem D.14) |
| Proxy-guided generation | DiffExplainer (Pennisi et al., 2025) DEXTER (Carnemolla et al., 2025) | Task (e.g., classification) | KLSC principle (soft) (e.g., label confidence) | Prompt-guided diffusion sampling | Information bottleneck (Theorem F.7) |
| **SAE energy-guided DPS** | EnergyDPS (ours) | SAE feature | KLSC principle (soft) (SAE feature activation) | Energy-guided diffusion posterior sampling | Faithful and interpretable (KL-minimal) |

in perspective via a distributional view. We argue that understanding the semantics of a feature should be treated not only as sample-wise pattern recognition, but also as a Kullback-Leibler (KL)-minimal distributional inference problem. Specifically, a faithful and interpretable explanation of a feature must characterize how this feature reshapes the natural image distribution. Under the distributional view, we introduce a unified taxonomy and decompose the previous visual MI paradigms into three components: *the object of study*, *the constraint formulation*, and *the sampling methodology* (Table 1). This taxonomy facilitates further analysis of the biases inherent in the previous paradigms.

Guided by this distributional view, we propose the KL-minimal soft-constraint (KLSC) principle for the constraint formulation. Unlike previous paradigms (which usually utilize hard-constraints), the KLSC seeks the induced distribution that more closely approximates the natural image distribution under identical faithfulness requirements to balance faithfulness and interpretability. To realize this principle, we introduce energy-guided diffusion posterior sampling (EnergyDPS). By treating the SAE feature activation as an intrinsic energy function, our sampling method samples faithfully from the distribution induced by the KLSC principle. Different from the previous prompt-guided diffusion sampling methods (Pennisi et al., 2025; Carnemolla et al., 2025), EnergyDPS avoids the information bottleneck associated with text prompts. The main contributions are:

- We establish a novel distributional view for visual MI and therefore identify several statistical biases in previous visual MI paradigms.

- We introduce the KLSC principle to theoretically balance faithfulness and interpretability, and we rigorously realize the principle via EnergyDPS.

- Extensive experiments validate the theoretical soundness of our distributional view and demonstrate the practical effectiveness of our paradigm on DINOv3.

**Roadmap.** In Section 2.2.1, we introduce a *distributional view* of visual mechanistic interpretability and a unified taxonomy that decomposes existing paradigms along three orthogonal axes: *the object of study*, *the constraint formulation*, and *the sampling methodology*. Secs. 2.2.2–2.2.4 formalize our framework by deriving the KLSC principle, distinguishing intrinsic scores from task-oriented objectives, and comparing retrieval, optimization-with-regularization, and prompt-guided diffusion as sampling methodologies. Building on these analyses, Section 2.3 realizes KLSC via *EnergyDPS*, yielding a practical sampler that preserves intrinsic semantics while remaining close to the natural image prior. Section 3 provides a practical evaluation on DINOv3 transcoder (an SAE variant) features. Section 4 discusses interpretability vs. realism, and clarifies the resulting limitations.

Complete related work is deferred to Appendix A. Additional preliminaries and SAE details are provided in Appendices B-C. Appendices D-F provide proofs and theoretical refinements for Secs. 2.2.2-2.2.4, and Appendix G proves EnergyDPS achieves the KLSC principle. Experiments implementation details are given in Appendices H-J. Appendix L gives more experiments in DINOv3.

## 2. Distributional View for Visual MI

### 2.1. Preliminaries

We work on a measurable space $(\mathcal{X}, \mathcal{F})$, where $\mathcal{X}$ denotes the image space and $\mathcal{F}$ is a $\sigma$-algebra. Let $\mathcal{P}(\mathcal{X})$ be the set of probability measures on $(\mathcal{X}, \mathcal{F})$. We use $p \in \mathcal{P}(\mathcal{X})$ to denote a reference distribution representing the natural image prior, and $s : \mathcal{X} \to \mathbb{R}$ to denote an intrinsic score (typically differentiable), e.g., a neuron or SAE feature activation. An induced distribution $q \in \mathcal{P}(\mathcal{X})$ is produced by imposing constraints on $s$ while staying close to $p$. We write $q \ll p$ to indicate absolute continuity: for all $A \in \mathcal{F}$, $p(A) = 0$ implies $q(A) = 0$.

Assume $p$ and $q$ admit densities (still denoted by $p(x)$ and

$q(x)$) with respect to a common dominating measure $\mu$. The KL divergence and total variation (TV) distance are

$$\mathrm{KL}(q\|p) := \mathbb{E}_{X \sim q}\left[\log \frac{q(X)}{p(X)}\right] = \int_{\mathcal{X}} q(x) \log \frac{q(x)}{p(x)}\, \mu(\mathrm{d}x), \tag{1}$$

$$\mathrm{TV}(q, p) := \sup_{A \in \mathcal{F}} |q(A) - p(A)| = \frac{1}{2}\int_{\mathcal{X}} |q(x) - p(x)|\, \mu(\mathrm{d}x). \tag{2}$$

By Pinsker's inequality (Lemma B.4), $\mathrm{TV}(q, p) \leq \sqrt{\frac{1}{2}\mathrm{KL}(q\|p)}$; in experiments, we report TV as an interpretable distributional discrepancy. For $x \in \mathcal{X}$, the Dirac measure $\delta_x \in \mathcal{P}(\mathcal{X})$ is $\delta_x(A) = \mathbf{1}_A(x)$ for all $A \in \mathcal{F}$. Detailed preliminaries and SAE formulation are deferred to Appendices B-C; key notation is summarized in Table 5.

## 2.2. Analysis from Distributional View

### 2.2.1. OVERVIEW

To systematically analyze the previous visual MI paradigms, we formalize feature interpretation not as a heuristic visualization task, but as a **KL-minimal optimization problem for distributions**. Understanding the intrinsic score $s$ involves obtaining an *induced distribution $q$* that reflects the influence of $s$ on the natural image distribution $p$. For an ideal induced distribution $q$, we argue that $q$ must balance two competing objectives:

- **Faithfulness:** $q$ should reflect the feature's semantics by concentrating on inputs that strongly activate the score. This can be expressed either as a hard-constraint, $q(\{x : s(x) \geq m\}) = 1$, or as a soft constraint, $\mathbb{E}_{x \sim q}[s(x)] \geq m$.

- **Interpretability:** samples from $q$ should remain human-interpretable, hence $q$ should stay close to $p$, quantified by minimizing the KL divergence $\mathrm{KL}(q\|p)$.

Crucially, these two objectives are often conflicting. Maximizing faithfulness may drive samples towards high-frequency noise (Olah et al., 2017), while maximizing interpretability produces generic images that ignore feature semantics. Addressing this trade-off requires a rigorous distributional view. Here, we define the abstract optimization problem for visual MI as:

$$\text{Interpretation} \sim \mathcal{S}_{\text{sampler}}\left(\mathcal{C}_{\text{constraint}}\left(\underbrace{\mathcal{O}_{\text{object}}(\mathcal{M})}_{\text{Score } s}, p\right)\right), \tag{3}$$

where $\mathcal{O}$ specifies *what* intrinsic score is being interpreted, $\mathcal{C}$ specifies *how* faithfulness-interpretability balance is formulated as a KL-minimal problem, and $\mathcal{S}$ specifies *how* the induced distribution is instantiated (typically via sampling). Previous visual MI paradigms can be systematically reformulated under $(\mathcal{O}, \mathcal{C}, \mathcal{S})$, as summarized in Table 1.

**Related works and their taxonomy.** We list *representative* works for each paradigm below to anchor our taxonomy. A more comprehensive discussion of related visual MI literature is deferred to Appendix A.

- Paradigm I: Activation maximization (Olah et al., 2017; Fel et al., 2023a; Gorgun et al., 2025). Activation maximization optimizes an input (with regularization) to maximize the target activation, yielding a maximum a posteriori (MAP)-like, mode-seeking induced distribution. The induced distribution from regularization can mismatch $p$, leading to regularization artifacts.

- Paradigm II: Dataset search (Thasarathan et al., 2025; Lim et al., 2025). Dataset search retrieves high-activation examples from a finite dataset, preserving perceptual plausibility. Yet, its hard thresholding/top-$K$ instantiation can incur boundary bias and threshold instability (detailed in Section 2.2.2).

- Paradigm III: Proxy paradigms (semantic gap) (Pennisi et al., 2025; Carnemolla et al., 2025). Proxy paradigms steer diffusion via text prompts as external controls to activate features. The cross-modal mapping introduces a semantic gap and an information bottleneck (detailed in Section 2.2.4).

- Paradigm IV: SAE EnergyDPS (ours). We follow the KLSC principle and instantiate it with EnergyDPS, directly sampling a faithful and interpretable induced distribution without proxy semantics (e.g., text prompts).

### 2.2.2. CONSTRAINT FORMULATION: HARD-CONSTRAINTS AND THE KLSC PRINCIPLE

The constraint formulation $\mathcal{C}$ defines the specific formulation of the KL-minimal optimization problem, determining how the intrinsic score $s(x)$ influences the natural image distribution $p$. In this section, we compare the hard-constraints against our proposed KLSC principle.

**Hard- and soft-constraint principle** For a hard-constraint optimization problem, the goal can be formulated by minimizing KL in the hard-constraint set:

**Definition 2.1** (Constraint formulation for hard-constraints). Fix $m \in \mathbb{R}$ and let $\mathbf{E}_m := \{x \in \mathcal{X} : s(x) > m\}$ with $p(\mathbf{E}_m) > 0$. Define the feasible set

$$\mathbf{Q}_m^{\text{hard}} := \{q \in \mathcal{P}(\mathcal{X}) : q \ll p,\ q(\mathbf{E}_m) = 1\}, \tag{4}$$

where $q \ll p$ denotes that $q$ is absolutely continuous with respect to $p$. The formulation of a hard-constraint problem is

$$\underset{q \in \mathbf{Q}_m^{\text{hard}}}{\arg\min}\ \mathrm{KL}(q\|p). \tag{5}$$

The unique solution to Eq. (5) is the truncated distribution (Lemma D.1):

$$q_m^{\text{hard}}(x) = \frac{p(x)\mathbf{1}_{\mathbf{E}_m}(x)}{p(\mathbf{E}_m)}. \tag{6}$$

In effect, hard-constraints impose a sharp cutoff and treat only the surviving high-activation region as valid evidence, which can distort the induced distribution and undermine faithfulness. To mitigate this, we propose the soft-constraint (namely KLSC) principle: instead of enforcing a hard constraint, we require achieving a target level *in expectation*:

**Definition 2.2** (Constraint formulation for KLSC principle). Fix baseline $p$ and intrinsic score $s$. For a target level $m \in \mathbb{R}$, define the feasible set

$$\mathbf{Q}_m := \{q \in \mathcal{P}(\mathcal{X}) : q \ll p, \ \mathbb{E}_q[s(X)] \geq m\}. \tag{7}$$

KLSC Principle defines the induced distribution as the KL-minimal distribution in $\mathbf{Q}_m$:

$$q_\beta^{\text{KLSC}} \in \underset{q \in \mathbf{Q}_m}{\arg\min} \ \text{KL}(q\|p). \tag{8}$$

Lemma D.4 establishes that the unique solution to (8) takes the exponentially tilted form

$$q_\beta^{\text{KLSC}}(x) = \frac{p(x)\exp(\beta s(x))}{Z(\beta)}, \tag{9}$$

where $Z(\beta) = \mathbb{E}_{X\sim p}[\exp(\beta s(X))]$. Here, $\beta \geq 0$ is the Lagrange multiplier corresponding to the constraint $m$, governing the strength of the guidance, and can be determined by $m$ (Remark D.5). In the following analysis, we contrast these two constraint formulations (Definition 2.1 and 2.2) through three aspects: boundary bias, threshold stability, and interpretability.

**Bias I: Boundary bias.** We quantify how probability concentrates near the threshold via the *boundary mass*.

**Definition 2.3** (Boundary mass). Fix a threshold $m$ and a margin $\delta > 0$. Define $\mathbf{E}_m := \{x \in \mathcal{X} : s(x) > m\}$ and the boundary shell $\mathbf{B}_{m,\delta} := \{x \in \mathcal{X} : m < s(x) < m + \delta\}$. For a distribution $q$ with $q(\mathbf{E}_m) > 0$, define

$$\text{BM}_{m,\delta}(q) := \mathbb{P}_{X\sim q}\big[X \in \mathbf{B}_{m,\delta} \mid X \in \mathbf{E}_m\big] = \frac{q(\mathbf{B}_{m,\delta})}{q(\mathbf{E}_m)}. \tag{10}$$

Intuitively, samples from $q_m^{\text{hard}}$ are often treated as representative of a feature's high-activation region. However, for rare events, the boundary mass tends to concentrate in a thin shell near the truncation boundary rather than deep inside the representative samples.

As derived in Lemma D.6, the boundary mass of the hard-constraint distribution $q_m^{\text{hard}}$ is asymptotically governed by the hazard rate of the intrinsic score $s$:

$$\text{BM}_{m,\delta}(q_m^{\text{hard}}) = h_S^{\text{hard}}(m)\delta + o(\delta), \quad \delta \to 0, \tag{11}$$

where $h_S^{\text{hard}}(m) := \frac{f_S(m)}{\int_m^\infty f_S(u)\mathrm{d}u}$ is the *hazard rate*, $S := s(X)$ is a random variable, $f_S$ is the probability density function of $S$.

In contrast, Lemma D.7 yields the effective hazard rate for the tilted distribution $q_\beta^{\text{KLSC}}$:

$$h_{S,\beta}^{\text{KLSC}}(m) := \frac{f_S(m)e^{\beta m}}{\int_m^\infty f_S(u)e^{\beta u}\,\mathrm{d}u}. \tag{12}$$

Since $e^{\beta u}$ is strictly increasing for $\beta > 0$, we have $h_{S,\beta}^{\text{KLSC}}(m) \leq h_S^{\text{hard}}(m)$. Hard-constraint methods tend to pick "just-barely-activated" examples. As a result, the visualization is brittle: changing the threshold slightly can swap out many samples and make the feature appear to mean something different. The KLSC path mitigates this threshold-driven effect in the regimes analyzed below, making interpretations less sensitive to small target-level changes.

**Bias II: Threshold instability.** A desirable constraint formulation should vary smoothly with the target faithfulness level. Hard-constraints induce an abrupt support change by truncating all mass below the threshold, whereas the KLSC principle reweights the entire distribution smoothly. Lemma D.8 shows that this discontinuity can make hard-constraints highly unstable: the KL divergence between two truncated distributions can become $+\infty$ under arbitrarily small threshold perturbations. In contrast, the KLSC induced distribution $q_\beta^{\text{KLSC}}$ forms a smooth exponential family (Lemma D.9); in particular, as $\beta' \to \beta$, $\text{KL}\big(q_\beta^{\text{KLSC}} \parallel q_{\beta'}^{\text{KLSC}}\big) \to 0$. This gives a precise KL-level distinction: hard thresholds are non-regular because their support can jump, whereas KLSC is locally continuous along the exponential-family path.

We further provide a complementary TV-level analysis in Appendix D.4. There, we derive the local TV speeds of hard and KLSC paths when both are indexed by matched faithfulness levels, and prove a strict hard-over-KLSC TV-instability ordering in a canonical Gaussian score model. In the controlled experiments below, we report the finite-step analogue $\text{TV}(q_m, q_{m+\epsilon})$.

**Bias III: Interpretability gap.** We compare interpretability under *matched faithfulness*, measured by the expected activation $\mathbb{E}_q[s(X)]$. Let $q_m^{\text{hard}}(x) = p(x \mid E_m)$ denote the truncated distribution induced by a hard threshold $m$, and choose $\beta$ such that $\mathbb{E}_{q_\beta^{\text{KLSC}}}[s(X)] = \mathbb{E}_{q_m^{\text{hard}}}[s(X)] < \infty$. Under this matching, Theorem D.14 guarantees that the KLSC solution is at least as interpretable as the hard-constraint solution in KL, i.e., $\text{KL}\big(q_\beta^{\text{KLSC}}\|p\big) \leq \text{KL}\big(q_m^{\text{hard}}\|p\big)$, where equality holds if and only if $q_\beta^{\text{KLSC}} = q_m^{\text{hard}}$. This inequality reveals an *interpretability gap*: even

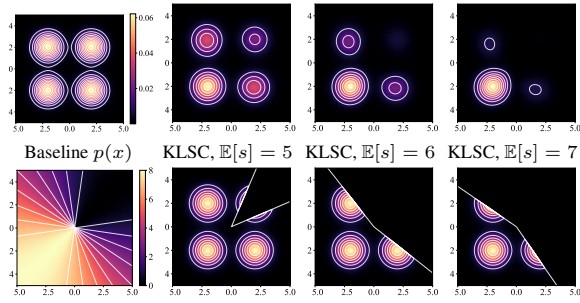

*Figure 1.* **Induced distributions under matched moments.** Top: baseline $p(x)$ and KLSC; bottom: score field $s(x)$ and hard truncation. For $m \in \{5, 6, 7\}$, both are calibrated to satisfy $\mathbb{E}_q[s] = m$; white curves are density contours. See Appendix H.

*Table 2.* **Quantitative diagnosis of constraint behaviors.** We evaluate (1) **Boundary bias** ($\mathrm{BM}_{m', \delta}$ $\downarrow$), (2) **Faithfulness** ($\mathrm{TV}(\cdot, p)$ $\downarrow$), and (3) **Instability** ($\mathrm{TV}(q_m, q_{m+\epsilon})$ $\downarrow$). We use $\delta = 0.05$ and $\epsilon = 0.08$. Details are given in Appendix H.

| $\mathbb{E}[s]$ | Constraints | $\mathrm{BM}_{m', \delta}(\cdot)$ | $\mathrm{TV}(\cdot, p)$ | $\mathrm{TV}(q_m, q_{m+\epsilon})$ |
|---|---|---|---|---|
| 5 | Hard | 0.0103 | 0.2030 | 0.0168 |
|   | KLSC | **0.0101** | **0.1499** | **0.0126** |
| 6 | Hard | 0.0140 | 0.4183 | 0.0319 |
|   | KLSC | **0.0104** | **0.3137** | **0.0147** |
| 7 | Hard | 0.0186 | 0.6198 | 0.0346 |
|   | KLSC | **0.0083** | **0.5095** | **0.0187** |

when both induced distributions achieve the same faithfulness level in expectation, the KLSC principle attains it with a smaller deviation from the natural image distribution $p$.

**Experimental validation.** We validate the predictions in a 2D Gaussian Mixture Model (GMM) toy setting with an angular score $s(x)$. We compare hard truncation $q_{m'}^{\mathrm{hard}}$ and KLSC $q_{\beta}^{\mathrm{KLSC}}$ under matched faithfulness by calibrating $(m', \beta)$ so that $\mathbb{E}_{q_{m'}^{\mathrm{hard}}}[s] = \mathbb{E}_{q_{\beta}^{\mathrm{KLSC}}}[s] = m$ for $m \in \{5, 6, 7\}$. We report boundary mass $\mathrm{BM}_{m', \delta}$, the finite-step TV sensitivity $\mathrm{TV}(q_m, q_{m+\epsilon})$, and prior shift $\mathrm{TV}(q, p)$. Table 2 and Fig. 1 are consistent with the theoretical picture: in this controlled GMM setting, hard truncation concentrates near the threshold and exhibits larger finite-step TV sensitivity, whereas KLSC changes more smoothly and stays closer to $p$ (experimental details in Appendix H). In practice, hard constraints make explanations unstable: tiny changes in $m$ can flip the selected samples and yield qualitatively different visualizations.

### 2.2.3. OBJECT OF STUDY: OBJECT DRIFT FROM TASK INJECTION

Intrinsic scores $s$ are intended to probe a model's *internal* mechanisms. Yet many visual MI pipelines (Pennisi et al., 2025; Carnemolla et al., 2025) inject an external task objective (e.g., classification confidence) as additional guidance. In our taxonomy, this changes the *object of study*: the ef-

fective objective becomes task-entangled, so the resulting explanation may primarily reflect the task rather than the internal mechanisms.

**Task-injected formulation under soft constraints.** Let $c$ denote a task target (e.g., a label or prompt concept) and let $f_c : \mathcal{X} \to \mathbb{R}$ be a measurable task score, e.g., $f_c(x) = \log p_\phi(c \mid x)$. Compared to intrinsic-only KLSC (Definition 2.2), task injection augments the feasible set:

**Definition 2.4** (Task-injected formulation). Fix $m \in \mathbb{R}$ and $r \in \mathbb{R}$. Define

$$q_{m,r,c}^{\mathrm{task}} \in \arg\min_{q \ll p} \mathrm{KL}(q \| p)$$
$$\text{s.t.} \quad \mathbb{E}_q[s(X)] \geq m, \ \mathbb{E}_q[f_c(X)] \geq r. \tag{13}$$

Under standard feasibility and integrability (Lemma E.1), any optimizer takes an exponential-family form

$$q_{m,r,c}^{\mathrm{task}}(x) = \frac{p(x) \exp\big(\beta^\star s(x) + \eta^\star f_c(x)\big)}{Z(\beta^\star, \eta^\star, c)}, \ \exists \beta^\star, \eta^\star \geq 0, \tag{14}$$

with $Z(\beta, \eta, c) = \mathbb{E}_{X \sim p}[\exp(\beta s(X) + \eta f_c(X))]$ and KKT complementary slackness.

**Task-injection distortion from the intrinsic induced distribution.** Let $q_\beta^{\mathrm{KLSC}}$ be the intrinsic-only KLSC solution with $\mathbb{E}_{q_\beta^{\mathrm{KLSC}}}[s] = m$ (when active). We measure the minimal KL deformation required to meet a task demand while preserving the same intrinsic level:

**Definition 2.5** (Task-injection distortion). Assume there exists at least one $q \ll q_\beta^{\mathrm{KLSC}}$ such that $\mathbb{E}_q[s(X)] = m$ and $\mathbb{E}_q[f_c(X)] \geq r$. Define

$$\delta(m, r; c) := \inf_{q \ll q_\beta^{\mathrm{KLSC}}} \mathrm{KL}\big(q \| q_\beta^{\mathrm{KLSC}}\big)$$
$$\text{s.t.} \quad \mathbb{E}_q[s(X)] = m, \ \mathbb{E}_q[f_c(X)] \geq r. \tag{15}$$

From Theorem E.3, $\delta(m, r; c)$ is bounded by

$$\delta(m, r; c) \geq \sup_{\eta \geq 0} \left\{ \eta r - \log \mathbb{E}_{q_\beta^{\mathrm{KLSC}}}\left[ e^{\eta f_c(X)} \right] \right\}, \tag{16}$$

in terms of the log-moment generating function of $f_c$ under $q_\beta^{\mathrm{KLSC}}$. Theorem E.3 shows an intrinsic-task trade-off: once $r$ exceeds the baseline $\mathbb{E}_{q_\beta^{\mathrm{KLSC}}}[f_c(X)]$, achieving the task demand incurs a strictly positive KL cost, i.e., unavoidable task-driven distortion of the intrinsic induced distribution.

**Experimental validation.** Recent pipelines (Rao et al., 2024) discover SAE features but explain them via CLIP prompt matching, introducing a language-alignment interface. This is suitable for task-target concepts, but can bottleneck *intrinsic* MI by restricting explanations to what text

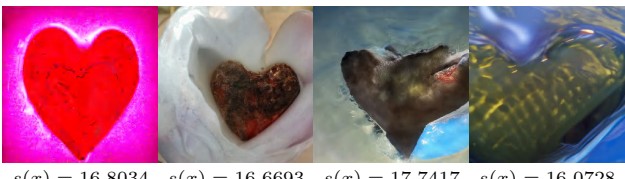

$s(x) = 16.8034 \quad s(x) = 16.6693 \quad s(x) = 17.7417 \quad s(x) = 16.0728$
$f_c(x) = 4.2130 \quad f_c(x) = 10.1400 \quad f_c(x) = 31.3358 \quad f_c(x) = 40.8028$

*Figure 2.* **Object drift under CLIP task injection.** EnergyDPS is guided by an intrinsic SAE score $s(x)$ and a CLIP alignment score $f_c(x)$ (fixed prompt $c$). While $s(x)$ is matched, larger $f_c(x)$ yields qualitatively different samples, showing task dominance over the intrinsic feature. See Appendix L.5.

can express. Consistently, Fig. 2 shows that with intrinsic activation $s(x)$ nearly matched, increasing a CLIP alignment score $f_c(x)$ still yields large semantic shifts, indicating task-dominated object drift. Details are in Appendix L.5.

### 2.2.4. SAMPLING METHODOLOGY

Having discussed the constraint formulation and object of study, the final component is the **sampling methodology**. This operator is responsible for drawing samples from the induced distribution. In this section, we analyze three dominant sampling paradigms, which include **retrieval**, **optimization with regularization**, and **proxy-guided diffusion sampling**.

**Methodology I: Retrieval** The retrieval sampling methodology approximates the induced distribution $q$ via a two-step procedure: (i) first drawing a finite *candidate pool* (i.e., the dataset) $\mathbf{D}_n := \{x_i\}_{i=1}^n$ from the natural image distribution $p$, and (ii) subsequently selecting (via hard-constraints) or reweighting (via soft constraints) these candidates to enforce a target intrinsic score. In the following analysis, we formalize this sampling paradigm and prove that it would introduce a winner-takes-most behavior.

First, we define the empirical distribution by the candidate pool $\hat{p}_n := \frac{1}{n} \sum_{i=1}^n \delta_{x_i}$. For hard-constraints, this sampling methodology suffers from the *boundary bias* (Lemma D.6). For KLSC principle, the sampling distribution is

$$\hat{q}_{n,\beta} := \sum_{i=1}^n \omega_i(\beta) \delta_{x_i}, \ \omega_i(\beta) := \frac{\exp(\beta s(x_i))}{\sum_{j=1}^n \exp(\beta s(x_j))}. \quad (17)$$

Crucially, even without an explicit hard threshold, finite-pool tilting can still degenerate into an extreme selector: as $\beta \to \infty$, the weights concentrate on the maximizers of $s$ within $\mathbf{D}_n$, and $\hat{q}_{n,\beta}$ converges to the uniform distribution over $\arg\max_{x_i \in \mathbf{D}_n} s(x_i)$ (Lemma F.1). This yields a quasi-hard, winner-takes-most behavior on the observed finite set, revealing an intrinsic selection bias induced purely by support restriction. In practice, this often collapses to a few extreme prototypes: retrieval repeatedly returns a small subset of activating patterns, making the final "explanations" unrepresentative of the feature under the natural prior.

**Methodology II: Optimization with regularization.** Traditional feature visualization methods (Olah et al., 2017) construct an explanation by optimizing a single input to increase the intrinsic score activation while penalizing undesirable image statistics (e.g., total variation (Olah et al., 2017)):

$$x^{\text{opt}} = \arg\max_{x \in \mathcal{X}} \big(s(x) - \lambda R(x)\big), \quad (18)$$

where $R$ is the regularization term, and $\lambda$ is the trade-off parameter. From our distributional view, this paradigm can be understood as a **MAP (mode-seeking) sampler**. Indeed, instantiating Lemma F.2 shows that Eq. (18) is exactly the mode of the (unnormalized) Gibbs law

$$\pi(x) \ \propto \ \exp\big(s(x) - \lambda R(x)\big), \quad (19)$$

equivalently interpreting the regularizer as a proxy prior $p_{\text{reg}}(x) \propto \exp(-\lambda R(x))$.

**Extreme hardness and typical-set mismatch.** Although Eq. (18) is continuous in $x$, MAP inference is intrinsically *mode-seeking*. Consider the tempered family $\pi_\beta(x) \propto \exp\big(\beta(s(x) - \lambda R(x))\big)$. As $\beta \to \infty$, $\pi_\beta$ concentrates on the maximizers of $s(x) - \lambda R(x)$; in particular, if the maximizer is unique, then $\pi_\beta \Rightarrow \delta_{x^{\text{opt}}}$. Hence, the resulting induced distribution collapses to a point-wise hard-constraint with vanishing entropy. In high dimensions, such modes can be exponentially unrepresentative of typical samples (Appendix F.3), leading to a *typical-set mismatch*: one may obtain "super-stimuli" with extreme activation that are nevertheless atypical under the natural image distribution, and hence are not semantically representative and perceptually interpretable.

Moreover, since regularization term $R$ defines an implicit prior $p_{\text{reg}}$, the optimized visualization would entangle the intrinsic score with handcrafted inductive bias. As a result, the output reflects a mixture of the feature semantics and the regularization's preferences (e.g., oversmoothing under TV regularization), rather than the intrinsic semantics alone.

**Methodology III: Prompt-guided diffusion sampling.** A fundamental limitation shared by both *retrieval* and *optimization* methodologies is their mismatch with the natural image distribution. In contrast, modern diffusion models (Dhariwal & Nichol, 2021; Song et al., 2021) provide a scalable approximation to this prior; we denote the resulting reference distribution by $p_\theta(x) \approx p(x)$. Recent interpretability visualization methods, such as Diff-Explainer (Pennisi et al., 2025) and DEXTER (Carnemolla et al., 2025), do not directly sample from the desired induced distribution. Instead, they pose interpretation as a **proxy optimization** in the prompt space. Let $p_\theta(x \mid c)$ denote a pre-trained text-conditioned diffusion model (e.g., stable

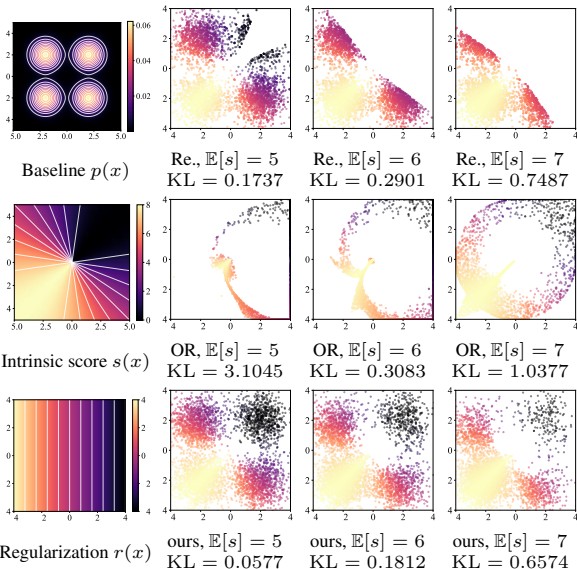

*Figure 3.* **Sampling comparisons under matched faithfulness in a toy GMM.** Left: prior $p(x)$, intrinsic score $s(x)$, and regularization $r(x)$. Columns vary the target $\mathbb{E}[s] \in \{5, 6, 7\}$ (calibrated per method). Rows show retrieval (Re.), optimization w/ regularization (OR), and EnergyDPS (Section 2.3). Numbers report $\mathrm{KL}(q\|p)$ (lower is closer to $p$). Experimental details in Appendix I.

diffusion (Rombach et al., 2022)) with prompt embedding $c \in \mathbf{C}$. These methods seek an embedding $c^\star$ that increases the expected intrinsic score:

$$c^\star \;=\; \arg\max_{c \in \mathbf{C}} \; \mathbb{E}_{x \sim p_\theta(\cdot \mid c)}\big[s(x)\big]. \tag{20}$$

Here, the optimization variable is the embedding $c$, updated via gradient ascent through the diffusion process.

Consequently, prompt-guided sampling does not search over the full probability space $\mathcal{P}(\mathcal{X})$, but is confined to the *prompt-parameterized* family

$$\mathbf{Q}_{\mathrm{prompt}} \;:=\; \big\{\, p_\theta(\cdot \mid c) \,:\, c \in \mathbf{C} \,\big\} \;\subset\; \mathcal{P}(\mathcal{X}). \tag{21}$$

From a distributional view, this can be interpreted as seeking a feasible approximation to the KLSC induced distribution *within* $\mathbf{Q}_{\mathrm{prompt}}$, e.g.,

$$q_m^{\mathrm{proxy}} \in \arg\min_{q \in \mathbf{Q}_{\mathrm{prompt}}} \; \mathrm{KL}(q\|p_\theta) \text{ s.t. } \mathbb{E}_q[s(X)] \geq m. \tag{22}$$

This restriction creates a representational bottleneck: the prompt family may be insufficient to capture fine-grained, polysemantic, or non-verbal feature semantics. We formalize this limitation as a **prompt information bottleneck** (Theorem F.7). In particular, letting $q_\beta^{\mathrm{KLSC}}$ be the (unrestricted) KLSC solution satisfying $\mathbb{E}_{q_\beta^{\mathrm{KLSC}}}[s(X)] = m$, we have

$$\begin{aligned}
&\min_{\substack{q \in \mathbf{Q}_{\mathrm{prompt}} \\ \mathbb{E}_q[s] \geq m}} \; \mathrm{KL}(q\|p_\theta) - \mathrm{KL}\big(q_\beta^{\mathrm{KLSC}}\|p_\theta\big) \\
&\geq \inf_{\substack{q \in \mathbf{Q}_{\mathrm{prompt}} \\ \mathbb{E}_q[s] \geq m}} \; \mathrm{KL}\big(q\|q_\beta^{\mathrm{KLSC}}\big).
\end{aligned} \tag{23}$$

Practically, this helps explain why prompt-guided methods can succeed on output logits yet fail on internal features; we empirically illustrate this behavior in subsequent experiments (Fig. 4). Prompts mainly steer *linguistically describable* concepts, whereas many SAE features are fine-grained, non-verbal, or polysemantic, so the desired $q_\beta^{\mathrm{KLSC}}$ may lie far outside $\mathbf{Q}_{\mathrm{prompt}}$, leading to unfaithful samples.

**Experimental validation.** Fig. 3 validates the predicted sampling pathologies under matched $\mathbb{E}[s]$. Retrieval concentrates near the selection boundary, whereas optimization with regularization departs substantially from $p$ due to prior mismatch. We additionally include EnergyDPS (Section 2.3) as a reference for an ideal sampler. Implementation details are deferred to Appendix I.

## 2.3. The Proposed Paradigm: SAE Energy-Guided Diffusion Posterior Sampling

Our analysis identifies systematic biases in previous visual MI paradigms along three orthogonal axes: *the object of study*, *the constraint formulation*, and *the sampling methodology*. To mitigate these issues, we instantiate the KLSC principle with a practical sampler and propose EnergyDPS. Let $p_\theta$ denote a diffusion prior over images, and let $s : \mathcal{X} \to \mathbb{R}$ be an intrinsic score derived from internal model features (here, SAE feature activations). We target the KLSC-induced tilted distribution

$$q_\beta(x) \;\propto\; p_\theta(x) \exp\big(\beta s(x)\big), \tag{24}$$

where $\beta \geq 0$ controls the strength of intrinsic guidance. Concretely, we leverage diffusion posterior sampling (DPS) and reinterpret the intrinsic score $s(x)$ as an energy over images. We then replace the measurement log-likelihood guidance in DPS (Chung et al., 2023) with energy guidance computed on the predicted clean image, i.e., injecting $\nabla_{x_i} s(\hat{x}_0)$ during the reverse process, yielding a training-free guided sampler (Algorithm 1).

In Appendix G, we prove that EnergyDPS is a faithful sampler that targets Eq. (24) under mild conditions. We show how EnergyDPS resolves the three axes.

- **Object of study.** We adopt SAE features as intrinsic units defining $s(x)$. Compared to single neurons under superposition, SAE features provide sparser and more interpretable units while avoiding task-injected semantics.

- **Constraint formulation.** We employ the KLSC principle, which induces a smooth exponential tilting of $p_\theta$ and avoids the boundary bias and threshold instability inherent to hard truncation.

- **Sampling methodology.** We instantiate the KLSC tilt via DPS-style gradient guidance, injecting $\nabla_{x_i} s(\hat{x}_0)$

**Algorithm 1** Energy-guided DPS (EnergyDPS)

**Require:** Diffusion steps $N$; DDPM noise schedule $\{\beta_i^{\mathrm{ddpm}}\}_{i=1}^N$ (with $\alpha_i = 1 - \beta_i^{\mathrm{ddpm}}$, $\bar{\alpha}_0 = 1$, $\bar{\alpha}_i = \prod_{j=1}^i \alpha_j$); reverse std $\{\tilde{\sigma}_i\}_{i=1}^N$; guidance step sizes $\{\zeta_i\}_{i=1}^N$ and guidance strength $\beta$; diffusion score model $u_\theta(\cdot, i)$; intrinsic score $s$ (also as energy function here).

1: Sample $x_N \sim \mathcal{N}(0, I)$
2: **for** $i = N$ **down to** $1$ **do**
3:   $\hat{u}_i \leftarrow u_\theta(x_i, i)$
4:   $\hat{x}_0 \leftarrow \frac{1}{\sqrt{\bar{\alpha}_i}}\left(x_i + (1 - \bar{\alpha}_i)\hat{u}_i\right)$
5:   Sample $z \sim \mathcal{N}(0, I)$
6:   $x'_{i-1} \leftarrow \frac{\sqrt{\bar{\alpha}_i}(1 - \bar{\alpha}_{i-1})}{1 - \bar{\alpha}_i} x_i + \frac{\sqrt{\bar{\alpha}_{i-1}} \beta_i}{1 - \bar{\alpha}_i} \hat{x}_0 + \tilde{\sigma}_i z$
7:   $x_{i-1} \leftarrow x'_{i-1} + \zeta_i \beta \nabla_{x_i} s(\hat{x}_0)$
8: **end for**
9: **return** $x_0$

at each reverse step. This avoids finite-support bias (retrieval), mode collapse under MAP (optimization), and the prompt-family bottleneck (proxy-guided diffusion).

**Experimental validation.** Returning to Fig. 3, we now instantiate the diffusion-based reference sampler as EnergyDPS and evaluate it under the matched faithfulness level. Fig. 3 shows that EnergyDPS consistently attains the smallest $\mathrm{KL}(q\|p)$ across different target levels, and shows that its samples vary smoothly as $\mathbb{E}[s]$ changes. These observations are empirically consistent with EnergyDPS approximating the KLSC induced distribution under the diffusion prior. Experimental details are deferred to Appendix I.

## 3. Experiments

### 3.1. Experiment Settings

We instantiate KLSC with EnergyDPS on DINOv3 ResNeXt Large (Siméoni et al., 2025). We train a transcoder (an SAE variant; Appendix C) on the stage-2, layer-20 using ImageNet-1K dataset and treat transcoder feature activations as intrinsic scores via a mix strategy. We compare EnergyDPS with: (i) dataset search (top-4 ImageNet-1K samples), (ii) MACO (Fel et al., 2023a) (activation maximization), and (iii) DiffExplainer (Pennisi et al., 2025) (proxy-guided generation paradigm). Details are in Appendix J.

We evaluate the generated visualizations using no-reference proxy metrics (Sec. 3.2) and a blinded interpretability study with human and AI raters (Sec. 3.3). Additional experiments are provided in Appendix L, including neuron-level superposition analysis (Appendix L.1), more ablation experiments on diffusion priors (Appendix L.2.1), semantic shift across activation levels (Appendix L.3), efficiency comparisons (Appendix L.4), object drift under task injection

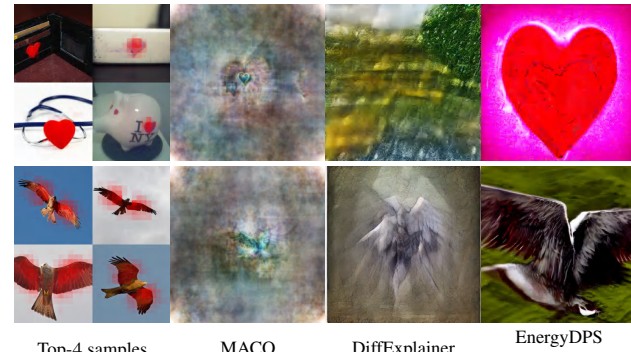

| Top-4 samples | MACO | DiffExplainer | EnergyDPS (ours) |

*Figure 4.* **Qualitative comparison on DINOv3 features.** Columns show Top-4 dataset samples, MACO, DiffExplainer, and EnergyDPS (ours). Rows show feature 81 (top) and feature 9863 (bottom), which visually suggest heart-shaped and wing-like patterns, respectively. These labels are post-hoc summaries for reader guidance, not ground-truth feature labels. Implementation details are provided in Appendix J.

*Table 3.* Quantitative evaluation on DINOv3 (QualiCLIP ↑, NRQM ↑) by MACO, DiffExplainer and EnergyDPS.

| Feature | Methods | $s(x)$ | QualiCLIP | NRQM |
|---|---|---|---|---|
| 9863 | MACO | 14.65 | 0.3846 | 8.1563 |
| | DiffExplainer | 16.53 | 0.3974 | 3.8901 |
| | EnergyDPS | 15.81 | **0.7286** | **8.8636** |
| 4831 | MACO | 16.58 | 0.2946 | 3.7578 |
| | DiffExplainer | 14.66 | 0.3410 | 5.1814 |
| | EnergyDPS | 15.53 | **0.7015** | **8.6307** |
| Random 100 features | MACO | 16.51 | 0.4228 | 4.4869 |
| | DiffExplainer | 15.34 | 0.3259 | 6.7460 |
| | EnergyDPS | 16.20 | **0.5791** | **6.7821** |

(Appendix L.5), and more qualitative comparisons (Appendix L.6).

### 3.2. No-reference Proxies Results

Since SAE features have no ground-truth label, we match *faithfulness* by tuning each method to reach a comparable activation level (aligned to the top-4 retrieval baseline), and evaluate *interpretability* using NRQM (Ma et al., 2017) and QualiCLIP (Agnolucci et al., 2024), which provides no-reference proxies for perceptual interpretability and aesthetic plausibility.

From Fig. 4, EnergyDPS produces human-interpretable samples for two features (81 and 9863), while activation maximization often exhibits high-frequency artifacts and proxy-guided generation is less interpretable. From Table 3, EnergyDPS achieves the best QualiCLIP and NRQM on 100 random additional high-activation features. It consistently improves QualiCLIP, while achieving NRQM comparable to MACO and higher than proxy-guided generation.

*Table 4.* Quantitative evaluation on DINOv3 by the blinded interpretability study (interpretability score ↑, weighted variance ↓). Detailed results are reported in Table 6 in Appendix K.

| Methods | Interpretability score | | | Weighted var. |
|---|---|---|---|---|
| | Human | GPT | Gemini | |
| MACO | 1.92 | 2.09 | 2.57 | 0.2917 |
| DiffExplainer | 1.95 | 2.14 | 1.90 | 0.2643 |
| EnergyDPS | **2.92** | **2.90** | **3.09** | **0.2505** |

### 3.3. Blinded Interpretability Study

We further conduct a blinded pilot study to evaluate whether the generated visualization sets convey clear shared semantics to independent raters. For each of the 100 random SAE features mentioned in Table 3, raters are shown, for each method, a set of four generated images without method labels and additional text labels. They are asked to provide: (i) a short free-form description of the shared semantic meaning, and (ii) an interpretability score on a 1–4 scale, where higher scores indicate clearer shared concepts and more visually informative images.

We collect ratings from six raters, including four human raters and two AI raters, GPT-5.4 and Gemini-3-flash-lite. All raters follow the same evaluation protocol and scoring rubric. To measure the agreement among semantic descriptions, we also compute the weighted variance of textual descriptions in CLIP text-embedding space, using the interpretability score as the weight. Lower variance indicates that raters converge to more similar semantic descriptions. The detailed scoring rubric and the prompts used for AI raters are provided in Appendix K.

From Table 4, EnergyDPS achieves the best blinded interpretability results across all raters. It obtains the highest scores from human, GPT, and Gemini raters, indicating that its generated image sets convey clearer shared semantics than MACO and DiffExplainer. Moreover, EnergyDPS yields the lowest weighted variance of descriptions, suggesting more consistent semantic interpretations across raters.

## 4. Discussions

### 4.1. Interpretability vs. Realism

We emphasize that interpretability is not equivalent to visual realism. Rather, interpretability concerns whether the model's internal behavior is expressed in a form that humans can understand. In our setting, the natural image distribution is a suitable reference space for two reasons. First, natural images are directly perceptible and carry semantic structures that humans can inspect. Second, the model under study is itself a natural-image representation model, so mapping the feature-induced distribution back to the natural image space provides a minimally lossy way to expose its intrinsic visual semantics. In contrast, further projecting the feature into an external prompt or text space may discard visual

information that is not easily verbalized, consistent with the prompt information bottleneck discussed in Sec. 2.2.4.

### 4.2. Reference Space Beyond Natural Images

More generally, KLSC should be understood as a distributional principle relative to a human-interpretable and minimally lossy reference space, rather than as a claim that raw-data realism always implies interpretability. In domains where raw observations are not themselves human-interpretable, the reference prior should instead be defined over an associated representation that is more directly understandable to humans. For example, in full-waveform inversion, one may define the prior over reconstructed velocity models rather than raw waveform measurements.

### 4.3. Limitations

Although the natural image distribution provides a useful and minimally lossy reference space for vision models, there remains a gap between natural images and truly human-understandable concepts. A natural-image prior may still bias explanations toward visually plausible or naturalistic modes, while missing non-natural but model-relevant evidence. Therefore, our current approach should be understood as an approximation to human-understandable interpretation rather than a complete solution to the gap between natural images and human concepts.

## 5. Conclusion

We proposed a *distributional view* of visual MI: interpreting an intrinsic score $s$ amounts to identifying an induced distribution $q$ that balances *faithfulness* to $s$ and *interpretability* via closeness to a natural image distribution $p$. This view yields a unified taxonomy along three axes—*object of study*, *constraint formulation*, and *sampling methodology*—and explains common failure modes of previous paradigms.

To address these issues, we introduced the KLSC principle, which defines interpretation as a KL-minimal optimization problem under a soft constraint, and achieved it with EnergyDPS. Experiments on GMMs and DINOv3 SAE features validate the effectiveness of our proposed view and show that EnergyDPS more closely matches the natural image distribution $p$ under matched faithfulness than previous paradigms. EnergyDPS incurs the cost of diffusion sampling, requiring many reverse steps, making it slower and more memory-intensive than optimization-based baselines. Improving efficiency and scalability (fewer-step samplers or distillation) is left for future work.

## Acknowledgements

This work is supported by Fundamental and Interdisciplinary Disciplines Breakthrough Plan of the Ministry of Education of China (No. JYB2025XDXM101), Tianyuan Fund for Mathematics of the National Natural Science Foundation of China (Grant No. 12426105), the China NSFC projects (Grant No. 62476214 and 124B2029).

## Impact Statement

This paper presents work whose goal is to advance the field of Machine Learning. There are many potential societal consequences of our work, none which we feel must be specifically highlighted here.

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

# A. Related Work

## A.1. Mechanistic Interpretability in Vision Models

To understand the internal mechanisms of the model, we need to study its internal computational units. Neurons are the most direct internal units of a model, and abundant researches have focused on them to reveal the internal mechanisms of models. Attribution-based methods (Muzellec et al., 2024; Novello et al., 2022; Dunefsky et al., 2024) trace the models in under specific stimuli, but they do not reveal "what" internal features are being computed (Fel et al., 2025). To move beyond attribution, concept-based methods (Poeta et al., 2023; Kim et al., 2018) shift the focus to interpreting the latent representations within the model (e.g., neurons). However, this neuron-centric methods faces fundamental theoretical limitations: individual neurons frequently exhibit polysemanticity, lack a privileged basis representation, and implicitly premise that the semantic feature space is dimensionally bounded by the number of units. To address this limitation, the subsequent series of researches (Park et al., 2024; Elhage et al., 2022; Fel et al., 2023b; 2025) crystallized into *Linear Representation Hypothesis* (LRH), which holds that models contain many more features than neurons, arranged as sparse, quasi-orthogonal directions (Fel et al., 2025). Under the LRH, we can learn this polysemantic concepts through the dictionary learning methods. SAE has been proven to be an effective research method. The polysemantic concepts learned by SAE are commonly referred to as features. To understand the meaning of these features, we typically need to employ feature visualization techniques.

## A.2. Feature Visualization in Vision Models

A common strategy to obtain the explanations of models' mechanisms is by delving into internal behavior (Yosinski et al., 2015; Zeiler & Fergus, 2014), and identify their preferred stimuli. Originally, the preferred stimuli is sought from a large dataset (Zeiler & Fergus, 2014), which we called it as **dataset search** paradigm. However, evaluating each neuron or feature across abundant images requires significant computational demands, compounded by the challenge of missing informative images and the non-trivial interpretation of specific visual features within images. **Activation maximization** paradigm automatically generate the preferred stimuli for specific models' internal units (Yosinski et al., 2015; Fel et al., 2023a; Gorgun et al., 2025; Mahendran & Vedaldi, 2016). Compared with dataset search, activation maximization paradigm needs lower computation demands. However, previous studies (Nguyen et al., 2015; Olah et al., 2017) demonstrate that unconstrained optimization may lead to uninterpretable patterns. The vast image distribution may contain "fooling" inputs that excite the internal units without resembling natural images (Pennisi et al., 2025). To address this issue, previous activation maximization paradigms design specific regularization to constraint the image distribution (Simonyan et al., 2014; Fel et al., 2023a; Gorgun et al., 2025; Olah et al., 2017) manually, which may introduce external biases. Recently, DiffExplainer (Pennisi et al., 2025) and DEXTER (Carnemolla et al., 2025) introduce a **diffusion-based proxy paradigms**, which optimize a soft or hard prompt to generate the preferred stimuli for specific internal units. These methods rely on prompt-guided diffusion models to obtain the natural image prior, but this process from image to prompt inevitably results in information loss. This information loss is not severe in visualizations of task-specific outputs, as the classifier's labels can typically be aligned with the prompt. However, when visualizing units within the model, this loss is amplified or even rendered ineffective (Fig. 4). To address this limitation, we theoretically analyze the above paradigms, and we propose EnergyDPS to get lossless visualization of internal units within the vision model.

# B. More Preliminary

Let $(\mathcal{X}, \mathcal{F})$ be a measurable space. We denote by $\mathcal{P}(\mathcal{X})$ the set of all probability measures on $(\mathcal{X}, \mathcal{F})$. To simplify notation, we assume throughout this work that all considered probability measures admit probability density functions (PDFs) with respect to a common $\sigma$-finite reference measure $\mu$ (e.g., the Lebesgue measure on $\mathbb{R}^d$). We use $p(x)$ and $q(x)$ to denote the density functions of distributions $P, Q \in \mathcal{P}(\mathcal{X})$, respectively.

**Definition B.1** (Expectation). For a random variable $X \sim P$ and a measurable function $f : \mathcal{X} \to \mathbb{R}$, the expectation is defined as:

$$\mathbb{E}_{x \sim p}[f(x)] := \int_{\mathcal{X}} f(x) \, p(x) \, \mu(\mathrm{d}x). \tag{25}$$

**Definition B.2** (Kullback–Leibler Divergence). The Kullback–Leibler (KL) divergence from $Q$ to $P$ is defined as the expected log-likelihood ratio:

$$\mathrm{KL}(q\|p) := \mathbb{E}_{x \sim q}\left[\log \frac{q(x)}{p(x)}\right] = \int_{\mathcal{X}} q(x) \log \frac{q(x)}{p(x)} \, \mu(\mathrm{d}x). \tag{26}$$

We adopt the convention that $0 \log 0 = 0$, and $\mathrm{KL}(q\|p) = +\infty$ if there exists a set $A$ with $\mu(A) > 0$ such that $q(x) > 0$ and $p(x) = 0$ for $x \in A$ (absolute continuity condition).

**Definition B.3** (Total Variation Distance). The Total Variation (TV) distance between two distributions $Q$ and $P$ quantifies the $L_1$ distance between their densities:

$$\mathrm{TV}(q, p) := \frac{1}{2} \int_{\mathcal{X}} |q(x) - p(x)| \, \mu(\mathrm{d}x). \tag{27}$$

Intuitively, this represents the largest possible difference in probabilities that the two distributions can assign to the same event.

**Lemma B.4** (Pinsker's Inequality, Theorem 2.33 in Yeung (2008)). *For any two distributions $q, p \in \mathcal{P}(\mathcal{X})$, the total variation distance is upper bounded by the square root of the KL divergence:*

$$\mathrm{TV}(q, p) \leq \sqrt{\frac{1}{2}\mathrm{KL}(q\|p)}. \tag{28}$$

**Definition B.5** (Dirac measure / Dirac delta). Let $(\mathcal{X}, \mathcal{B})$ be a measurable space and let $x \in \mathcal{X}$. The *Dirac measure* at $x$, denoted by $\delta_x$, is the probability measure on $(\mathcal{X}, \mathcal{B})$ defined by

$$\delta_x(A) := \begin{cases} 1, & x \in A, \\ 0, & x \notin A, \end{cases} \quad \forall A \in \mathcal{B}. \tag{29}$$

Equivalently, for any measurable function $\varphi : \mathcal{X} \to \mathbb{R}$ that is integrable with respect to $\delta_x$,

$$\int_{\mathcal{X}} \varphi(u) \, d\delta_x(u) = \varphi(x). \tag{30}$$

When $\mathcal{X} \subseteq \mathbb{R}^d$, one may also view $\delta_x$ as the *Dirac delta distribution* satisfying

$$\int_{\mathbb{R}^d} \varphi(u) \, \delta(u - x) \, du = \varphi(x) \tag{31}$$

for all test functions $\varphi \in C_c^\infty(\mathbb{R}^d)$.

We further provide a key notation table in Table 5.

## C. The Detailed Formulation of SAE

We first restate the sparse autoencoder (SAE) formulation, which serves as the base model throughout this paper. Let $x \in \mathbb{R}^{B \times S \times d}$ denote a batch of intermediate representations (e.g., patch-/token-wise activations), where $B$ is the batch size, $S$ the number of spatial positions/tokens, and $d$ the channel dimension. A standard SAE comprises an encoder, a sparsifying nonlinearity, and a shared dictionary decoder:

$$h = xW_E + \mathbf{1}b_E^\top, \quad W_E \in \mathbb{R}^{d \times d_{\mathrm{sae}}}, \; b_E \in \mathbb{R}^{d_{\mathrm{sae}}}, \tag{32}$$
$$z = \mathrm{Top}\text{-}k(h), \tag{33}$$
$$\hat{x} = zW_D + \mathbf{1}b_D^\top, \quad W_D \in \mathbb{R}^{d_{\mathrm{sae}} \times d}, \; b_D \in \mathbb{R}^d. \tag{34}$$

Here $W_D$ is a learned dictionary shared across positions, and $z \in \mathbb{R}^{B \times S \times d_{\mathrm{sae}}}$ is the sparse code.

**Top-$K$ sparsification.** The operator $\mathrm{Top}\text{-}k(\cdot)$ is applied along the feature dimension for each position: for each $(b, s)$, it retains the $k$ largest components of $h_{b,s,:} \in \mathbb{R}^{d_{\mathrm{sae}}}$ (by value) and sets the remaining entries to zero. Thus, each code vector $z_{b,s,:}$ is exactly $k$-sparse.

| Symbol | Meaning (in this paper) |
|---|---|
| $\mathcal{X}; x \in \mathcal{X}; X$ | Image space; an image; a random image variable. |
| $p \in \mathcal{P}(\mathcal{X})$ | Natural (reference) image distribution. |
| $q \in \mathcal{P}(\mathcal{X})$ | Induced/explanatory distribution (the object we solve for). |
| $s : \mathcal{X} \to \mathbb{R}$ | *Intrinsic score* (mechanism-derived scalar), e.g., neuron/SAE feature activation. |
| $f_c : \mathcal{X} \to \mathbb{R}$ | *Task-target score* associated with an external target $c$ (e.g., label log-likelihood). |
| $m \in \mathbb{R}$ | Faithfulness target level (used as a threshold in hard-constraints or as a moment level in soft constraints). |
| $\mathbf{E}_m := \{x \in \mathcal{X} : s(x) > m\}$ | Hard-constraint event set (activation exceeds threshold). |
| $q_m^{\text{hard}}$ | Hard truncation solution, typically $p(\cdot \mid \mathbf{E}_m)$. |
| $q_\beta^{\text{KLSC}}$ | KL-minimal Soft-constraint (KLSC) solution via exponential tilting: $q_\beta(x) \propto p(x)\exp(\beta s(x))$. |
| $Z(\beta)$ | Partition function $Z(\beta) := \mathbb{E}_p[\exp(\beta s(X))]$ (assumed finite where used). |
| $\beta$ | Lagrange multiplier / inverse temperature controlling the strength of tilting (soft constraint). |
| $\eta$ | Weight for task injection when combining intrinsic and task-target scores, e.g., $\exp(\beta s + \eta f_c)$. |
| $\text{KL}(q\|p)$ | KL divergence used to quantify distributional deviation from $p$ (interpretability/naturalness proxy). |
| $\text{TV}(q, p)$ | Total variation distance; used to quantify distributional gaps and instability. |
| $\mathbf{D}_n = \{x_i\}_{i=1}^n$ | Candidate pool / dataset drawn from $p$ (retrieval-style sampling). |
| $\delta_{x_i}$ | Dirac measure at $x_i$ (used for empirical distributions). |
| $\hat{p}_n := \frac{1}{n}\sum_{i=1}^n \delta_{x_i}$ | Empirical distribution supported on $\mathbf{D}_n$. |
| $\hat{q}_{n,\beta} := \sum_{i=1}^n \omega_i(\beta)\delta_{x_i}$ | Soft reweighting on a finite pool under KLSC-style weights. |
| $\omega_i(\beta) \propto \exp(\beta s(x_i))$ | Normalized exponential weights on the candidate pool. |
| $\text{BM}_{m,\delta}(q)$ | Boundary mass (probability mass within a $\delta$-shell above threshold $m$), used to quantify boundary bias. |
| $p_\theta$ | Generative prior (e.g., diffusion model) approximating the natural distribution $p$. |

*Table 5.* Key notation used in the main text.

**Training objective and the role of $\mathcal{L}_s$.** SAEs are trained to reconstruct a target representation $y$ (e.g., a hooked activation) while promoting sparsity and/or stability of the codes:

$$\min_\theta \ \mathcal{L}_{\text{rec}}(x, \hat{x}) + \lambda_s \mathcal{L}_s(z), \qquad \mathcal{L}_{\text{rec}}(x, \hat{x}) := \frac{1}{B}\sum_{b=1}^B \sum_{s=1}^S \left\| \hat{x}_{b,s,:} - y_{b,s,:} \right\|_2^2, \tag{35}$$

where $\theta$ collects encoder/decoder parameters. The sparsity regularizer $\mathcal{L}_s$ is a *code-level penalty* that biases the learned representation toward sparse and well-conditioned activations. A common and publication-standard choice is an averaged $\ell_1$ penalty,

$$\mathcal{L}_s(z) := \frac{1}{BS}\sum_{b=1}^B \sum_{s=1}^S \left\| z_{b,s,:} \right\|_1, \tag{36}$$

or other monotone sparsity surrogates. Importantly, under Top-$K$ sparsification the *support size* is already fixed (exact $k$-sparsity per position); in this case $\mathcal{L}_s$ primarily controls the *magnitude distribution* of the active coefficients, improving numerical stability and discouraging degenerate scaling behaviors during training.

### C.1. A CNN-Architecture Transcoder as an SAE Variant

We now introduce a CNN-architecture *transcoder* as a structured *variant* of the standard SAE. Conceptually, a transcoder maps dense intermediate representations into sparse feature representations (and back) via reconstruction. Crucially, in our setting the transcoder *remains an SAE*: it preserves (i) a position-shared dictionary decoder and (ii) the same Top-$K$ sparsification mechanism, while replacing the position-wise linear encoder with a convolutional encoder to explicitly model local spatial context.

**Spatial reshaping.** For vision backbones that produce 2D activations, we index the $S$ positions by a spatial grid. Assume $S = H \cdot W$ and reshape

$$X = \text{reshape}(x) \in \mathbb{R}^{B \times d \times H \times W}, \qquad x = \text{flatten}(X) \in \mathbb{R}^{B \times S \times d}. \tag{37}$$

**Convolutional encoding, shared dictionary decoding.** The transcoder uses a 2D convolutional encoder to produce pre-activations over the grid:

$$H = \text{Conv2D}(X; W_{\text{conv}}, b_{\text{conv}}) \in \mathbb{R}^{B \times d_{\text{sae}} \times H \times W}, \qquad W_{\text{conv}} \in \mathbb{R}^{d_{\text{sae}} \times d \times K \times K}. \tag{38}$$

Flattening yields $h \in \mathbb{R}^{B \times S \times d_{\text{sae}}}$ and Top-$K$ sparsification produces the sparse code:

$$h = \text{flatten}(H), \qquad z = \text{Top-}k(h), \qquad z \in \mathbb{R}^{B \times S \times d_{\text{sae}}}. \tag{39}$$

Decoding uses the *same* position-shared linear dictionary as in the standard SAE:

$$\hat{x} = zW_D + \mathbf{1}b_D^\top, \qquad \hat{x} \in \mathbb{R}^{B \times S \times d}. \tag{40}$$

Therefore, the CNN-architecture transcoder implements

$$x \xrightarrow{\text{reshape}} X \xrightarrow{\text{Conv2D}} H \xrightarrow{\text{Top-}k} z \xrightarrow{\text{shared } W_D} \hat{x}, \tag{41}$$

where the encoder is convolutional (capturing local neighborhoods), while the decoder remains a global dictionary shared across spatial positions.

**Relation to standard SAE.** When $K = 1$ (a $1 \times 1$ convolution), the convolutional encoder reduces to a spatially shared linear map applied independently at each position, recovering the standard SAE encoder up to reshape/flatten operations. Hence, the CNN-architecture transcoder strictly generalizes the standard SAE by injecting local spatial context into the encoding stage, without changing the dictionary-decoder interpretation of the learned features.

**Training objective.** The transcoder is trained under the same SAE objective,

$$\min_{\theta} \quad \mathcal{L}_{\text{rec}}(x, \hat{x}) + \lambda_s \mathcal{L}_s(z), \tag{42}$$

with $\theta = \{W_{\text{conv}}, b_{\text{conv}}, W_D, b_D\}$. Thus, the transcoder viewpoint does not introduce a new learning principle; it provides a CNN-structured SAE parameterization aligned with 2D vision activations.

# D. Deferred Proof for Constraint Formulation in Section 2.2.2

### D.1. Proof for The Solution of Hard Constraints and KLSC Principle

**Lemma D.1.** *For the hard-constraint problem, the unique minimizer of Eq. 5 is*

$$q_m^{\text{hard}}(\cdot) = p(\cdot \mid \mathbf{E}_m) = \frac{p(\cdot)\mathbf{1}_{\mathbf{E}_m}}{p(\mathbf{E}_m)}. \tag{43}$$

*Proof of Lemma D.1.* For any $q \in \mathbf{Q}_m^{\text{hard}}$, since $q$ is supported on $\mathbf{E}_m$, we can write

$$\text{KL}(q\|p) = \int_{\mathbf{E}_m} q(x) \log \frac{q(x)}{p(x)} \, dx. \tag{44}$$

Let $p_{\mathbf{E}} := p(\cdot \mid \mathbf{E}_m)$ denote the conditional distribution. On $\mathbf{E}_m$ we have $\frac{p_{\mathbf{E}}}{p} = \frac{1}{p(\mathbf{E}_m)}$ and thus

$$\log \frac{q(x)}{p(x)} = \log \frac{q(x)}{p_{\mathbf{E}}(x)} + \log \frac{p_{\mathbf{E}}(x)}{p(x)} = \log \frac{q(x)}{p_{\mathbf{E}}(x)} - \log p(\mathbf{E}_m), \qquad x \in \mathbf{E}_m. \tag{45}$$

Taking expectation under $q$ yields the decomposition

$$\text{KL}(q\|p) = \text{KL}(q\|p_{\mathbf{E}}) - \log p(\mathbf{E}_m). \tag{46}$$

The second term is constant over $\mathbf{Q}_m^{\text{hard}}$; the first term satisfies $\text{KL}(q\|p_{\mathbf{E}}) \geq 0$ with equality iff $q = p_{\mathbf{E}}$. Hence the unique minimizer is $q_m^{\text{hard}} = p(\cdot \mid \mathbf{E}_m)$. $\square$

**Assumption D.2** (Exponential integrability). There exists $\beta > 0$ such that $\log \mathbb{E}_p[\exp(\beta s(X))] < \infty$.

*Remark* D.3 (Remark for Assumption D.2). This assumption is mild and holds for almost all practical neural network settings. Since natural image data resides on a compact domain (bounded pixel values) and the intrinsic score $s$ is typically continuous (e.g., composed of finite weights and ReLU activations), the random variable $s(X)$ is bounded almost surely, implying that its moment-generating function exists for all $\beta \in \mathbb{R}$.

**Lemma D.4** (KLSC solution). *Assume Assumption D.2 and let $Z(\beta) := \mathbb{E}_p[\exp(\beta s(X))]$ (finite for $\beta$ in a neighborhood of 0). Let $\mu_0 := \mathbb{E}_p[s(X)]$, which is finite under Assumption D.2. Then the optimization problem* (8) *has a unique minimizer* $q_\beta^{\text{KLSC}}$ *characterized as follows:*

- *If $m \leq \mu_0$, the constraint is inactive and $q_\beta^{\text{KLSC}} = p$ (equivalently $\beta = 0$).*

- *If $m > \mu_0$ and there exists $\beta > 0$ such that the* tilted *distribution*

$$q_\beta^{\text{KLSC}}(x) := \frac{p(x) \exp(\beta s(x))}{Z(\beta)} \tag{47}$$

*satisfies $\mathbb{E}_{q_\beta^{\text{KLSC}}}[s(X)] = m$.*

*Proof of Lemma D.4.* We work with densities $r := \frac{q}{p}$, so that $r \geq 0$, $\mathbb{E}_p[r] = 1$ and

$$\text{KL}(q\|p) = \int q(x) \log\left(\frac{q(x)}{p(x)}\right) \, \mathrm{d}x = \mathbb{E}_p[r \log r]. \tag{48}$$

The constraint $\mathbb{E}_q[s(X)] \geq m$ becomes $\mathbb{E}_p[r\, s] \geq m$. Thus (8) is equivalent to the convex program

$$\min_{r \geq 0} \mathbb{E}_p[r \log r] \quad \text{s.t.} \quad \mathbb{E}_p[r] = 1, \ \mathbb{E}_p[r\, s] \geq m. \tag{49}$$

**Case $m \leq \mu_0$ (inactive constraint).** Recall $\mu_0 = \mathbb{E}_p[s(X)]$. If $m \leq \mu_0$, the choice $r \equiv 1$ (equivalently $q = p$) is feasible for (49) since $\mathbb{E}_p[r\, s] = \mathbb{E}_p[s] = \mu_0 \geq m$. Moreover, for any feasible $r$ we have

$$\mathbb{E}_p[r \log r] = \text{KL}(q\|p) \geq 0, \tag{50}$$

with equality iff $q = p$ (i.e., $r \equiv 1$) by Gibbs' inequality (MacKay, 2003),. Hence $q_\beta^{\text{KLSC}} = p$ is the unique minimizer when $m \leq \mu_0$. In the remainder of the proof we assume $m > \mu_0$.

For any $\beta \geq 0$ with $Z(\beta) = \mathbb{E}_p[e^{\beta s}] < \infty$ and any density $r$ with $\mathbb{E}_p[r] = 1$,

$$\mathbb{E}_p[r \log r] \geq \beta \mathbb{E}_p[r\, s] - \log Z(\beta), \tag{51}$$

with equality iff $r(x) = \frac{e^{\beta s(x)}}{Z(\beta)}$ $p$-a.e. To prove (51), define the density ratio

$$r_\beta(x) := \frac{e^{\beta s(x)}}{Z(\beta)} \quad \text{and} \quad q_\beta^{\text{KLSC}}(x) := p(x)\, r_\beta(x). \tag{52}$$

Since $r_\beta \geq 0$ and $\mathbb{E}_p[r_\beta] = 1$, $q_\beta$ is a valid probability density. By *Gibbs' inequality* (i.e., the non-negativity of KL divergence), we have

$$\text{KL}(q\|q_\beta) \geq 0. \tag{53}$$

Using the density-ratio representation,

$$\text{KL}(q\|q_\beta^{\text{KLSC}}) = \int q(x) \log \frac{q(x)}{q_\beta^{\text{KLSC}}(x)} \, \mathrm{d}x = \int p(x)\, r(x) \log \frac{r(x)}{r_\beta(x)} \, \mathrm{d}x = \mathbb{E}_p\left[r \log \frac{r}{r_\beta}\right], \tag{54}$$

where $r = \frac{q}{p}$. Expanding $\log r_\beta(x) = \beta s(x) - \log Z(\beta)$ yields

$$0 \leq \mathbb{E}_p\left[r \log \frac{r}{r_\beta}\right] = \mathbb{E}_p[r \log r] - \beta \mathbb{E}_p[r\, s] + \log Z(\beta), \tag{55}$$

which rearranges to (51). Equality holds iff $\mathrm{KL}(q\|q_\beta^{\mathrm{KLSC}}) = 0$, i.e. $q = q_\beta^{\mathrm{KLSC}}$ (equivalently $r = r_\beta$) $p$-a.e.

If $q$ is feasible for (8), then $\mathbb{E}_p[r\,s] = \mathbb{E}_q[s] \geq m$. Plugging into (51) gives

$$\mathrm{KL}(q\|p) = \mathbb{E}_p[r\log r] \;\geq\; \beta m - \log Z(\beta) \qquad \forall \beta \geq 0 \text{ with } Z(\beta) < \infty. \tag{56}$$

Hence

$$\inf_{q\in\mathbf{Q}_m} \mathrm{KL}(q\|p) \;\geq\; \sup_{\beta\geq 0}\Big\{\beta m - \log Z(\beta)\Big\}. \tag{57}$$

For any $\beta \geq 0$ with $Z(\beta) < \infty$, define $q_\beta$ by (47). Then $\frac{q_\beta}{p} = r_\beta$, and by direct calculation

$$\mathrm{KL}(q_\beta\|p) = \mathbb{E}_{q_\beta}\left[\log\frac{q_\beta}{p}\right] = \mathbb{E}_{q_\beta}[\beta s] - \log Z(\beta) = \beta\,\mathbb{E}_{q_\beta}[s] - \log Z(\beta). \tag{58}$$

Moreover, $m(\beta) := \mathbb{E}_{q_\beta}[s]$ is well-defined whenever $Z(\beta) < \infty$, and

$$m(\beta) = \frac{\mathrm{d}}{\mathrm{d}\beta}\log Z(\beta), \tag{59}$$

where differentiation under the integral is justified by Assumption D.2 (local finiteness of the moment generating function implies finiteness in a neighborhood and permits dominated convergence for the derivative). Therefore, if the constraint is active and $m$ lies in the achievable range of $m(\beta)$, there exists $\beta^\star \geq 0$ such that $m(\beta^\star) = m$. For this $\beta^\star$, $q_{\beta^\star}$ is feasible, and by (58) we have

$$\mathrm{KL}(q_{\beta^\star}\|p) = \beta^\star m - \log Z(\beta^\star). \tag{60}$$

Combining with the lower bound (56) (applied at $\beta = \beta^\star$) shows that $q_{\beta^\star}$ attains the infimum of (8).

The functional $q \mapsto \mathrm{KL}(q\|p)$ is strictly convex on $\{q : q \ll p\}$, and the feasible set $\mathbf{Q}_m$ is convex. Therefore the minimizer is unique, and must coincide with $q_{\beta^\star}$. $\qquad\square$

*Remark* D.5 (Determining $\beta$ from $m$). For $\beta \geq 0$ with $Z(\beta) := \mathbb{E}_p[\exp(\beta s(X))] < \infty$, define the exponential tilt $q_\beta(x) = \frac{p(x)\exp(\beta s(x))}{Z(\beta)}$ and the achieved mean score

$$\mu(\beta) \;:=\; \mathbb{E}_{q_\beta}[s(X)]. \tag{61}$$

Let $\Lambda(\beta) := \log Z(\beta)$. Then $\mu(\beta) = \Lambda'(\beta)$ and

$$\mu'(\beta) \;=\; \Lambda''(\beta) \;=\; \mathrm{Var}_{q_\beta}(s) \;\geq\; 0, \tag{62}$$

hence $\mu$ is non-decreasing in $\beta$ (and strictly increasing whenever $\mathrm{Var}_{q_\beta}(s) > 0$). Consequently, for a target level $m$: (i) if $m \leq \mu(0) = \mathbb{E}_p[s(X)]$, the constraint in (8) is inactive and $\beta^\star = 0$ with $q_m^\star = p$; (ii) if $m > \mathbb{E}_p[s(X)]$ and $m$ lies in the attainable range of $\mu$, then the unique optimizer is $q_m^\star = q_{\beta^\star}$ where $\beta^\star$ is uniquely determined by the complementary-slackness identity $\mu(\beta^\star) = m$ (and equivalently by the dual maximizer $\beta^\star \in \arg\max_{\beta\geq 0}\{\beta m - \log Z(\beta)\}$). In practice, $\beta^\star$ can be found by solving $\mu(\beta) = m$ via one-dimensional root finding (e.g. bisection).

### D.2. Proof for The Boundary Bias

**Lemma D.6** (Boundary Bias). *Let $X \sim p$ and define the scalar random variable $S := s(X)$ with distribution function $F_S$ and survival function $\bar{F}_S(m) := \mathbb{P}(S > m)$. Assume $\bar{F}_S(m) > 0$. For the hard-conditioned explanation $q_m^{\mathrm{hard}} = p(\cdot \mid S > m)$, the boundary mass satisfies*

$$\begin{aligned}\mathrm{BM}_{m,\delta}(q_m^{\mathrm{hard}}) &= \mathbb{P}\big(m < S < m + \delta \mid S > m\big) \\ &= \frac{F_S(m+\delta) - F_S(m)}{\bar{F}_S(m)}.\end{aligned} \tag{63}$$

*If $S$ admits a density $f_S$ that is continuous at $m$, then as $\delta \to 0$,*

$$\mathrm{BM}_{m,\delta}(q_m^{\mathrm{hard}}) = \frac{f_S(m)}{\bar{F}_S(m)}\,\delta + o(\delta), \tag{64}$$

*where $h_S(m) := \frac{f_S(m)}{\bar{F}_S(m)}$ is the hazard rate of $S$.*

*Proof of Lemma D.6.* By definition,

$$\mathrm{BM}_{m,\delta}(q_m^{\mathrm{hard}}) = \mathbb{P}_{X \sim p}\big(m < s(X) < m + \delta \mid s(X) > m\big) = \frac{\mathbb{P}(m < S < m + \delta)}{\mathbb{P}(S > m)}. \tag{65}$$

Since $\mathbb{P}(m < S < m + \delta) = F_S(m + \delta) - F_S(m)$, we obtain (63). If $f_S$ is continuous at $m$, then $F_S(m + \delta) - F_S(m) = f_S(m)\delta + o(\delta)$ as $\delta \to 0$, yielding (64). $\qquad\square$

**Lemma D.7** (Exponential tilting reduces tail hazard at a fixed threshold). *Let $X \sim p$ and define the scalar score random variable $S := s(X)$ with density $f_S$ and survival function $\bar{F}_S(m) := \mathbb{P}(S > m)$. Fix any threshold $m \in \mathbb{R}$ with $\bar{F}_S(m) > 0$ and any $\beta \geq 0$ such that $\mathbb{E}_p[\exp(\beta S)] < \infty$. Let $q_\beta$ denote the exponentially tilted distribution*

$$q_\beta(x) := \frac{p(x)\exp\big(\beta s(x)\big)}{Z(\beta)}, \qquad Z(\beta) := \mathbb{E}_p[\exp(\beta S)], \tag{66}$$

*which is well-defined by the assumed finiteness of $Z(\beta)$. Consider the tail-conditioned distribution $q_{\beta,m}^{\mathrm{KLSC}} := q_\beta(\cdot \mid S > m)$. Then the corresponding hazard rate at the boundary $m$ (for the induced distribution of $S$ under $q_{\beta,m}^{\mathrm{KLSC}}$) is*

$$h_{S,\beta}(m) = \frac{f_S(m)\, e^{\beta m}}{\int_m^\infty f_S(u)\, e^{\beta u}\, \mathrm{d}u}. \tag{67}$$

*Moreover,*

$$h_{S,\beta}(m) \leq h_S(m) := \frac{f_S(m)}{\bar{F}_S(m)}. \tag{68}$$

*If $\beta > 0$ and $\mathbb{P}(S > m) > 0$ with $S$ non-degenerate on $(m, \infty)$, the inequality is strict.*

*Proof of Lemma D.7.* Under $q_\beta$, the induced (unnormalized) density of $S$ is proportional to $f_S(s)e^{\beta s}$. Conditioning further on $S > m$ cancels the global normalizer, yielding the tail-normalization constant $\int_m^\infty f_S(u)e^{\beta u}\, \mathrm{d}u$. Hence (67) follows from the definition of the hazard rate. For (68), note that for all $u \geq m$ we have $e^{\beta u} \geq e^{\beta m}$, and therefore

$$\int_m^\infty f_S(u)e^{\beta u}\, \mathrm{d}u \geq e^{\beta m} \int_m^\infty f_S(u)\, \mathrm{d}u = e^{\beta m}\, \bar{F}_S(m). \tag{69}$$

Substituting into (67) yields $h_{S,\beta}(m) \leq f_S(m)/\bar{F}_S(m) = h_S(m)$. Strictness holds whenever $e^{\beta u} > e^{\beta m}$ on a set of positive $f_S$-measure in $(m, \infty)$. $\qquad\square$

### D.3. Proof for the Threshold Instability

**Lemma D.8** (Hard thresholds are non-regular in KL). *Let $S := s(X)$ for $X \sim p$, and define $q_m^{\mathrm{hard}} := p(\cdot \mid S > m)$. For any $m_2 > m_1$ such that $\mathbb{P}(m_1 < S \leq m_2) > 0$, we have*

$$\mathrm{KL}\big(q_{m_1}^{\mathrm{hard}} \,\big\|\, q_{m_2}^{\mathrm{hard}}\big) = +\infty. \tag{70}$$

*Proof of Lemma D.8.* Let $A := \{x : \; m_1 < s(x) \leq m_2\}$. Under $q_{m_1}^{\mathrm{hard}}$ we have

$$q_{m_1}^{\mathrm{hard}}(A) = \mathbb{P}(m_1 < S \leq m_2 \mid S > m_1) = \frac{\mathbb{P}(m_1 < S \leq m_2)}{\mathbb{P}(S > m_1)} > 0. \tag{71}$$

However, $q_{m_2}^{\mathrm{hard}}(A) = 0$ since $q_{m_2}^{\mathrm{hard}}$ is supported on $\{S > m_2\}$. Therefore, KL divergence is $+\infty$ whenever the first distribution assigns positive mass to a set where the second assigns zero density/mass. Hence (70) holds. $\qquad\square$

**Lemma D.9** (Soft tilts are continuous in KL). *Assume $Z(\beta) := \mathbb{E}_p[\exp(\beta s(X))]$ is finite for all $\beta$ in an open interval $I \subset \mathbb{R}$. For $\beta \in I$, define the tilted distribution $q_\beta(x) := \frac{p(x)\exp(\beta s(x))}{Z(\beta)}$ and the log-partition function $\psi(\beta) := \log Z(\beta)$. Then for any $\beta, \beta' \in I$,*

$$\mathrm{KL}\big(q_\beta \,\big\|\, q_{\beta'}\big) = (\beta - \beta')\,\psi'(\beta) - \big(\psi(\beta) - \psi(\beta')\big). \tag{72}$$

*Moreover, there exists $\tilde{\beta}$ between $\beta$ and $\beta'$ such that*

$$\mathrm{KL}\big(q_\beta \,\|\, q_{\beta'}\big) = \frac{1}{2}\,\psi''(\tilde{\beta})\,(\beta - \beta')^2 = \frac{1}{2}\,\mathrm{Var}_{q_{\tilde{\beta}}}\big(s(X)\big)\,(\beta - \beta')^2. \tag{73}$$

*In particular, $\mathrm{KL}(q_\beta \| q_{\beta'}) \to 0$ as $\beta' \to \beta$ (KL-continuity).*

*Proof of Lemma D.9.* By definition,

$$\log\frac{q_\beta(x)}{q_{\beta'}(x)} = (\beta - \beta')\,s(x) \ - \ \big(\psi(\beta) - \psi(\beta')\big). \tag{74}$$

Taking expectation under $q_\beta$ yields

$$\mathrm{KL}(q_\beta\|q_{\beta'}) = (\beta - \beta')\,\mathbb{E}_{q_\beta}[s(X)] \ - \ \big(\psi(\beta) - \psi(\beta')\big). \tag{75}$$

Since $\psi'(\beta) = \mathbb{E}_{q_\beta}[s(X)]$ on $I$, this gives (72). By Taylor's theorem applied to $\psi$ between $\beta$ and $\beta'$, there exists $\tilde{\beta}$ in between such that

$$\psi(\beta') = \psi(\beta) + \psi'(\beta)(\beta' - \beta) + \tfrac{1}{2}\psi''(\tilde{\beta})(\beta' - \beta)^2. \tag{76}$$

Substituting into (72) yields (73). Finally, finiteness of $Z$ on the open interval $I$ ensures $\psi''(\tilde{\beta}) = \mathrm{Var}_{q_{\tilde{\beta}}}(s(X))$ is finite, implying KL-continuity. $\qquad\square$

## D.4. Average TV Instability Across Faithfulness Levels

This section provides a complementary stability analysis under TV distance. Unlike the KL analysis, which highlights the support discontinuity of hard constraints, here we quantify how fast the induced distribution changes when the faithfulness level varies. We further average this local sensitivity over a range of matched faithfulness levels.

Assume that the score random variable $s(X)$ induced by $X \sim p$ admits a density $f$, and that the survival function $\bar{F}$ defined above satisfies $\bar{F}(m) > 0$ on the considered threshold range.

### D.4.1. HARD CONSTRAINTS

For a threshold $m$, define

$$E_m := \{x : s(x) > m\}, \qquad q_m^{\mathrm{hard}} := p(\cdot \mid E_m). \tag{77}$$

The corresponding faithfulness level is

$$\mu_{\mathrm{hard}}(m) := \mathbb{E}_{q_m^{\mathrm{hard}}}[s(X)] = \mathbb{E}[s(X) \mid s(X) > m]. \tag{78}$$

We also define the hazard rate and the mean residual life:

$$h(m) := \frac{f(m)}{\bar{F}(m)}, \qquad e(m) := \mathbb{E}[s(X) - m \mid s(X) > m] = \mu_{\mathrm{hard}}(m) - m. \tag{79}$$

**Lemma D.10** (Exact TV distance for hard constraints). *For any $m_2 > m_1$ with $\bar{F}(m_2) > 0$,*

$$\mathrm{TV}\big(q_{m_1}^{\mathrm{hard}}, q_{m_2}^{\mathrm{hard}}\big) = 1 - \frac{\bar{F}(m_2)}{\bar{F}(m_1)}. \tag{80}$$

*Consequently, if $f$ is continuous at $m$, then*

$$\mathrm{TV}\big(q_m^{\mathrm{hard}}, q_{m+\delta}^{\mathrm{hard}}\big) = h(m)\delta + o(\delta), \qquad \delta \to 0^+. \tag{81}$$

*Proof.* The density of $q_m^{\mathrm{hard}}$ with respect to $p$ is

$$\frac{dq_m^{\mathrm{hard}}}{dp}(x) = \frac{\mathbf{1}_{E_m}(x)}{\bar{F}(m)}. \tag{82}$$

Since $E_{m_2} \subset E_{m_1}$, we have

$$
\begin{aligned}
&\mathrm{TV}\big(q_{m_1}^{\mathrm{hard}}, q_{m_2}^{\mathrm{hard}}\big) \\
&= \frac{1}{2} \int \left| \frac{\mathbf{1}_{E_{m_1}}}{\bar{F}(m_1)} - \frac{\mathbf{1}_{E_{m_2}}}{\bar{F}(m_2)} \right| dp \\
&= \frac{1}{2} \left[ \frac{\mathbb{P}(E_{m_1} \setminus E_{m_2})}{\bar{F}(m_1)} + \mathbb{P}(E_{m_2}) \left( \frac{1}{\bar{F}(m_2)} - \frac{1}{\bar{F}(m_1)} \right) \right] \\
&= \frac{1}{2} \left[ \frac{\bar{F}(m_1) - \bar{F}(m_2)}{\bar{F}(m_1)} + 1 - \frac{\bar{F}(m_2)}{\bar{F}(m_1)} \right] \\
&= 1 - \frac{\bar{F}(m_2)}{\bar{F}(m_1)}.
\end{aligned}
\tag{83}
$$

Taking $m_2 = m + \delta$ and using

$$
\bar{F}(m + \delta) = \bar{F}(m) - f(m)\delta + o(\delta)
\tag{84}
$$

gives the local expansion. $\qquad \square$

**Lemma D.11** (Hard-constraint TV speed under matched faithfulness). *Assume $\mu_{\mathrm{hard}}(m)$ is differentiable and strictly increasing. Then the local TV speed of the hard-constraint path with respect to the faithfulness level $\mu$ is*

$$
v_{\mathrm{hard}}^{(\mu)}(m) := \lim_{\Delta\mu \to 0+} \frac{\mathrm{TV}\big(q_{\mu}^{\mathrm{hard}}, q_{\mu+\Delta\mu}^{\mathrm{hard}}\big)}{\Delta\mu} = \frac{1}{e(m)},
\tag{85}
$$

*where $q_{\mu}^{\mathrm{hard}}$ denotes $q_m^{\mathrm{hard}}$ with $\mu = \mu_{\mathrm{hard}}(m)$.*

*Proof.* Let

$$
N(m) := \int_m^\infty u f(u)\, du.
\tag{86}
$$

Then

$$
\mu_{\mathrm{hard}}(m) = \frac{N(m)}{\bar{F}(m)}.
\tag{87}
$$

Since $N'(m) = -mf(m)$ and $\bar{F}'(m) = -f(m)$,

$$
\begin{aligned}
\frac{d}{dm}\mu_{\mathrm{hard}}(m) &= \frac{N'(m)\bar{F}(m) - N(m)\bar{F}'(m)}{\bar{F}(m)^2} \\
&= \frac{-mf(m)\bar{F}(m) + N(m)f(m)}{\bar{F}(m)^2} \\
&= h(m)\,(\mu_{\mathrm{hard}}(m) - m) \\
&= h(m)e(m).
\end{aligned}
\tag{88}
$$

By Lemma D.10,

$$
\mathrm{TV}\big(q_m^{\mathrm{hard}}, q_{m+\delta}^{\mathrm{hard}}\big) = h(m)\delta + o(\delta).
\tag{89}
$$

Meanwhile,

$$
\mu_{\mathrm{hard}}(m + \delta) - \mu_{\mathrm{hard}}(m) = h(m)e(m)\delta + o(\delta).
\tag{90}
$$

Dividing the two expansions gives

$$
v_{\mathrm{hard}}^{(\mu)}(m) = \frac{1}{e(m)}.
\tag{91}
$$

$\qquad \square$

### D.4.2. KLSC CONSTRAINTS

The KLSC induced distribution is

$$q_\beta^{\text{KLSC}}(x) = \frac{p(x)\exp(\beta s(x))}{Z(\beta)}, \qquad Z(\beta) := \mathbb{E}_p[\exp(\beta s(X))]. \tag{92}$$

Let

$$\psi(\beta) := \log Z(\beta), \qquad \mu_{\text{KLSC}}(\beta) := \mathbb{E}_{q_\beta^{\text{KLSC}}}[s(X)] = \psi'(\beta). \tag{93}$$

Assume that $Z(\beta)$ is finite in an open interval and that

$$\text{Var}_{q_\beta^{\text{KLSC}}}(s(X)) > 0 \tag{94}$$

on the considered range.

**Lemma D.12** (KLSC TV speed under matched faithfulness)**.** *The local TV speed of the KLSC path with respect to the faithfulness level $\mu$ is*

$$v_{\text{KLSC}}^{(\mu)}(\beta) := \lim_{\Delta\mu\to 0} \frac{\text{TV}\left(q_\mu^{\text{KLSC}}, q_{\mu+\Delta\mu}^{\text{KLSC}}\right)}{|\Delta\mu|} = \frac{\mathbb{E}_{q_\beta^{\text{KLSC}}}\left[|s(X) - \mu_{\text{KLSC}}(\beta)|\right]}{2\text{Var}_{q_\beta^{\text{KLSC}}}(s(X))}. \tag{95}$$

*Proof.* The density ratio between two tilted distributions is

$$\frac{dq_{\beta+\varepsilon}^{\text{KLSC}}}{dq_\beta^{\text{KLSC}}}(x) = \exp\left(\varepsilon s(x) - \psi(\beta+\varepsilon) + \psi(\beta)\right). \tag{96}$$

Using

$$\psi(\beta+\varepsilon) = \psi(\beta) + \varepsilon\psi'(\beta) + o(\varepsilon), \tag{97}$$

we obtain

$$\frac{dq_{\beta+\varepsilon}^{\text{KLSC}}}{dq_\beta^{\text{KLSC}}}(x) = 1 + \varepsilon\left(s(x) - \psi'(\beta)\right) + o(\varepsilon). \tag{98}$$

Therefore,

$$\begin{aligned}
\text{TV}\left(q_\beta^{\text{KLSC}}, q_{\beta+\varepsilon}^{\text{KLSC}}\right) &= \frac{1}{2}\mathbb{E}_{q_\beta^{\text{KLSC}}}\left[\left|\frac{dq_{\beta+\varepsilon}^{\text{KLSC}}}{dq_\beta^{\text{KLSC}}} - 1\right|\right] \\
&= \frac{|\varepsilon|}{2}\mathbb{E}_{q_\beta^{\text{KLSC}}}\left[|s(X) - \psi'(\beta)|\right] + o(|\varepsilon|).
\end{aligned} \tag{99}$$

Moreover,

$$\frac{d}{d\beta}\mu_{\text{KLSC}}(\beta) = \psi''(\beta) = \text{Var}_{q_\beta^{\text{KLSC}}}(s(X)). \tag{100}$$

Thus,

$$|\Delta\mu| = \text{Var}_{q_\beta^{\text{KLSC}}}(s(X))|\varepsilon| + o(|\varepsilon|). \tag{101}$$

Dividing the local TV expansion by $|\Delta\mu|$ yields the claim. $\square$

### D.4.3. AVERAGE TV INSTABILITY OVER FAITHFULNESS LEVELS

For any path $\{q_\mu\}_{\mu\in[\mu_-,\mu_+]}$ indexed by matched faithfulness level $\mu$, define its average local TV instability as

$$\overline{v}_{[\mu_-,\mu_+]} := \frac{1}{\mu_+ - \mu_-}\int_{\mu_-}^{\mu_+} v(\mu)\,d\mu, \tag{102}$$

where

$$v(\mu) := \lim_{\Delta\mu\to 0} \frac{\text{TV}(q_\mu, q_{\mu+\Delta\mu})}{|\Delta\mu|}. \tag{103}$$

For the hard-constraint path, if

$$\mu_- = \mu_{\text{hard}}(m_-), \qquad \mu_+ = \mu_{\text{hard}}(m_+), \tag{104}$$

then

$$
\begin{aligned}
\bar{v}_{\text{hard}} &= \frac{1}{\mu_+ - \mu_-} \int_{\mu_-}^{\mu_+} \frac{1}{e(m(\mu))} \, d\mu \\
&= \frac{1}{\mu_{\text{hard}}(m_+) - \mu_{\text{hard}}(m_-)} \int_{m_-}^{m_+} h(m) \, dm,
\end{aligned}
\tag{105}
$$

where we used $d\mu = h(m)e(m) \, dm$.

For the KLSC path, if

$$\mu_- = \mu_{\text{KLSC}}(\beta_-), \qquad \mu_+ = \mu_{\text{KLSC}}(\beta_+), \tag{106}$$

then

$$
\begin{aligned}
\bar{v}_{\text{KLSC}} &= \frac{1}{\mu_+ - \mu_-} \int_{\mu_-}^{\mu_+} \frac{\mathbb{E}_{q_\beta^{\text{KLSC}}}[|s(X) - \mu_{\text{KLSC}}(\beta)|]}{2\text{Var}_{q_\beta^{\text{KLSC}}}(s(X))} \, d\mu \\
&= \frac{1}{\mu_{\text{KLSC}}(\beta_+) - \mu_{\text{KLSC}}(\beta_-)} \int_{\beta_-}^{\beta_+} \frac{1}{2} \mathbb{E}_{q_\beta^{\text{KLSC}}}[|s(X) - \mu_{\text{KLSC}}(\beta)|] \, d\beta.
\end{aligned}
\tag{107}
$$

Therefore, hard-threshold instability is governed by the accumulated tail hazard rate, whereas KLSC instability is governed by the internal score fluctuation along a smooth exponential-family path.

In experiments, for a finite set of faithfulness levels $\mu_1, \ldots, \mu_K$, we report the discrete average local TV instability

$$\text{MeanTV}_\epsilon := \frac{1}{K} \sum_{k=1}^{K} \text{TV}(q_{\mu_k}, q_{\mu_k + \epsilon}), \tag{108}$$

where each $q_{\mu_k}$ is calibrated to the same expected activation level $\mu_k$. This metric measures the average distributional change caused by a small increase in faithfulness.

### D.4.4. GAUSSIAN EXAMPLE

We now instantiate the above quantities in a canonical one-dimensional Gaussian setting. Assume

$$s(X) \sim \mathcal{N}(0, 1). \tag{109}$$

Denote the standard normal density and survival function by

$$\phi(m) := \frac{1}{\sqrt{2\pi}} e^{-m^2/2}, \qquad \bar{\Phi}(m) := \int_m^\infty \phi(t) \, dt, \tag{110}$$

and define the inverse Mills ratio

$$\lambda(m) := \frac{\phi(m)}{\bar{\Phi}(m)}. \tag{111}$$

For the hard-threshold distribution

$$q_m^{\text{hard}} = \mathcal{L}(X \mid s(X) > m), \tag{112}$$

the faithfulness level is

$$\mu_{\text{hard}}(m) = \mathbb{E}[s(X) \mid s(X) > m] = \lambda(m), \tag{113}$$

and the mean residual life is

$$e(m) = \mu_{\text{hard}}(m) - m = \lambda(m) - m. \tag{114}$$

Thus,

$$v_{\text{hard}}^{(\mu)}(m) = \frac{1}{\lambda(m) - m}. \tag{115}$$

The average TV instability over a threshold range $[m_-, m_+]$ is

$$\bar{v}_{\text{hard}} = \frac{\int_{m_-}^{m_+} \lambda(m)\, dm}{\lambda(m_+) - \lambda(m_-)}. \tag{116}$$

Since

$$\frac{d}{dm} \log \bar{\Phi}(m) = -\lambda(m), \tag{117}$$

this can also be written as

$$\bar{v}_{\text{hard}} = \frac{\log \bar{\Phi}(m_-) - \log \bar{\Phi}(m_+)}{\lambda(m_+) - \lambda(m_-)}. \tag{118}$$

For KLSC, we have

$$q_\beta(x) \propto p(x) e^{\beta s(x)}. \tag{119}$$

At the score-distribution level, this induces

$$s(X) \sim \mathcal{N}(\beta, 1) \quad \text{under } q_\beta^{\text{KLSC}}. \tag{120}$$

Hence

$$\mu_{\text{KLSC}}(\beta) = \beta, \qquad \text{Var}_{q_\beta^{\text{KLSC}}}(s(X)) = 1. \tag{121}$$

Moreover,

$$\mathbb{E}_{q_\beta^{\text{KLSC}}}\left[|s(X) - \mu_{\text{KLSC}}(\beta)|\right] = \mathbb{E}_{Z \sim \mathcal{N}(0,1)}[|Z|] = \sqrt{\frac{2}{\pi}}. \tag{122}$$

Therefore,

$$v_{\text{KLSC}}^{(\mu)} = \frac{\sqrt{2/\pi}}{2} = \frac{1}{\sqrt{2\pi}}, \tag{123}$$

which is constant over all faithfulness levels. Consequently, for any interval $[\mu_-, \mu_+]$,

$$\bar{v}_{\text{KLSC}} = \frac{1}{\sqrt{2\pi}}. \tag{124}$$

**Proposition D.13** (Gaussian hard thresholds are more TV-unstable). *For $s(X) \sim \mathcal{N}(0,1)$ and any threshold $m \geq 0$,*

$$v_{\text{hard}}^{(\mu)}(m) > v_{\text{KLSC}}^{(\mu)} = \frac{1}{\sqrt{2\pi}}. \tag{125}$$

*Consequently, over any interval $[m_-, m_+]$ with $m_- \geq 0$,*

$$\bar{v}_{\text{hard}} > \bar{v}_{\text{KLSC}}. \tag{126}$$

*Proof.* Recall that

$$v_{\text{hard}}^{(\mu)}(m) = \frac{1}{\lambda(m) - m}. \tag{127}$$

Let

$$e(m) := \lambda(m) - m. \tag{128}$$

For the standard normal score distribution,

$$\mathbb{E}[s(X) \mid s(X) > m] = \lambda(m), \tag{129}$$

and

$$\mathbb{E}[s(X)^2 \mid s(X) > m] = 1 + m\lambda(m). \tag{130}$$

Thus,

$$\begin{aligned}
\text{Var}(s(X) \mid s(X) > m) &= 1 + m\lambda(m) - \lambda(m)^2 \\
&= 1 - \lambda(m)(\lambda(m) - m) \\
&= 1 - \lambda(m)e(m).
\end{aligned} \tag{131}$$

Since the conditional distribution is non-degenerate, this variance is strictly positive. Hence

$$\lambda(m)e(m) < 1. \tag{132}$$

Moreover,

$$\lambda'(m) = \lambda(m)(\lambda(m) - m) = \lambda(m)e(m). \tag{133}$$

Therefore,

$$e'(m) = \lambda'(m) - 1 = -\mathrm{Var}(s(X) \mid s(X) > m) < 0. \tag{134}$$

Thus $e(m)$ is strictly decreasing. For $m \geq 0$,

$$e(m) \leq e(0) = \lambda(0) = \sqrt{\frac{2}{\pi}}. \tag{135}$$

Therefore,

$$v_{\mathrm{hard}}^{(\mu)}(m) = \frac{1}{e(m)} \geq \sqrt{\frac{\pi}{2}} > \frac{1}{\sqrt{2\pi}} = v_{\mathrm{KLSC}}^{(\mu)}. \tag{136}$$

The average inequality follows by integrating the pointwise inequality over the corresponding faithfulness interval. $\square$

Finally, the inverse Mills ratio satisfies

$$\lambda(m) = m + \frac{1}{m} + O(m^{-3}), \qquad m \to \infty. \tag{137}$$

Thus,

$$e(m) = \lambda(m) - m = \frac{1}{m} + O(m^{-3}), \tag{138}$$

and therefore

$$v_{\mathrm{hard}}^{(\mu)}(m) = \frac{1}{e(m)} = m + O(m^{-1}) \to \infty. \tag{139}$$

By contrast,

$$v_{\mathrm{KLSC}}^{(\mu)} = \frac{1}{\sqrt{2\pi}} \tag{140}$$

is constant. Hence, in the high-faithfulness regime, hard thresholds become increasingly sensitive in TV distance, whereas the KLSC path remains uniformly stable.

## D.5. Proof for The Interpretability Gap

**Theorem D.14** (Interpretability gap: soft dominates hard). *Fix any threshold $m \in \mathbb{R}$ with $p(E_m) > 0$, where $E_m := \{x \in \mathcal{X} :\ s(x) > m\}$, and let*

$$q_m^{\mathrm{hard}}(x) := p(x \mid E_m) = \frac{p(x)\mathbf{1}_{E_m}(x)}{p(E_m)}. \tag{141}$$

*Assume $\mathbb{E}_{q_m^{\mathrm{hard}}}[|s(X)|] < \infty$ and let $\mu := \mathbb{E}_{q_m^{\mathrm{hard}}}[s(X)]$. Let $q_\beta^{\mathrm{KLSC}}$ be a KLSC solution satisfying $\mathbb{E}_{q_\beta^{\mathrm{KLSC}}}[s(X)] = \mu$. Then*

$$\mathrm{KL}\big(q_\beta^{\mathrm{KLSC}}\|p\big) \leq \mathrm{KL}\big(q_m^{\mathrm{hard}}\|p\big). \tag{142}$$

*Moreover, equality holds in* (142) *if and only if $q_\beta^{\mathrm{KLSC}} = q_m^{\mathrm{hard}}$ $p$-a.e.*

*Proof of Lemma D.14.* Let $\mu := \mathbb{E}_{q_m^{\mathrm{hard}}}[s(X)] < \infty$. Then $q_m^{\mathrm{hard}} \ll p$ and $\mathbb{E}_{q_m^{\mathrm{hard}}}[s(X)] = \mu$, hence $q_m^{\mathrm{hard}}$ is feasible for

$$\mathbf{Q}_\mu := \{q \in \mathcal{P}(\mathcal{X}) :\ q \ll p,\ \mathbb{E}_q[s(X)] \geq \mu\}. \tag{143}$$

By definition, any KLSC solution $q_\beta^{\mathrm{KLSC}} \in \arg\min_{q \in \mathbf{Q}_\mu} \mathrm{KL}(q\|p)$ satisfies

$$\mathrm{KL}\big(q_\beta^{\mathrm{KLSC}}\|p\big) \leq \mathrm{KL}\big(q_m^{\mathrm{hard}}\|p\big), \tag{144}$$

which proves (142).

For the equality condition, note that equality in (142) holds if and only if $q_m^{\mathrm{hard}}$ also attains the minimum of $\mathrm{KL}(\cdot\|p)$ over $\mathbf{Q}_\mu$, i.e., iff $q_m^{\mathrm{hard}}$ is itself a KLSC minimizer at level $\mu$. Under the standing uniqueness condition for the KLSC optimizer (Lemma D.4 and Remark D.5), this is equivalent to $q_m^{\mathrm{hard}} = q_\beta^{\mathrm{KLSC}}$ $p$-a.e. Conversely, if $q_m^{\mathrm{hard}} = q_\beta^{\mathrm{KLSC}}$ $p$-a.e., then equality is immediate. $\square$

# E. Deferred Proof for Object of Study in Section 2.2.3

**Lemma E.1** (Task-injection problem solution). *Assume that the feasible set of* (13) *is nonempty, an optimal solution exists, and there exist* $\beta, \eta \geq 0$ *such that*

$$Z(\beta, \eta, c) := \mathbb{E}_{X \sim p}\big[\exp\big(\beta s(X) + \eta f_c(X)\big)\big] < \infty. \tag{145}$$

*Then any optimal solution* $q_{m,r,c}^{\text{task}}$ *of* (13) *is of the form*

$$q_{m,r,c}^{\text{task}}(x) = \frac{p(x)\exp\big(\beta^\star s(x) + \eta^\star f_c(x)\big)}{Z(\beta^\star, \eta^\star, c)}, \qquad \exists \beta^\star, \eta^\star \geq 0, \tag{146}$$

*where* $(\beta^\star, \eta^\star)$ *satisfy the KKT complementary-slackness conditions associated with the constraints in* (13).

*Proof of Lemma E.1.* Let $q$ denote a candidate probability density (or Radon–Nikodym derivative) with $q \ll p$. Consider the convex optimization problem (13) over the convex set $\{q \ll p\}$. Introduce Lagrange multipliers $\beta, \eta \geq 0$ for the two inequality constraints, and let $\lambda \in \mathbb{R}$ denote the multiplier for the normalization constraint $\int q(x)\,dx = 1$. The Lagrangian is

$$\mathcal{L}(q; \beta, \eta, \lambda) = \mathrm{KL}(q\|p) - \beta(\mathbb{E}_q[s] - m) - \eta(\mathbb{E}_q[f_c] - r) + \lambda\Big(\int q(x)\,dx - 1\Big). \tag{147}$$

Writing $\mathrm{KL}(q\|p) = \int q(x)\log\frac{q(x)}{p(x)}\,dx$, a first-order optimality condition in the direction of any perturbation $\delta q$ with $\int \delta q = 0$ yields

$$\int \delta q(x)\Big(\log\frac{q(x)}{p(x)} + 1 - \beta s(x) - \eta f_c(x) + \lambda\Big)\,dx = 0. \tag{148}$$

Since this holds for all such $\delta q$, we obtain (almost everywhere under $p$)

$$\log\frac{q(x)}{p(x)} + 1 - \beta s(x) - \eta f_c(x) + \lambda = 0, \tag{149}$$

equivalently

$$q(x) = p(x)\exp\big(\beta s(x) + \eta f_c(x) - 1 - \lambda\big). \tag{150}$$

Letting $Z(\beta, \eta, c) = \mathbb{E}_{X \sim p}[\exp(\beta s(X) + \eta f_c(X))]$ and choosing $\lambda$ to enforce normalization gives

$$q(x) = \frac{p(x)\exp\big(\beta s(x) + \eta f_c(x)\big)}{Z(\beta, \eta, c)}. \tag{151}$$

Finally, by KKT conditions for inequality constraints, at an optimum there exist $\beta^\star, \eta^\star \geq 0$ such that

$$\beta^\star(\mathbb{E}_q[s] - m) = 0, \qquad \eta^\star(\mathbb{E}_q[f_c] - r) = 0, \tag{152}$$

together with feasibility $\mathbb{E}_q[s] \geq m$ and $\mathbb{E}_q[f_c] \geq r$. Substituting these multipliers yields (146). $\square$

**Lemma E.2** (Gibbs variational principle). *Let* $\nu$ *be a probability measure on* $\mathcal{X}$ *and let* $g : \mathcal{X} \to \mathbb{R}$ *be measurable with* $\mathbb{E}_\nu[e^g] < \infty$. *Then*

$$\log\mathbb{E}_\nu[e^g] = \sup_{q \ll \nu}\Big\{\mathbb{E}_q[g] - \mathrm{KL}(q\|\nu)\Big\}, \tag{153}$$

*and the supremum is achieved uniquely by* $q^\star(dx) = \frac{e^{g(x)}}{\mathbb{E}_\nu[e^g]}\,\nu(dx)$.

*Proof.* Define $q^\star(dx) = \frac{e^{g(x)}}{\mathbb{E}_\nu[e^g]}\nu(dx)$, which is well-defined by the integrability assumption. For any $q \ll \nu$,

$$\mathrm{KL}(q\|q^\star) = \mathbb{E}_q\Big[\log\frac{dq}{dq^\star}\Big] = \mathbb{E}_q\Big[\log\frac{dq}{d\nu} - g + \log\mathbb{E}_\nu[e^g]\Big] = \mathrm{KL}(q\|\nu) - \mathbb{E}_q[g] + \log\mathbb{E}_\nu[e^g]. \tag{154}$$

Rearranging gives

$$\mathbb{E}_q[g] - \mathrm{KL}(q\|\nu) = \log\mathbb{E}_\nu[e^g] - \mathrm{KL}(q\|q^\star) \leq \log\mathbb{E}_\nu[e^g], \tag{155}$$

with equality iff $q = q^\star$. This proves the variational formula and uniqueness. $\square$

**Theorem E.3** (lower bound for $\delta(m, r; c)$). *Assume the intrinsic-only baseline $q_\beta^{\mathrm{KLSC}}$ exists (with $\mathbb{E}_{q_\beta^{\mathrm{KLSC}}}[s(X)] = m$) and $\mathbb{E}_{q_\beta^{\mathrm{KLSC}}}[e^{\eta f_c(X)}] < \infty$ for all $\eta$ in a neighborhood of $0$. Then*

$$\delta(m, r; c) \geq \sup_{\eta \geq 0} \left\{ \eta r - \log \mathbb{E}_{q_\beta^{\mathrm{KLSC}}}[e^{\eta f_c(X)}] \right\}. \tag{156}$$

*Proof.* Fix any feasible $q \ll q_\beta^{\mathrm{KLSC}}$ such that $\mathbb{E}_q[s(X)] = m$ and $\mathbb{E}_q[f_c(X)] \geq r$. For any $\eta \geq 0$, apply Lemma E.2 with $\nu = q_\beta^{\mathrm{KLSC}}$ and $g = \eta f_c$. Then

$$\mathrm{KL}(q \| q_\beta^{\mathrm{KLSC}}) \geq \eta \, \mathbb{E}_q[f_c(X)] - \log \mathbb{E}_{q_\beta^{\mathrm{KLSC}}}[e^{\eta f_c(X)}] \geq \eta r - \log \mathbb{E}_{q_\beta^{\mathrm{KLSC}}}[e^{\eta f_c(X)}]. \tag{157}$$

Taking the infimum over all feasible $q$ yields

$$\delta(m, r; c) \geq \eta r - \log \mathbb{E}_{q_\beta^{\mathrm{KLSC}}}[e^{\eta f_c(X)}]. \tag{158}$$

Finally, taking the supremum over $\eta \geq 0$ completes the proof. $\square$

# F. Deferred Proof for Sampling Methodology in Section 2.2.4

## F.1. Proof for The Retrieval Methods

**Lemma F.1** ($\beta \to \infty$ recovers top selection on the finite set). *Let $S_{\max} := \{i : s(x_i) = \max_{1 \leq j \leq n} s(x_j)\}$. Then, as $\beta \to \infty$, the weights in (17) satisfy*

$$w_i(\beta) \to \begin{cases} 1/|S_{\max}|, & i \in S_{\max}, \\ 0, & i \notin S_{\max}. \end{cases} \tag{159}$$

*Consequently, $\hat{q}_{n,\beta}$ converges to the uniform distribution over the maximizers of $s$ within the finite set.*

*Proof.* Let $s_{\max} := \max_j s(x_j)$. For any $i$,

$$w_i(\beta) = \frac{\exp(\beta(s_i - s_{\max}))}{\sum_{j=1}^n \exp(\beta(s_j - s_{\max}))}. \tag{160}$$

If $i \notin S_{\max}$ then $s_i - s_{\max} < 0$, so $\exp(\beta(s_i - s_{\max})) \to 0$ as $\beta \to \infty$. If $i \in S_{\max}$ then $\exp(\beta(s_i - s_{\max})) = 1$. Hence the denominator tends to $|S_{\max}|$ and the stated limit follows. $\square$

The above lemma shows that empirical tilting becomes an extreme selection rule as $\beta$ grows: it approaches a top-sample interpretation on the observed finite set. Thus, while exponential reweighting avoids an explicit hard threshold, in finite samples it can still exhibit a sharp "winner-takes-most" behavior, which is a discrete analogue of boundary/selection bias.

## F.2. Proof for The Optimization with Regularization Methods

**Lemma F.2** (Optimization with regularization equals MAP of a Gibbs distribution). *Fix a reference measure $\mu$ on $\mathcal{X}$. Let $u : \mathcal{X} \to \mathbb{R}$ be a score and $\mathcal{R} : \mathcal{X} \to \mathbb{R}$ be a regularizer. Consider the optimization problem*

$$x^\star \in \arg\max_{x \in \mathcal{X}} \big( u(x) - \lambda \mathcal{R}(x) \big). \tag{161}$$

*Define an (unnormalized) Gibbs density w.r.t. $\mu$:*

$$\pi(x) \propto \exp\big( u(x) - \lambda \mathcal{R}(x) \big). \tag{162}$$

*Then any maximizer $x^\star$ of (161) is a MAP point (mode) of $\pi$, i.e. $x^\star \in \arg\max_x \pi(x)$.*

*Proof of Lemma F.2.* Taking logarithms of (162) gives

$$\log \pi(x) = u(x) - \lambda \mathcal{R}(x) - \log Z, \tag{163}$$

where $Z = \int \exp(u(x) - \lambda \mathcal{R}(x)) \, \mathrm{d}\mu(x)$ is a constant independent of $x$. Therefore

$$\arg \max_{x} \pi(x) = \arg \max_{x} \log \pi(x) = \arg \max_{x} \big( u(x) - \lambda \mathcal{R}(x) \big), \tag{164}$$

which is exactly (161). Hence any optimizer is a MAP point of $\pi$. $\qquad\square$

### F.3. Typical Set Mismatch in Optimization with Regularization

In the Opt+Reg paradigm, the explanation distribution can be viewed as a Gibbs law $\pi(x) \propto \exp\big(s(x) - \lambda R(x)\big)$ (or its tempered variant), and the produced visualization is a MAP point $x_{\mathrm{MAP}}$ (denoted $x^{\mathrm{opt}}$ in the main text). This entails an inherent *mode-seeking* bias: in high dimensions, modes can be exponentially unrepresentative of typical samples. The Gaussian example below provides a canonical witness of this phenomenon.

**Definition F.3** ($\varepsilon$-typical set)**.** Fix $q \in \mathcal{P}(\mathcal{X})$ and $\varepsilon \in (0,1)$. Any measurable set $T_\varepsilon(q) \subseteq \mathcal{X}$ satisfying $q\big(T_\varepsilon(q)\big) \geq 1 - \varepsilon$ is called an $\varepsilon$-typical region of $q$.

Typical regions provide a minimal, distributional notion of "representative samples". Indeed, for any bounded measurable statistic $f : \mathcal{X} \to \mathbb{R}$ and any $\varepsilon$-typical region $T_\varepsilon(q)$,

$$\Big| \mathbb{E}_q[f(X)] - \mathbb{E}_q\big[f(X) \mid X \in T_\varepsilon(q)\big] \Big| \ \leq \ 2\varepsilon \, \|f\|_\infty. \tag{165}$$

Thus, when explanations are communicated via samples, focusing on typical regions preserves distributional semantics up to $O(\varepsilon)$.

**Lemma F.4** (MAP can be exponentially unrepresentative in high dimension)**.** *Let $X \sim \mathcal{N}(0, I_d)$ on $\mathbb{R}^d$. The MAP (mode) of the density is at $0$. However, the typical set concentrates on a thin shell of radius $\|X\| \approx \sqrt{d}$: for any $\alpha \in (0,1)$, letting $\delta := 1 - \alpha^2 \in (0,1)$,*

$$\mathbb{P}\big(\|X\| \leq \alpha\sqrt{d}\big) = \mathbb{P}\big(\|X\|^2 \leq (1-\delta)d\big) \ \leq \ \exp\Big(-\frac{\delta^2}{4} d\Big) = \exp\Big(-\frac{(1-\alpha^2)^2}{4} d\Big). \tag{166}$$

*In particular, taking $\alpha = \frac{1}{2}$ gives $\mathbb{P}(\|X\| \leq \frac{1}{2}\sqrt{d}) \leq \exp(-9d/64)$.*

*Proof.* Let $Y := \|X\|^2 = \sum_{i=1}^d X_i^2$, so $Y \sim \chi_d^2$. For any $\lambda > 0$ and any $a > 0$, Markov's inequality gives

$$\mathbb{P}(Y \leq a) = \mathbb{P}(e^{-\lambda Y} \geq e^{-\lambda a}) \leq e^{\lambda a} \, \mathbb{E}[e^{-\lambda Y}]. \tag{167}$$

The Laplace transform of $\chi_d^2$ yields $\mathbb{E}[e^{-\lambda Y}] = (1 + 2\lambda)^{-d/2}$ for $\lambda > 0$, hence

$$\mathbb{P}(Y \leq a) \leq \exp\Big(\lambda a - \frac{d}{2} \log(1 + 2\lambda)\Big). \tag{168}$$

Set $a = (1-\delta)d$ with $\delta \in (0,1)$ and choose $\lambda = \frac{\delta}{2(1-\delta)}$, which implies $1 + 2\lambda = \frac{1}{1-\delta}$. Substituting gives

$$\mathbb{P}\big(Y \leq (1-\delta)d\big) \leq \exp\Big(\frac{\delta}{2(1-\delta)}(1-\delta)d - \frac{d}{2} \log \frac{1}{1-\delta}\Big) = \exp\Big(\frac{d}{2}\big(\delta + \log(1-\delta)\big)\Big). \tag{169}$$

Finally, since $\log(1-\delta) \leq -\delta - \delta^2/2$ for $\delta \in (0,1)$ (from the Taylor series with alternating remainder, or by integrating $(1-x)^{-1} \geq 1 + x$ on $[0,\delta]$), we have $\delta + \log(1-\delta) \leq -\delta^2/2$. Therefore

$$\mathbb{P}\big(Y \leq (1-\delta)d\big) \leq \exp\Big(-\frac{\delta^2}{4}d\Big), \tag{170}$$

which proves (166). $\qquad\square$

Lemma F.4 shows that even when the induced Gibbs distribution $\pi$ is benign, its MAP can lie in a region of exponentially small probability mass. Thus "one optimized image" can be a poor proxy for the distributional semantics of a feature.

**Assumption F.5** (Noisy-MAP approximation). Let $q \in \mathcal{P}(\mathbb{R}^d)$ be the distribution induced by Opt+Reg (e.g., a Gibbs law $\pi$ or a temperature-smoothed variant $q_T$), and let $x_{\mathrm{MAP}} \in \arg\max_x q(x)$. An approximate optimizer is said to satisfy an $(r, \eta)$ Noisy-MAP condition if its output $\hat{X}$ obeys

$$\mathbb{P}\big(\|\hat{X} - x_{\mathrm{MAP}}\| \leq r\big) \; \geq \; 1 - \eta, \tag{171}$$

for some radius $r > 0$ and failure probability $\eta \in [0, 1)$.

**Lemma F.6** (Noisy-MAP cannot cover typical mass unless the error scale is large). *Let $q \in \mathcal{P}(\mathbb{R}^d)$ and let $\rho := \mathcal{L}(\hat{X})$ be the output distribution of an approximate optimizer satisfying Assumption F.5. Then for any $r > 0$,*

$$\mathrm{TV}(q, \rho) \; \geq \; (1 - \eta) \; - \; q\big(B(x_{\mathrm{MAP}}, r)\big), \tag{172}$$

*where $B(x, r) := \{z : \|z - x\| \leq r\}$. In particular, if $q(B(x_{\mathrm{MAP}}, r)) \leq \varepsilon$, then $\mathrm{TV}(q, \rho) \geq 1 - \eta - \varepsilon$ and the complement $T_\varepsilon(q) := B(x_{\mathrm{MAP}}, r)^c$ is an $\varepsilon$-typical region of $q$ that the optimizer hits with probability at most $\eta$.*

*Moreover, for $q = \mathcal{N}(0, I_d)$, $x_{\mathrm{MAP}} = 0$, and $r = \alpha\sqrt{d}$ with $\alpha \in (0, 1)$, Proposition F.4 implies*

$$q\big(B(0, r)\big) \; \leq \; \exp\left(-\frac{(1 - \alpha^2)^2}{4} d\right), \tag{173}$$

*hence $\mathrm{TV}(q, \rho) \geq 1 - \eta - \exp\big(-\frac{(1-\alpha^2)^2}{4}d\big)$.*

*Proof.* For any measurable set $A$, $\mathrm{TV}(q, \rho) \geq |q(A) - \rho(A)|$. Take $A = B(x_{\mathrm{MAP}}, r)$. Assumption F.5 gives $\rho(A) \geq 1 - \eta$, hence

$$\mathrm{TV}(q, \rho) \; \geq \; \rho(A) - q(A) \; \geq \; (1 - \eta) - q\big(B(x_{\mathrm{MAP}}, r)\big). \tag{174}$$

If $q(B) \leq \varepsilon$, then $T_\varepsilon(q) := B^c$ satisfies $q(T_\varepsilon(q)) \geq 1 - \varepsilon$ and $\rho(T_\varepsilon(q)) \leq \eta$. The Gaussian specialization follows by substituting (166). $\square$

Lemma F.6 formalizes a key limitation of optimization with regularization: bounded optimization error merely randomizes a neighborhood of the mode, which can carry exponentially small probability mass in high dimension. To achieve non-negligible coverage of typical regions, the perturbation scale must be comparable to the typical radius (e.g., $r = \Theta(\sqrt{d})$ for isotropic Gaussians), at which point the procedure begins to resemble an explicit sampler rather than a point estimator.

### F.4. Proof for The Prompt-Guided Diffusion Sampling

**Theorem F.7** (Prompt information bottleneck). *Fix a baseline distribution $p_\theta$ on $\mathcal{X}$ and an intrinsic score $s : \mathcal{X} \to \mathbb{R}$. Let $q_\beta^{\mathrm{KLSC}}$ be the (unrestricted) KLSC solution satisfying $\mathbb{E}_{q_\beta^{\mathrm{KLSC}}}[s(X)] = m$. Then, for any prompt-restricted family $\mathbf{Q}_{\mathrm{prompt}} \subset \mathcal{P}(\mathcal{X})$,*

$$\min_{\substack{q \in \mathbf{Q}_{\mathrm{prompt}} \\ \mathbb{E}_q[s] \geq m}} \mathrm{KL}(q \| p_\theta) \; \geq \; \mathrm{KL}\big(q_\beta^{\mathrm{KLSC}} \| p_\theta\big) \; + \; \inf_{\substack{q \in \mathbf{Q}_{\mathrm{prompt}} \\ \mathbb{E}_q[s] \geq m}} \mathrm{KL}\big(q \| q_\beta^{\mathrm{KLSC}}\big). \tag{175}$$

*Proof.* Fix any $q$ such that $q \in \mathbf{Q}_{\mathrm{prompt}}$ and $\mathbb{E}_q[s] \geq m$. By the chain rule for KL divergence,

$$\mathrm{KL}(q \| p_\theta) = \mathrm{KL}\big(q \| q_\beta^{\mathrm{KLSC}}\big) + \int q(x) \log \frac{q_\beta^{\mathrm{KLSC}}(x)}{p_\theta(x)} \, \mathrm{d}x. \tag{176}$$

Since $q_\beta^{\mathrm{KLSC}}(x) \propto p_\theta(x) \exp(\beta s(x))$, we have

$$\log \frac{q_\beta^{\mathrm{KLSC}}(x)}{p_\theta(x)} = \beta s(x) - \log Z(\beta), \qquad Z(\beta) = \mathbb{E}_{p_\theta}\Big[e^{\beta s(X)}\Big]. \tag{177}$$

Substituting (177) into (176) yields

$$\mathrm{KL}(q \| p_\theta) = \mathrm{KL}\big(q \| q_\beta^{\mathrm{KLSC}}\big) + \beta \, \mathbb{E}_q[s(X)] - \log Z(\beta). \tag{178}$$

Applying (178) to $q_\beta^{\text{KLSC}}$ and using $\mathbb{E}_{q_\beta^{\text{KLSC}}}[s(X)] = m$ gives

$$\text{KL}\big(q_\beta^{\text{KLSC}}\|p_\theta\big) = \beta m - \log Z(\beta). \tag{179}$$

Combining (178)–(179) and using $\mathbb{E}_q[s(X)] \geq m$ (with $\beta \geq 0$) yields the pointwise bound

$$\text{KL}(q\|p_\theta) \geq \text{KL}\big(q_\beta^{\text{KLSC}}\|p_\theta\big) + \text{KL}\big(q\|q_\beta^{\text{KLSC}}\big). \tag{180}$$

Taking the infimum over all feasible $q \in \mathbf{Q}_{\text{prompt}}$ proves (175). $\qquad\square$

# G. The Faithfulness of EnergyDPS

This appendix formalizes the relationship between the **KLSC** induced distribution $q_\beta^{\text{KLSC}} \propto p_\theta \exp(\beta s)$ and **EnergyDPS**. At a high level, we prove an *ideal* statement: *if* one can implement the reverse-time diffusion dynamics for the tilted marginals $\{q_{\beta,t}\}_{t \in [0,T]}$, then the resulting sampler is *exact* for the KLSC tilt. We then explain how EnergyDPS provides a scalable *plug-in* approximation of this ideal sampler, and why CEP-style approaches (Lu et al., 2023) are incompatible with our paradigm.

## G.1. Setup: score-based diffusion and time reversal

We adopt the standard score-based diffusion formalism on $\mathcal{X} = \mathbb{R}^d$. Throughout, $\nabla$ and $\Delta$ denote the gradient and Laplacian w.r.t. the spatial variable.

**Assumption G.1** (Forward diffusion and regularity). Assume the diffusion prior $p_\theta$ is realized as the $t = 0$ marginal of a forward SDE

$$\mathrm{d}X_t = f(X_t, t)\,\mathrm{d}t + g(t)\,\mathrm{d}W_t, \qquad t \in [0, T], \tag{181}$$

where $g(t) > 0$ depends only on $t$. Let $p_{\theta,t}$ denote the density of $X_t$ when $X_0 \sim p_\theta$, and assume each $p_{\theta,t}$ is $C^2$ and strictly positive on $\mathbb{R}^d$.

The following lemma is a standard time-reversal statement (e.g., Anderson/Haussmann–Pardoux type results) specialized to scalar diffusion $g(t)$.

**Lemma G.2** (Reverse-time SDE with prescribed marginals). *Let $\nu$ be any initial distribution on $\mathbb{R}^d$, and let $p_t^\nu$ be the density of the forward SDE (181) at time $t$ when $X_0 \sim \nu$. Assume $p_t^\nu$ is $C^2$ and strictly positive for all $t \in [0, T]$. Define $\tilde{p}_t(x) := p_{T-t}^\nu(x)$ and consider the SDE (run forward in $t \in [0, T]$)*

$$\mathrm{d}Y_t = \Big(-f(Y_t, T-t) + g(T-t)^2 \nabla \log p_{T-t}^\nu(Y_t)\Big)\mathrm{d}t + g(T-t)\,\mathrm{d}\bar{W}_t, \tag{182}$$

*initialized with $Y_0 \sim p_T^\nu$. Then for all $t \in [0, T]$, the law of $Y_t$ has density $\tilde{p}_t = p_{T-t}^\nu$. In particular, $Y_T \sim \nu$.*

*Proof.* The forward SDE (181) implies that $p_t^\nu$ satisfies the Fokker–Planck equation

$$\partial_t p_t^\nu = -\nabla \cdot \big(f(\cdot, t)\,p_t^\nu\big) + \frac{1}{2}g(t)^2\,\Delta p_t^\nu. \tag{183}$$

Define $\tilde{p}_t(x) := p_{T-t}^\nu(x)$; then by the chain rule,

$$\partial_t \tilde{p}_t(x) = -\big(\partial_\tau p_\tau^\nu(x)\big)\big|_{\tau = T-t}. \tag{184}$$

Let $b(x, t) := -f(x, T-t) + g(T-t)^2 \nabla \log p_{T-t}^\nu(x)$ and $\tilde{g}(t) := g(T-t)$. The Fokker–Planck equation associated with (182) is

$$\partial_t \rho_t = -\nabla \cdot \big(b(\cdot, t)\,\rho_t\big) + \frac{1}{2}\tilde{g}(t)^2\,\Delta \rho_t. \tag{185}$$

We verify that $\rho_t = \tilde{p}_t$ satisfies (185). Indeed,

$$b(\cdot, t)\tilde{p}_t = -f(\cdot, T-t)\,p_{T-t}^\nu + g(T-t)^2\,\nabla p_{T-t}^\nu, \tag{186}$$

hence

$$-\nabla \cdot \big(b(\cdot,t)\tilde{p}_t\big) = \nabla \cdot \big(f(\cdot,T-t)\,p^\nu_{T-t}\big) - g(T-t)^2\,\Delta p^\nu_{T-t}. \tag{187}$$

Adding the diffusion term $\frac{1}{2}\tilde{g}(t)^2\Delta\tilde{p}_t = \frac{1}{2}g(T-t)^2\Delta p^\nu_{T-t}$ gives

$$-\nabla \cdot \big(b(\cdot,t)\tilde{p}_t\big) + \frac{1}{2}\tilde{g}(t)^2\Delta\tilde{p}_t = \nabla \cdot \big(f(\cdot,T-t)\,p^\nu_{T-t}\big) - \frac{1}{2}g(T-t)^2\,\Delta p^\nu_{T-t}. \tag{188}$$

On the other hand, combining (184) with (183) at $\tau = T - t$ yields

$$\partial_t\tilde{p}_t = \nabla \cdot \big(f(\cdot,T-t)\,p^\nu_{T-t}\big) - \frac{1}{2}g(T-t)^2\,\Delta p^\nu_{T-t}, \tag{189}$$

which matches (185). Therefore $\tilde{p}_t$ satisfies the Fokker–Planck equation of (182). Uniqueness of smooth solutions implies $\rho_t = \tilde{p}_t$ for all $t$. $\qquad\square$

## G.2. KLSC tilt and the ideal guided reverse dynamics

Fix an intrinsic score $s : \mathcal{X} \to \mathbb{R}$ and a guidance strength $\beta \geq 0$. Assume the tilt is normalizable, i.e., $Z(\beta) := \mathbb{E}_{p_\theta}[e^{\beta s(X)}] < \infty$.

**Theorem G.3** (Guided reverse SDE targets the KLSC tilt). *Fix $\beta \geq 0$ and define the KLSC tilt at $t = 0$ by*

$$q_\beta(x) := q_\beta^{\mathrm{KLSC}}(x) \propto p_\theta(x)\exp\big(\beta s(x)\big). \tag{190}$$

*Let $q_{\beta,t}$ denote the density at time $t$ obtained by evolving $X_0 \sim q_\beta$ through the forward SDE (181), and assume each $q_{\beta,t}$ is $C^2$ and strictly positive. Define the time-dependent density-ratio potential*

$$E_t(x) := -\frac{1}{\beta}\log\frac{q_{\beta,t}(x)}{p_{\theta,t}(x)} \quad (\beta > 0), \qquad \text{and} \qquad E_t(x) \equiv 0 \ (\beta = 0). \tag{191}$$

*Then for all $t \in [0,T]$,*

$$q_{\beta,t}(x) \propto p_{\theta,t}(x)\exp\big(-\beta E_t(x)\big), \qquad \nabla\log q_{\beta,t}(x) = \nabla\log p_{\theta,t}(x) - \beta\nabla E_t(x). \tag{192}$$

*Moreover, the reverse-time SDE associated with $\nu = q_\beta$ in Lemma G.2 can be implemented using the* modified score

$$\nabla_x\log p_{\theta,t}(x) \quad \leadsto \quad \nabla_x\log p_{\theta,t}(x) - \beta\nabla_x E_t(x). \tag{193}$$

*When initialized from $Y_0 \sim q_{\beta,T}$ and run to $t = T$, it produces $Y_T \sim q_\beta$.*

*Proof.* The proportionality in (192) is immediate from (191): for $\beta > 0$, $q_{\beta,t}(x) = p_{\theta,t}(x)\exp(-\beta E_t(x))$; for $\beta = 0$ it is trivial. Taking logarithms and differentiating yields the score identity $\nabla\log q_{\beta,t} = \nabla\log p_{\theta,t} - \beta\nabla E_t$.

Now apply Lemma G.2 with $\nu = q_\beta$. The lemma states that the reverse SDE (182) whose drift uses $\nabla\log q_{\beta,T-t}$ generates marginals $q_{\beta,T-t}$ and, in particular, returns $q_{\beta,0} = q_\beta$ at the terminal time. By (192), $\nabla\log q_{\beta,t}$ equals the modified score in (193). Therefore implementing the reverse SDE using the baseline score $\nabla\log p_{\theta,t}$ plus the density-ratio correction $-\beta\nabla E_t$ realizes the reverse process for $\{q_{\beta,t}\}$ and outputs samples from $q_\beta$. $\qquad\square$

## G.3. From the ideal statement to EnergyDPS

Theorem G.3 clarifies the *mathematically exact* route to sample from the KLSC tilt: one must correct the base score $\nabla\log p_{\theta,t}$ by the time-dependent term $-\beta\nabla E_t$, where $E_t$ encodes the (unknown) density ratio $q_{\beta,t}/p_{\theta,t}$. EnergyDPS provides a scalable *plug-in* approximation to this correction.

*Remark* G.4 (Connection to EnergyDPS, plug-in guidance, and contrastive energy prediction (CEP)). Theorem G.3 is an *ideal* statement: exact sampling from the KLSC tilt $q_\beta \propto p_\theta e^{\beta s}$ can be realized by a reverse-time SDE if one has access to the time-dependent correction $-\beta\nabla E_t$, where $E_t(x) = -(1/\beta)\log\big(q_{\beta,t}(x)/p_{\theta,t}(x)\big)$. In general, $E_t$ is intractable because $q_{\beta,t}$ is unknown.

EnergyDPS adopts a *plug-in* approximation that avoids modeling $E_t$ explicitly. At each reverse step it forms a denoised estimate $\hat{x}_0 = \hat{x}_0(x_t, t)$ (Algorithm 1) and applies guidance via $\nabla_{x_t}s(\hat{x}_0)$. This guidance can be viewed as a practical

surrogate for the ideal correction in (193): it injects the intrinsic signal directly into the reverse dynamics while retaining the diffusion prior. A key advantage is that this construction is *score-agnostic*: given any differentiable intrinsic score $s$, one can instantiate a sampler for the corresponding KLSC tilt without additional training.

This differs fundamentally from contrastive energy prediction (CEP) (Lu et al., 2023). CEP addresses the same obstacle—the unknown intermediate-time density-ratio potential—by *learning* a time-dependent predictor that approximates the diffused energy/ratio term $E_t(\cdot)$ (equivalently, its score contribution) from noisy states $(x_t, t)$ via a contrastive objective. Crucially, this predictor is *energy-specific*: changing the target tilt (i.e., changing $s$ or $\beta$) changes the entire family $\{E_t\}_{t \in [0,T]}$, requiring retraining or re-fitting for each new energy.

Such a requirement is incompatible with our interpretability setting. Here the "energy" is not a single fixed objective but an *intrinsic score family* $\{s_k\}$ indexed by SAE features (often thousands), together with interactive choices of reductions and target moments. Training a separate CEP-style predictor for each $(k, \beta)$ would be prohibitively expensive and would defeat the goal of a *post-hoc, plug-and-play* interpretability paradigm. EnergyDPS therefore trades exactness for generality: it reuses the pretrained diffusion score $\nabla \log p_{\theta,t}$ and injects the intrinsic signal through $\nabla s(\hat{x}_0)$, enabling scalable visualization across many SAE features without training any additional energy models.

### G.4. Discrete-time viewpoint (relation to Algorithm 1)

In practice we implement reverse-time sampling by discretizing the reverse SDE and replacing the true base score $\nabla \log p_{\theta,t}$ with a learned score network (or an oracle score in toy experiments). Algorithm 1 can be read as an Euler-type update of the base reverse dynamics, augmented by the plug-in guidance term $\zeta_i \beta \nabla_{x_i} s(\hat{x}_0)$. Under exact scores and vanishing step size, the un-guided part recovers sampling from the base prior $p_\theta$; the additional guidance term realizes an explicit drift perturbation that steers the trajectory toward the KLSC tilt. We refer to Appendix I for the oracle-score construction used in the GMM experiments.

## H. Experimental Details for Section 2.2.2

This appendix provides implementation details for the toy validation summarized in Section 2.2.2 (Table 2, Fig. 1). We compare hard truncation (Definition 2.1) and KLSC (Definition 2.2) under *matched expected activation* $\mathbb{E}[s(X)]$, and evaluate boundary bias, threshold instability, and interpretability.

### H.1. Baseline distribution and intrinsic score

**Baseline prior $p$.** We use a symmetric 2D Gaussian mixture model with four modes:

$$p(x) = \frac{1}{4} \sum_{\mu \in \{(\pm 2, \pm 2)\}} \mathcal{N}(x; \mu, \sigma^2 I_2), \qquad \sigma = 0.8. \tag{194}$$

Sampling from $p$ is straightforward by first sampling a mode uniformly, then sampling from the corresponding Gaussian.

**Intrinsic score $s(x)$.** Let $x = (x_1, x_2) \in \mathbb{R}^2$ and define the polar angle $\theta(x) = \operatorname{atan2}(x_2, x_1) \in (-\pi, \pi]$. Fix $\theta_0 = \pi/4$ and define the angular deviation score

$$s(x) := 4\big[1 - \cos(\theta(x) - \theta_0)\big]. \tag{195}$$

Thus $s(x)$ is minimized at $\theta(x) = \theta_0$ and increases as the direction deviates from the diagonal.

### H.2. Induced distributions

**Hard constraint (truncation).** For a truncation threshold $m' \in \mathbb{R}$, define the event $E_{m'} := \{x \in \mathcal{X} : s(x) > m'\}$ and the induced hard-constraint distribution

$$q_{m'}^{\text{hard}}(x) = p(x \mid E_{m'}) = \frac{p(x)\mathbf{1}_{E_{m'}}(x)}{p(E_{m'})}. \tag{196}$$

In practice, we sample $q_{m'}^{\text{hard}}$ via rejection sampling from $p$.

**KLSC (exponential tilting).** For $\beta \geq 0$, the KLSC solution has density

$$q_\beta^{\text{KLSC}}(x) = \frac{p(x) \exp(\beta s(x))}{Z(\beta)}, \qquad Z(\beta) = \mathbb{E}_{X \sim p}\big[\exp(\beta s(X))\big]. \tag{197}$$

We sample from $q_\beta^{\text{KLSC}}$ using self-normalized importance sampling with proposal $p$.

### H.3. Matched-moment calibration protocol

To isolate differences due to the constraint formulation (rather than different faithfulness levels), we sweep a set of target expected activations $\{m\}$ (the same values reported in Table 2), and for each $m$ calibrate the hard and soft families to satisfy

$$\mathbb{E}_{q_{m'}^{\text{hard}}}[s(X)] = \mathbb{E}_{q_\beta^{\text{KLSC}}}[s(X)] = m. \tag{198}$$

**Calibrating the hard threshold $m'$.** Define

$$\mu_{\text{hard}}(m') := \mathbb{E}_{X \sim p}[s(X) \mid s(X) > m']. \tag{199}$$

We solve $\mu_{\text{hard}}(m') = m$ by bisection over $m'$ using Monte Carlo estimates under $p$. (We use a bracket $[m'_{\min}, m'_{\max}]$ that covers the score range observed under $p$.)

**Calibrating the tilt strength $\beta$.** Define

$$\mu_{\text{KLSC}}(\beta) := \mathbb{E}_{X \sim q_\beta^{\text{KLSC}}}[s(X)]. \tag{200}$$

We solve $\mu_{\text{KLSC}}(\beta) = m$ by bisection over $\beta \geq 0$. Using samples $\{x_i\}_{i=1}^N \sim p$, we estimate $\mu_{\text{KLSC}}(\beta)$ via importance weights $w_i(\beta) = \exp(\beta s(x_i))$:

$$\widehat{\mu}_{\text{KLSC}}(\beta) = \frac{\sum_{i=1}^N w_i(\beta)\, s(x_i)}{\sum_{i=1}^N w_i(\beta)}. \tag{201}$$

### H.4. Metrics

**Boundary bias.** We report boundary mass (Definition 2.3) with margin $\delta = 0.05$:

$$\text{BM}_{m',\delta}(q) = \frac{q(B_{m',\delta})}{q(E_{m'})}, \qquad B_{m',\delta} := \{x : \ m' < s(x) < m' + \delta\}. \tag{202}$$

**Threshold instability.** For a small perturbation $\epsilon > 0$ (we use a fixed $\epsilon = 0.08$ throughout all runs), we re-calibrate each constraint family to match $m$ and $m + \epsilon$ in (198), producing $q_m$ and $q_{m+\epsilon}$. We measure the induced shift by

$$\text{TV}(q_m, q_{m+\epsilon}) = \frac{1}{2} \int_{\mathcal{X}} \big| q_m(x) - q_{m+\epsilon}(x) \big| \, dx. \tag{203}$$

(We use TV as a finite proxy because the KL divergence between truncated distributions can become infinite under small threshold perturbations.)

## I. Implementation Details for Toy Sampling Validation

This appendix provides implementation details for the toy sampling comparison shown in Fig. 3. The goal is to compare samplers under a *matched-faithfulness* protocol, i.e., all methods are calibrated to match the same target moment $\mathbb{E}[s] \in \{5, 6, 7\}$, and then evaluated by their deviation from the reference prior $p$.

**Baseline prior $p$ and intrinsic score $s$.** We use the same 2D GMM prior as in the toy constraint validation. Let $x \in \mathbb{R}^2$ and define

$$p(x) = \frac{1}{K} \sum_{k=1}^K \mathcal{N}(x; \mu_k, \sigma^2 I_2), \qquad K = 4, \ \ \mu_k \in \{(\pm 2, \pm 2)\}, \ \ \sigma = 0.8. \tag{204}$$

The intrinsic score is the angular deviation score

$$s(x) = 4\Big(1 - \cos\big(\theta(x) - \tfrac{\pi}{4}\big)\Big), \qquad \theta(x) = \arctan 2(x_2, x_1) \in (-\pi, \pi]. \tag{205}$$

**Matched-faithfulness calibration.** For each target level $m \in \{5, 6, 7\}$, each method is calibrated to satisfy $\widehat{\mathbb{E}}[s] \approx m$ using the *same sampling budget* across methods. We use a tolerance $\varepsilon_m$ and stop calibration when $|\widehat{\mathbb{E}}[s] - m| \leq \varepsilon_m$. Each method has a method-specific knob: top-$k$ size for retrieval, regularization weight $\lambda$ for OR, and guidance strength $\gamma$ for EnergyDPS (Exact budgets and tolerances are fixed across all $m$; see the released code).

### I.1. Methodology I: Top-activation retrieval (finite pool)

We draw a candidate pool $\mathbf{D}_n = \{x_i\}_{i=1}^n$ i.i.d. from $p$ and compute $\{s(x_i)\}$. We sort the pool by $s(x_i)$ in descending order and select the top-$k$ points with largest scores. The integer $k$ is chosen (via a prefix search over the sorted list) such that the empirical mean of $s$ over the selected set matches the target:

$$\widehat{\mathbb{E}}_{\text{top-}k}[s] = \frac{1}{k} \sum_{i=1}^k s(x_{(i)}) \approx m, \tag{206}$$

where $x_{(i)}$ is the $i$-th largest-scoring element. We treat the selected set as samples from retrieval. This finite-pool procedure mimics a finite dataset or bounded search budget used in practice.

### I.2. Methodology II: Optimization with regularization (OR)

OR generates samples via multi-start optimization from random initializations in a bounded box (e.g., $[-5, 5]^2$), using the objective in Eq. (18). In this toy experiment, we use the regularizer

$$R(x) = -x|_2, \tag{207}$$

where $x|_2$ is the 2-nd index of $x$, i.e., the index in y-axis. And, we tune the regularization weight $\lambda$ so that the resulting optimized solutions (aggregated over random restarts) satisfy $\widehat{\mathbb{E}}[s] \approx m$. We treat the collection of optimized solutions as samples from OR.

### I.3. Methodology III: EnergyDPS with an oracle diffusion score

EnergyDPS requires a diffusion score model $u_\theta(x_i, i) \approx \nabla_{x_i} \log p_i(x_i)$ at each noise level $i$. In the toy GMM setting, we use a closed-form oracle score.

**Forward noising and the noised prior $p_i$.** We adopt a standard DDPM forward process:

$$x_i = \sqrt{\bar{\alpha}_i}\, x_0 + \sqrt{1 - \bar{\alpha}_i}\, \varepsilon, \qquad x_0 \sim p, \ \varepsilon \sim \mathcal{N}(0, I_2), \tag{208}$$

where $\alpha_i = 1 - \beta_i^{\text{ddpm}}$ and $\bar{\alpha}_i = \prod_{j=1}^i \alpha_j$. Since Gaussian convolution preserves mixture structure, the marginal $p_i$ remains a $K$-component GMM:

$$p_i(x) = \frac{1}{K} \sum_{k=1}^K \mathcal{N}(x;\ \sqrt{\bar{\alpha}_i}\mu_k,\ \Sigma_i), \qquad \Sigma_i := (\bar{\alpha}_i \sigma^2 + (1 - \bar{\alpha}_i)) I_2. \tag{209}$$

**Closed-form oracle score $u_i(x) = \nabla \log p_i(x)$.** Let $\phi_{i,k}(x) := \mathcal{N}(x; \sqrt{\bar{\alpha}_i}\mu_k, \Sigma_i)$ and define responsibilities

$$w_{i,k}(x) := \frac{\phi_{i,k}(x)}{\sum_{j=1}^K \phi_{i,j}(x)}, \qquad \sum_{k=1}^K w_{i,k}(x) = 1. \tag{210}$$

Then the score admits the closed form

$$u_i(x) := \nabla_x \log p_i(x) = \sum_{k=1}^K w_{i,k}(x)\, \nabla_x \log \phi_{i,k}(x) = \Sigma_i^{-1} \left( \sum_{k=1}^K w_{i,k}(x)\, \sqrt{\bar{\alpha}_i}\mu_k - x \right). \tag{211}$$

In Algorithm 1 we set $u_\theta(\cdot, i) \equiv u_i(\cdot)$.

**Intrinsic guidance and calibration.** At each reverse step, EnergyDPS forms a predicted clean image $\hat{x}_0 = \hat{x}_0(x_i, i)$ and injects intrinsic guidance using $\nabla_{x_i} s(\hat{x}_0)$ with step size $\{\zeta_i\}$, as in Algorithm 1. The guidance strength $\gamma$ is calibrated so that the resulting samples satisfy $\widehat{\mathbb{E}}[s] \approx m$ for each $m \in \{5, 6, 7\}$.

### I.4. Estimating $\mathrm{KL}(q\|p)$ from samples

For sample-based methods, the induced distribution is represented by a finite set of samples and is not absolutely continuous with respect to $p$, making $\mathrm{KL}(q\|p)$ ill-defined if interpreted literally. To obtain a well-defined and comparable metric, we compute KL on a *smoothed density estimate* shared across methods. Given samples $\{x_j\}_{j=1}^N$ from a method, we form a KDE

$$\tilde{q}(x) = \frac{1}{N} \sum_{j=1}^N \mathcal{N}(x; x_j, h^2 I_2), \tag{212}$$

with a fixed bandwidth $h$ (the same $h$ for all methods). We then approximate $\mathrm{KL}(\tilde{q}\|p)$ by numerical integration on a fixed grid over the displayed domain (same grid and integration rule for all methods). This estimator is used to produce the values reported in Figure 3.

## J. Implementation Details for DINOv3 Experiments

This appendix specifies the exact experimental configurations used in Section 3, including (i) transcoder training on DINOv3 activations, (ii) intrinsic-score construction and stabilization, and (iii) method-specific settings for activation maximization (MACO (Fel et al., 2023a)), proxy-guided generation (DiffExplainer (Pennisi et al., 2025)), and EnergyDPS.

### J.1. Transcoder (SAE Variant) Training

**Backbone and hook point.** We use DINOv3 ResNeXt Large (Siméoni et al., 2025) and extract activations at the 20-th layer of Stage 2. For an input image $x \in \mathcal{X}$, the hooked feature map has shape $d \times H \times W = 768 \times 16 \times 16$. We sample 1,000,000 images from ImageNet-1K, resized to $256 \times 256$.

**Training objective and hyperparameters.** We train a transcoder (Appendix C) with top-64 sparsification and reconstruction loss only (no auxiliary losses). We use learning rate $1 \times 10^{-4}$. Training is continued until the explained-variance target $\mathrm{EV} = 0.85$ is reached.

### J.2. Intrinsic Score from Transcoder Activations: Mix Reduction

Let $A_k(x) \in \mathbb{R}^{H \times W}$ denote the spatial activation map of transcoder feature $k$ on input $x$ (after encoding/decoding as used in our implementation). Unless otherwise stated, we reduce $A_k(x)$ to a scalar intrinsic score via a *mix* operator:

$$s_k(x) = \alpha \cdot \max_{u,v} A_k(x)_{u,v} + (1 - \alpha) \cdot \frac{1}{HW} \sum_{u,v} \max\big(A_k(x)_{u,v}, 0\big), \tag{213}$$

with $\alpha = 0.3$. This mixed reduction empirically improves stability compared to pure max-reduction, and helps suppress repetitive texture artifacts.

**Guidance control via cap strategy.** To avoid occasional extreme activations dominating guidance, we adopt a `cap_value` control: when the intrinsic score exceeds a pre-specified cap (relative to the target activation level used for faithfulness matching), we *disable* the intrinsic-score supervision term for that step (i.e., we set the guidance contribution from $\nabla s_k(\cdot)$ to zero). This prevents over-steering once the constraint is already met.

### J.3. Baselines

#### J.3.1. MACO: MASKED AUGMENTATION–CONSISTENCY OPTIMIZATION

We instantiate the optimization-based paradigm using **MACO** (Fel et al., 2023a), which synthesizes an input by directly optimizing pixels to increase the target intrinsic score while enforcing stability under simple transformations. Concretely, starting from random initialization, we run 1000 optimization steps of gradient ascent on a stabilized objective, using the same intrinsic score $s(x)$ as in our framework.

J.3.2. PROXY-GUIDED GENERATION: DIFFEXPLAINER

We implement the soft-prompt optimization variant of DiffExplainer (Pennisi et al., 2025). We use 5 soft tokens with a fixed prefix, optimizing the prompt

```
''photo of <SOFT PROMPT> <SOFT PROMPT> <SOFT PROMPT> <SOFT PROMPT> <SOFT PROMPT>''.
```
$$(214)$$

All remaining settings follow the original DiffExplainer protocol (Pennisi et al., 2025).

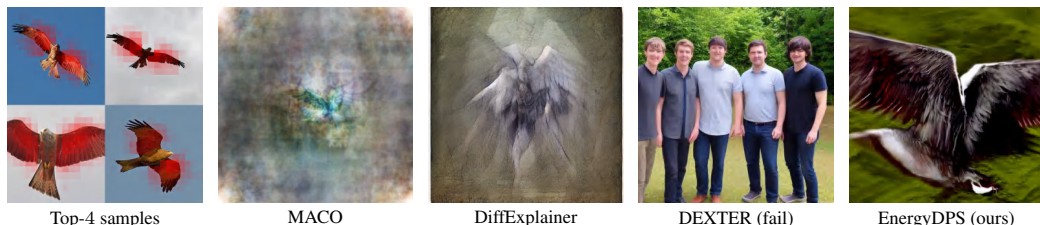

| Top-4 samples | MACO | DiffExplainer | DEXTER (fail) | EnergyDPS (ours) |

*Figure 5.* **DEXTER fails on SAE feature visualization.** Qualitative comparison for a DINOv3 SAE feature (index 9863). We report representative outputs from Top-$K$ dataset search, MACO, DiffExplainer, DEXTER (hard-prompt optimization), and EnergyDPS. While other methods can produce samples with high feature activation, DEXTER frequently collapses to low-activation outputs (final activation $s(x) = 1.4$ in our run), indicating a loss of faithfulness under feature-level objectives.

**Note on DEXTER (hard-prompt optimization).** DEXTER (Carnemolla et al., 2025) optimizes *discrete* (hard) prompts. In our feature-level setting, hard-prompt optimization was unstable and often failed to maintain the target SAE activation. In Fig. 5, for feature 9863, the best output we obtained reaches only $s(x) = 1.4$, where others reach $\sim 15$. This behavior is consistent with a *stronger information bottleneck* of hard prompts: restricting the search to a discrete prompt family can make it difficult to realize fine-grained, non-linguistic feature semantics, and the optimizer may converge to prompt-induced generations that are visually plausible but not faithful to the intrinsic feature objective.

### J.4. EnergyDPS

We follow the DPS procedure (Chung et al., 2023) without modification to the diffusion schedule and sampler details. The hyperparameters in Algorithm 1 (e.g., $N$, $\{\beta_i^{\mathrm{ddpm}}\}_{i=1}^N$, $\{\tilde{\sigma}_i\}_{i=1}^N$, and $\{\zeta_i\}_{i=1}^N$) follow the DPS. We use an ImageNet-256 diffusion model checkpoint and set the EnergyDPS guidance learning rate to $0.1$. The intrinsic guidance is applied through $s_k(\hat{x}_0)$ (with the mix reduction in Eq. (213)).

## K. Blinded Interpretability Study Settings by Different Raters

We conduct a blinded interpretability study to evaluate whether the generated visualization sets convey clear and consistent shared semantics. For each SAE feature and each compared method, we arrange four generated images into a single $2 \times 2$ grid. Raters are shown only the grid image, without access to the method name or feature index. Each rater is asked to identify the most salient concept shared by the four images and assign an interpretability score.

**Scoring rubric.** The interpretability score is defined on a 1–4 scale:

- 1: the shared concept cannot really be seen, or the images are too unclear or uninformative;

- 2: there is only a weak or vague hint of a shared concept;

- 3: the shared concept is reasonably visible and supported by useful visual evidence;

- 4: the shared concept is very clear and strongly supported by visually informative images.

A higher score indicates that the generated image set provides clearer and more visually informative evidence for a shared semantic concept.

*Table 6.* Detailed quantitative evaluation on DINOv3 by the blinded interpretability study (interpretability score ↑, weighted variance of description ↓).

| Methods | Interpretability score | | | | | | Weighted var. |
| | Human 1 | Human 2 | Human 3 | Human 4 | GPT | Gemini | |
| --- | --- | --- | --- | --- | --- | --- | --- |
| MACO | 2.77 | 1.97 | 1.02 | 2.31 | 2.09 | 2.57 | 0.2917 |
| DiffExplainer | 1.77 | 1.85 | 2.24 | 1.31 | 2.14 | 1.90 | 0.2643 |
| EnergyDPS | **2.90** | **2.31** | **3.57** | **2.73** | **2.90** | **3.09** | **0.2505** |

**Human raters.** Human raters follow the same scoring rubric. For each $2 \times 2$ grid, they are asked to provide: (i) a short free-form description of the shared semantic meaning, and (ii) an interpretability score according to the above rubric.

**AI raters.** For AI raters, we use the same visual input and scoring rubric. The system prompt is:

> You are a visual concept summarizer.
>
> The user will provide one image arranged as a 2x2 grid containing four sub-images. Identify the most salient shared concept across all provided images. Focus only on what is common to all images, and ignore image-specific details. Then rate how clearly this shared concept is supported by the image on a scale from 1 to 4, where 1 means the concept cannot really be seen or the images are too unclear or uninformative, 2 means there is only a weak or vague hint of a shared concept, 3 means the shared concept is reasonably visible and supported by useful visual evidence, and 4 means the shared concept is very clear and strongly supported by visually informative images. Return exactly two lines in the following format: concept: <short sentence under 20 words> score: <1-4 integer>. Do not provide any explanation or extra text.

The user prompt is:

> This is a single 2x2 grid image containing four related sub-images. Identify the shared concept they point to, and rate how clearly that concept is visible from 1 to 4. Return exactly in this format:
>
> concept: <short sentence>
>
> score: <1-4 integer>

We parse the returned concept description and score from the two-line output. The concept descriptions are further used to compute the weighted variance in CLIP text-embedding space, while the scores are used as the corresponding weights.

Detailed results for Table 4 are reported in Table 6.

# L. Additional Experiments for DINOv3

## L.1. Superposition in Neurons

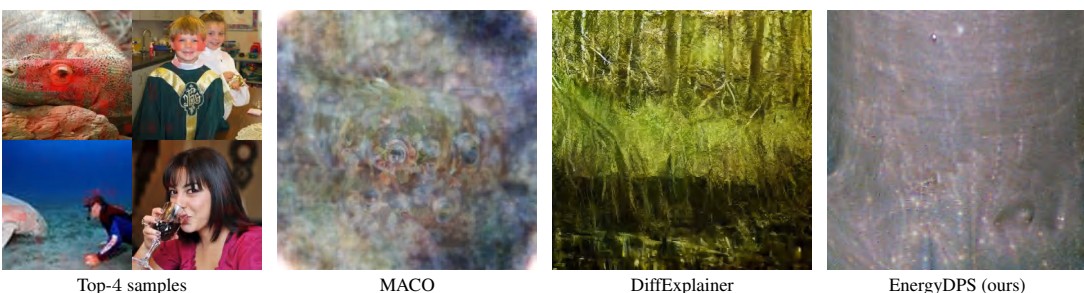

| Top-4 samples | MACO | DiffExplainer | EnergyDPS (ours) |

*Figure 6.* **Neuron-level interpretation is unstable.** Top-4 retrieval and three synthesis methods (MACO, DiffExplainer, EnergyDPS) for the same neuron yield heterogeneous or texture-like results, consistent with superposition and a lack of a single coherent semantic.

Fig. 6 visualizes a representative neuron using retrieval and three generation/optimization baselines. The Top-4 highest-activating dataset samples span heterogeneous semantics, and the synthesized images vary drastically across methods, frequently collapsing to texture-like patterns. This lack of a consistent, single-mode semantic explanation is *suggestive of neuron-level superposition*, where one unit mixes multiple underlying factors.

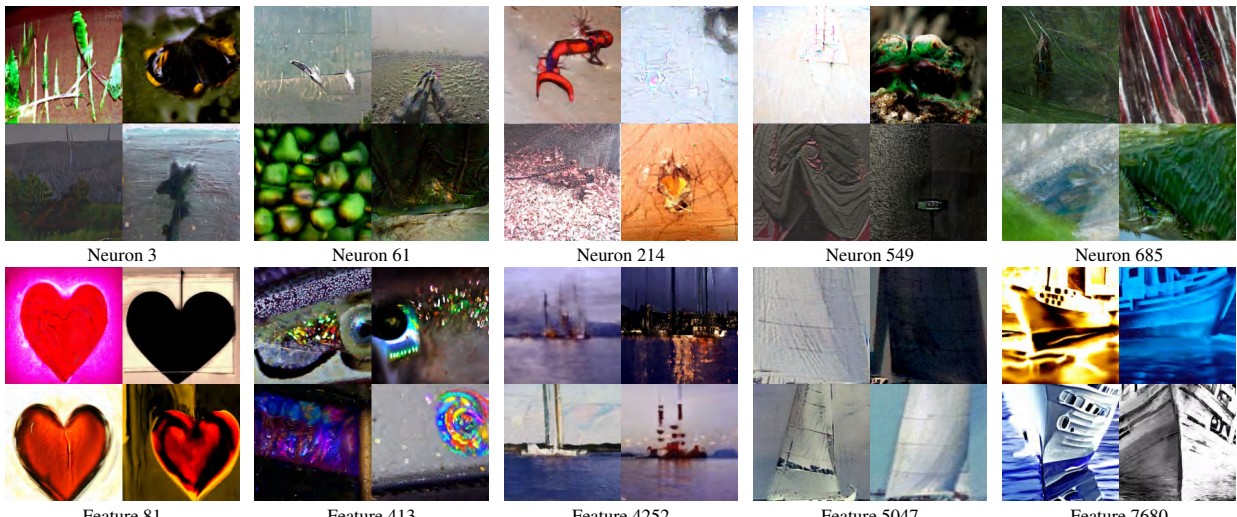

Neuron 3  Neuron 61  Neuron 214  Neuron 549  Neuron 685

Feature 81  Feature 413  Feature 4252  Feature 5047  Feature 7680

*Figure 7.* **Randomly selected neurons vs. SAE features under EnergyDPS.** Each unit is visualized by four images generated using the same EnergyDPS sampler. The top row shows randomly selected individual neurons, and the bottom row shows randomly selected SAE features.

*Table 7.* Pilot automated evaluation results (interpretability score ↑, success rate ↑).

| Methods | Mean interpretability score | Success rate (%) |
|---|---|---|
| Neurons | 2.24 | 28 |
| SAE Features | 2.90 | 74 |

As shown in Fig. 7, neuron-level visualizations (first row) often exhibit weaker cross-sample semantic consistency: the four generated images for the same neuron may emphasize different local textures, object fragments, or mixed visual cues. This suggests that individual neurons do not always provide a stable single-concept object for mechanistic interpretation, which is consistent with the superposition view that one neuron may entangle multiple latent factors. In contrast, SAE feature visualizations tend to show more coherent shared visual evidence across samples. This comparison supports our choice of sparse dictionary features, rather than individual neurons, as the primary object of study.

To further evaluate the superposition in neurons, we conducted a small pilot automated evaluation on generated visualization sets for 100 randomly sampled SAE features and 100 randomly sampled neurons. For each object (feature or neuron), GPT-5.4 was shown a set of four generated images via EnergyDPS and asked to assign an interpretability score on a 1–4 scale (details in Appendix K), where higher means that the shared concept is clearer and the images are more visually informative. We use 2.5 as a conservative threshold to calculate the success rate below. The results (shown in Table 7) indicate that SAE features are substantially more interpretable than neurons under this protocol.

Consequently, neuron activations can be an unreliable object of study for mechanistic interpretation, motivating the use of sparse dictionary features (e.g., SAE features) that better isolate individual factors.

### L.2. More Ablation Experiments

#### L.2.1. ABLATION EXPERIMENT ON THE SAE LAYER

For the main study, we selected stage-2 layer-20 as a representative intermediate layer in DINOv3 ResNeXt Large and used it as the primary testbed to study EnergyDPS. To examine robustness to layer choice, we additionally trained transcoders on other layers and evaluated the corresponding features. The additional results (shown in Table 8) show that EnergyDPS remains competitive on both a lower layer (e.g., layer 5) and a higher layer (e.g., layer 25), suggesting that the main conclusions are not tied to the specific choice of layer 20.

*Table 8.* Quantitative evaluation across different layers (QualiCLIP ↑, NRQM ↑).

| Layer | Methods | QualiCLIP | NRQM |
|-------|---------|-----------|------|
| | MACO | 0.4440 | 5.4322 |
| 5 | DiffExplainer | 0.3527 | 6.8261 |
| | EnergyDPS | **0.7187** | **8.0481** |
| | MACO | 0.4228 | 4.4869 |
| 20 | DiffExplainer | 0.3259 | 6.7460 |
| | EnergyDPS | **0.5791** | **6.7821** |
| | MACO | 0.4274 | 4.4635 |
| 25 | DiffExplainer | 0.3170 | 5.8579 |
| | EnergyDPS | **0.4283** | **7.8664** |

*Table 9.* Quantitative evaluation (QualiCLIP ↑, NRQM ↑) on different diffusion priors.

| Methods | QualiCLIP | NRQM |
|---------|-----------|------|
| MACO | 0.4228 | 4.4869 |
| DiffExplainer | 0.3259 | 6.7460 |
| EnergyDPS-FFHQ | 0.4790 | 6.7421 |
| EnergyDPS | 0.5791 | 6.7821 |

### L.2.2. ABLATION EXPERIMENT ON DIFFERENT DIFFUSION PRIORS

The diffusion prior does affect the interpretation results, and the generated explanations should be understood relative to the chosen prior. To examine this dependence, we additionally evaluated a DDPM pretrained on FFHQ (Kazemi & Sullivan, 2014) (denoted EnergyDPS-FFHQ), which is trained only on 70k face images, as an alternative prior for the experiment corresponding to Table 3 of the main paper. Compared with the ImageNet-pretrained prior, the explanation quality drops noticeably (shown in Table 9). This suggests that a narrower or more biased prior can indeed distort the posterior samples and degrade explanatory quality.

### L.2.3. ABLATION EXPERIMENT ON STRENGTH OF THE DIFFUSION PRIOR

*Table 10.* **Induced distribution distance under different guidance strengths.** For feature 81, we generate an unguided reference set of 2k prior samples, an independent unguided evaluation set, and 2k EnergyDPS-guided samples under different guidance strengths. We report mean activation and compute KID against the unguided reference set.

| Sample source | Mean activation | KID |
|---------------|-----------------|-----|
| Unguided set | 1.23e-03 | 0.32 |
| | 6.17 | 29.7 |
| EnergyDPS | 10.28 | 49.47 |
| | 16.17 | 64.66 |

To examine how guidance strength changes the induced distribution, we generate an unguided reference set of 2k prior samples, an independent unguided evaluation set, and 2k EnergyDPS-guided samples for feature 81. We report mean activation and compute KID against the unguided reference set; the unguided row therefore measures the finite-sample discrepancy between two independent prior draws. Table 10 shows that KID increases monotonically as the achieved mean activation increases. This indicates that stronger intrinsic guidance progressively moves the sampled distribution away from the unguided diffusion prior, which is consistent with the KLSC view that EnergyDPS samples from a feature-induced posterior rather than simply reproducing the prior.

We further examine how the generated samples change as the intrinsic guidance becomes stronger. Specifically, for feature 81, we start from a reference image and generate EnergyDPS samples under different guidance strengths, resulting in

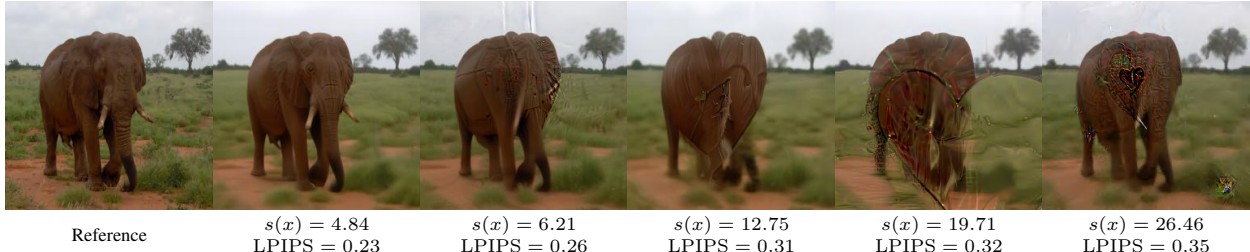

| | $s(x) = 4.84$ | $s(x) = 6.21$ | $s(x) = 12.75$ | $s(x) = 19.71$ | $s(x) = 26.46$ |
| Reference | LPIPS = 0.23 | LPIPS = 0.26 | LPIPS = 0.31 | LPIPS = 0.32 | LPIPS = 0.35 |

*Figure 8.* **Image-level deviation under increasing guidance.** For feature 81, we compare EnergyDPS samples generated with different activation levels against a reference image and report LPIPS values.

samples with progressively increasing activation scores $s(x)$. We then compute LPIPS between each generated sample and the reference image to quantify the perceptual image-level deviation. The results are shown in Fig. 8. As the achieved activation increases, LPIPS also increases, indicating that stronger EnergyDPS guidance induces larger perceptual deviations from the reference while moving the sample toward higher feature activation.

### L.3. An Illustrative Case for Semantic Shift across Activation Levels

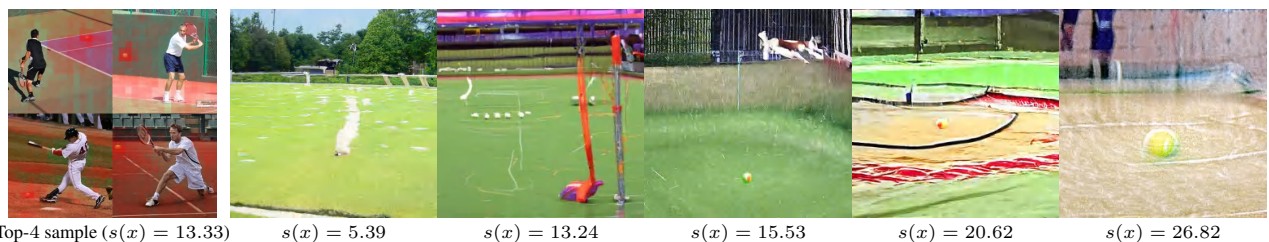

| Top-4 sample ($s(x) = 13.33$) | $s(x) = 5.39$ | $s(x) = 13.24$ | $s(x) = 15.53$ | $s(x) = 20.62$ | $s(x) = 26.82$ |

*Figure 9.* **Semantic shift across activation levels.** Left: the Top-4 highest-activating dataset sample for the same feature (a retrieval baseline). Right: samples generated by our sampler, ordered by increasing intrinsic score $s_k(x)$ (annotated below each image). As $s_k(x)$ increases, the dominant evidence shifts from coarse, context-like court/field cues to a localized object cue (the tennis ball), illustrating that extreme activations may emphasize only one semantic mode.

We observe that the *dominant visual evidence* associated with a fixed feature can vary across activation levels. Figure 9 shows samples (generated by EnergyDPS) ordered by increasing intrinsic score $s(x)$. At lower scores, the feature is primarily supported by *contextual cues*, such as court-like green surfaces and line-marking textures, while an explicit ball cue is often absent or weak. As $s(x)$ increases, the evidence progressively concentrates on a *localized object cue*: high-score samples consistently contain a salient tennis ball, even when the surrounding context becomes more variable. This case suggests that restricting interpretation to extreme activations can over-emphasize a single semantic mode (here, the ball) and under-represent other co-occurring modes (here, court-like context), motivating a distributional characterization of feature semantics rather than a single-threshold view.

### L.4. Efficiency Comparisons for Computation

Table 11 reports end-to-end runtime and peak GPU memory per sample on a single NVIDIA H100 (80GB) under each method's standard instantiation (all methods follow the same settings as in our main experiments). Dataset Search is prohibitively expensive (4252 s/sample) because it performs a full scan over ImageNet-1K to retrieve maximally activating examples (batch size 32), highlighting a scalability bottleneck that grows with the candidate pool size. DiffExplainer incurs substantially higher memory (26.5GB) and runtime (217 s/sample), consistent with prompt-space proxy optimization through a text-conditioned diffusion model.

EnergyDPS achieves a more favorable accuracy–efficiency trade-off in practice. Notably, EnergyDPS uses an unconditional DDPM prior, whereas DiffExplainer relies on a stronger text-conditioned backbone (Stable Diffusion v1.5); prior evidence suggests unconditional diffusion often fits the data worse than conditional generation, making this comparison conservative for EnergyDPS in terms of generation quality. Despite this disadvantage, EnergyDPS remains faster and more memory-efficient than DiffExplainer (96 vs. 217 s/sample; 14.6GB vs. 26.5GB) while producing substantially more interpretable and

faithful visualizations in Table 3 and Fig. 4. Compared to MACO, EnergyDPS introduces only a moderate overhead (96 vs. 48 s/sample) but yields a marked improvement in faithfulness and semantic coherence, indicating that the gains do not come at an excessive computational cost.

*Table 11.* **Efficiency comparisons.** Processing time and peak GPU memory per sample on a single NVIDIA H100 80GB under the same settings as our main experiments.

| Methods | Processing time (s/sample) | GPU memory cost (MiB) |
|---|---|---|
| Dataset Search | 4252 | 3865 |
| MACO (Fel et al., 2023a) | 48 | 7765 |
| DiffExplainer (Pennisi et al., 2025) | 217 | 26471 |
| EnergyDPS (ours) | 96 | 14567 |

### L.5. Object Drift under Task Injection

This section provides an empirical diagnostic of *object drift* under task injection, which is discussed in Sec. 2.2.3. We start from the intrinsic EnergyDPS target $q_\beta(x) \propto p_\theta(x) \exp(\beta s(x))$ and add an external guidance term $f_c$ with weight $\eta \geq 0$, yielding

$$q_{\beta,\eta,c}(x) \propto p_\theta(x) \exp\big(\beta s(x) + \eta f_c(x)\big). \tag{215}$$

Our goal is not to argue that task-guided explanations are invalid—they are appropriate when one aims to explain *task-conditioned* behavior (e.g., spurious features for a target label). Rather, we use these experiments to show that, under *intrinsic* MI, introducing $f_c$ can change what is being visualized: even when the intrinsic activation level is held fixed, increasing task alignment can qualitatively alter the semantics of samples.

**Protocol (matched intrinsic activation).** Across all settings, we generate samples using EnergyDPS with both intrinsic and external guidance. We then compare samples while keeping the intrinsic score approximately matched, $s(x) \approx \bar{s}$, but allowing the external alignment $f_c(x)$ to vary. If the visualization were purely driven by the intrinsic score, then samples with matched $s(x)$ should exhibit broadly consistent semantics; large semantic changes under matched $s(x)$ indicate object drift toward the injected task objective.

L.5.1. SEMANTIC TASK INJECTION VIA CLIP PROMPT ALIGNMENT

We instantiate $f_c$ using CLIP text–image alignment for a fixed prompt $c$. In this experiment, the prompt is: *"photorealistic underwater wildlife photo, great white shark swimming, side view slightly from below, close to camera, mouth slightly open, sharp teeth visible, clear deep blue ocean, sunlight rays and surface ripples above, small fish around, natural colors, high detail, sharp focus, 8k"*. We report the intrinsic score $s(x)$ (SAE feature activation) and the CLIP alignment score $f_c(x)$. Fig. 10 shows that samples can exhibit markedly different semantics as $f_c(x)$ increases, even when $s(x)$ remains nearly unchanged, indicating that the injected prompt objective can dominate the resulting visualization.

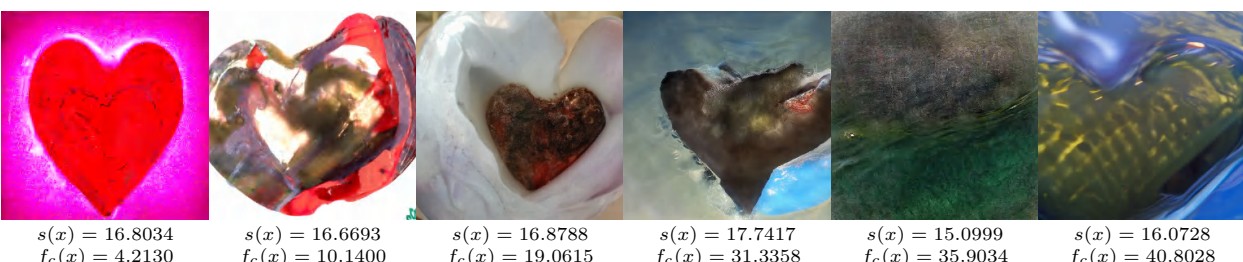

| $s(x) = 16.8034$ | $s(x) = 16.6693$ | $s(x) = 16.8788$ | $s(x) = 17.7417$ | $s(x) = 15.0999$ | $s(x) = 16.0728$ |
| $f_c(x) = 4.2130$ | $f_c(x) = 10.1400$ | $f_c(x) = 19.0615$ | $f_c(x) = 31.3358$ | $f_c(x) = 35.9034$ | $f_c(x) = 40.8028$ |

*Figure 10.* **Object drift under semantic task injection (CLIP prompt).** EnergyDPS is guided by an intrinsic SAE score $s(x)$ and an external CLIP alignment score $f_c(x)$ for a fixed prompt $c$. Although the intrinsic activations are nearly matched across samples, increasing $f_c(x)$ induces large semantic changes, indicating that prompt-level task injection can dominate the visualization and shift the explanation away from the intrinsic feature.

## L.5.2. REFERENCE INJECTION VIA PIXEL-LEVEL $\ell_2$ GUIDANCE

We further consider a stronger, non-linguistic task objective defined by a reference image $y$. To keep the "larger-is-better" convention consistent with Eq. (215), we define

$$f_y(x) := -\|x - y\|_2^2, \tag{216}$$

which is equivalent to minimizing the pixel MSE to $y$. Fig. 11 shows the same phenomenon: under approximately matched intrinsic activations $s(x)$, decreasing MSE (stronger reference alignment) drives samples toward reconstructing the reference, overriding feature-specific variation. This provides a complementary instance of object drift where the injected objective is purely pixel-level rather than language-mediated.

**Takeaway.** Across both semantic (CLIP) and pixel-level ($\ell_2$) objectives, we observe a consistent pattern: *matched intrinsic activation does not guarantee matched semantics once an external objective is introduced*. This supports the distributional view in Section 2.2.3: task injection changes the effective object of study and can therefore be misaligned with intrinsic feature interpretation.

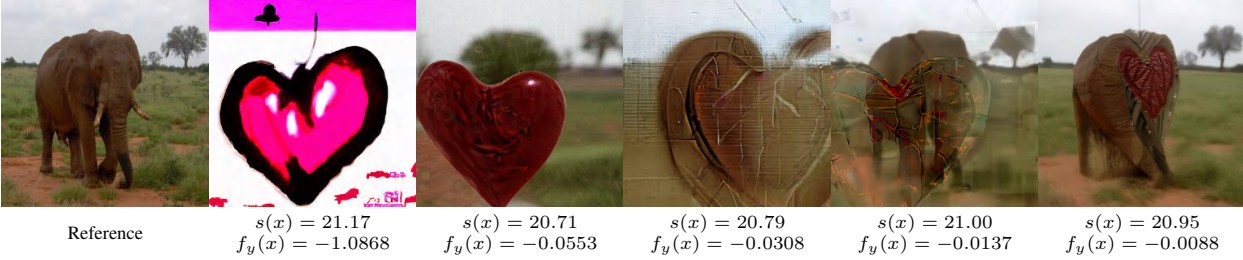

| Reference | $s(x) = 21.17$ $f_y(x) = -1.0868$ | $s(x) = 20.71$ $f_y(x) = -0.0553$ | $s(x) = 20.79$ $f_y(x) = -0.0308$ | $s(x) = 21.00$ $f_y(x) = -0.0137$ | $s(x) = 20.95$ $f_y(x) = -0.0088$ |

*Figure 11.* **Object drift under reference injection (pixel $\ell_2$ guidance).** EnergyDPS is guided by the intrinsic SAE score $s(x)$ and a reference objective $f_y(x) = -\|x - y\|_2^2$ for a fixed reference image $y$. Even when $s(x)$ is approximately matched, reducing MSE increasingly forces samples to resemble $y$, showing that pixel-level task injection can override intrinsic feature semantics.

## L.6. More Comparisons in Features

We provide additional qualitative comparisons for SAE features in DINOv3. Fig. 12 shows a $10 \times 10$ grid of visualizations for 100 randomly selected features using MACO, DiffExplainer, and EnergyDPS. Fig. 13 further reports several representative features with clearer semantics.

Qualitatively, DiffExplainer exhibits a clear semantic mismatch relative to the other references (Top-4 dataset samples, MACO, and EnergyDPS): across many features, its generations drift toward different high-level semantics rather than preserving feature-specific patterns suggested by the intrinsic baselines (Fig. 12–13). This observation is consistent with our distributional view that proxy/prompt-guided optimization restricts the search to a prompt-parameterized family, which can be poorly aligned with fine-grained, non-verbal, or polysemantic internal features.

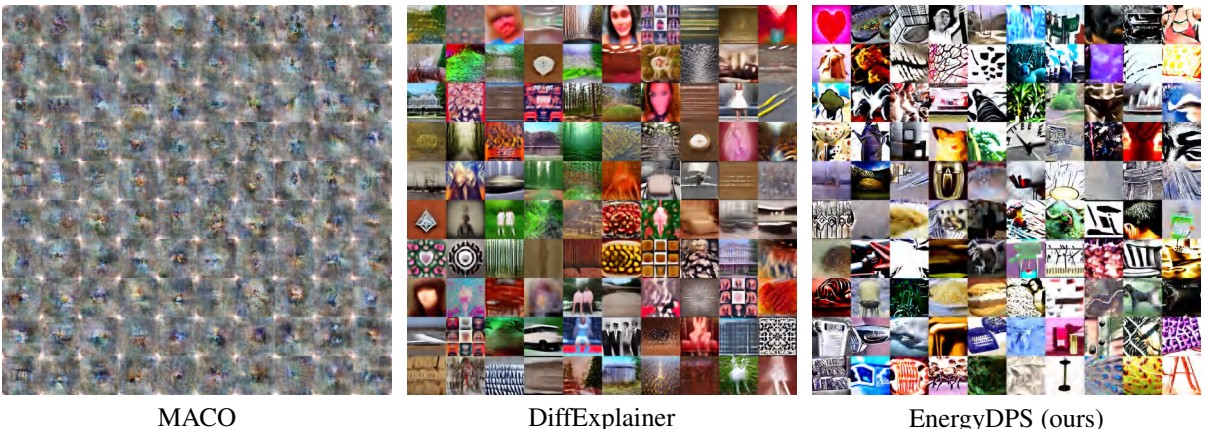

MACO                          DiffExplainer                     EnergyDPS (ours)

*Figure 12.* **Additional feature visualizations on 100 random SAE features.** Each panel shows a $10 \times 10$ grid of feature visualizations produced by the corresponding method under the same evaluation protocol as in the main experiment.

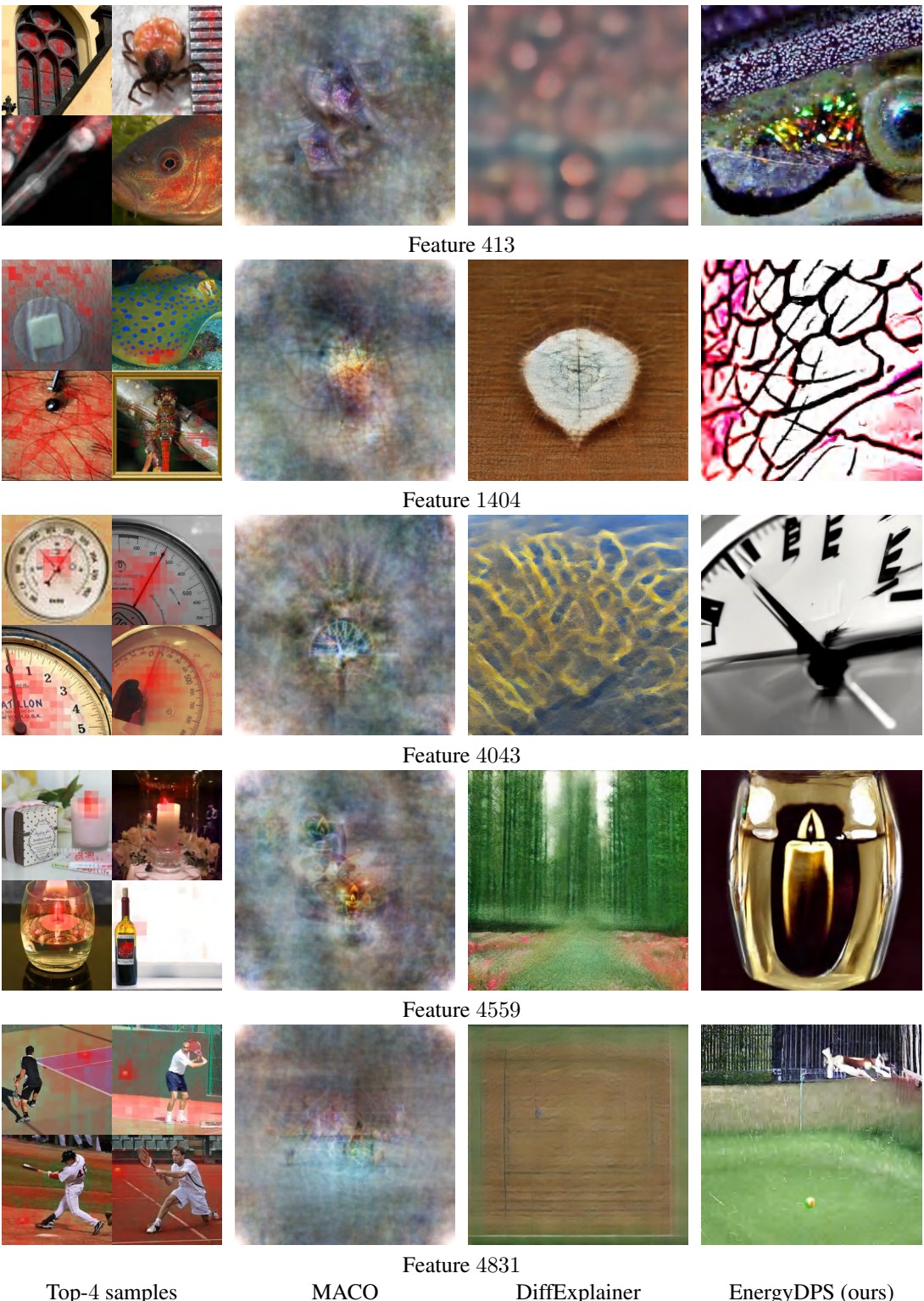

Feature 413

Feature 1404

Feature 4043

Feature 4559

Feature 4831

| Top-4 samples | MACO | DiffExplainer | EnergyDPS (ours) |

*Figure 13.* **Representative feature examples.** For each feature, we show Top-4 dataset samples (retrieval baseline), MACO, DiffExplainer, and EnergyDPS. DiffExplainer often produces qualitatively different semantics from the other references, while EnergyDPS remains closer to the dataset exemplars and intrinsic-guided optimization under the same protocol. These features visually appears to capture eye-shaped, mesh texture, panel-shaped, candle, tennis structures, respectively. These textual descriptions are only post-hoc visual summaries for reader guidance; we do not assume or enforce any one-to-one alignment between word labels and SAE features, nor treat them as ground-truth labels.

