# OpenReview forum: "A Distributional View for Visual Mechanistic Interpretability: KL-Minimal Soft-Constraint Principle"
_ICML.cc/2026/Conference — ICML 2026 spotlight_

### Official Review · Reviewer_uVx3 · 2026-03-08

**Soundness:** 3
**Presentation:** 3
**Significance:** 3
**Originality:** 3
**Overall Recommendation:** 5
**Confidence:** 2

**Summary:**

This paper identifies the limitations of previous mechanistic interpretability (MI) and proposes a solution to address them. It outlines the problems in MI by comparing the differences between language data and image data. While image data has continuous characteristics, existing hard-constraints create a boundary bias. To mitigate this, the authors define a hazard rate and propose a soft-constraint based on this definition. Finally, the paper defines the bounds of distortion that can occur due to task-injection during the MI process. To alleviate this, it proposes energy-guided DPS, which samples from diffusion models.

**Compliance With Llm Reviewing Policy:**

Affirmed.

**Final Justification:**

The authors' response has addressed my concerns.

**Key Questions For Authors:**

It is difficult to assume that all diffusion models perfectly capture the true distribution of image data. However, since the continuous nature of image data is the main focus, it seems that even early diffusion models that roughly capture the image distribution could generate results, albeit at a lower quality. Have you conducted any experiments regarding this?

**Limitations:**

The proposed explanation method is dependent on the diffusion model.

**Strengths And Weaknesses:**

The proposed approach is theoretically grounded for each method and is experimentally validated through a toy model. The mathematical concepts are well-organized for utilizing the diffusion model. However, it is somewhat disappointing that definitions I personally consider important are relegated to the appendix. Nevertheless, this is understandable given the strict page limits of conference papers, which make it difficult to present both mathematical rigor and experimental results simultaneously.

---

> ### Author Rebuttal · Authors · 2026-03-29
>
> We sincerely thank your positive comments and constructive suggestions. We reply to the comments below.
>
> ***
>
> **Organization and presentation of paper.**
>
> We appreciate the comment regarding the placement of important definitions and intuitions. Due to the page limit, the current manuscript is necessarily dense, and several technical details were pushed to the appendix. In the camera-ready version, we will use the additional page to improve the exposition by moving several central intuitions and definitions into the main text.
>
> ***
>
> **The dependence on the diffusion prior.**
>
> Our method uses the distribution learned by the diffusion model as an approximation to the natural image prior, so the quality of this learned prior does affect the quality of the final explanations. Importantly, however, our method does not require the diffusion prior to perfectly match the true natural image distribution. Rather, the role of the diffusion model is to provide a continuous and sampleable approximation to the image manifold. If this approximation becomes weaker, narrower, or more biased, the generated explanations are still possible, but their quality is expected to degrade.
>
> To verify this point, we additionally evaluated a DDPM pretrained on FFHQ [1] (denoted EnergyDPS-FFHQ), which is trained only on 70k face images, as an alternative prior for the experiment corresponding to Table 3 of the main paper. Although this is not an “early” diffusion model, it is trained on a much narrower domain and therefore does not adequately capture the broader natural image distribution. In our qualitative inspection, EnergyDPS-FFHQ can still generate recognizable results, but the samples exhibit more unnatural textures and a noticeably stylized appearance. We will include these qualitative examples in the camera-ready version. The quantitative results are shown below.
>
> |Methods|QualiCLIP|NRQM|
> |-|-|-|
> |MACO|0.4228|4.4869|
> |DiffExplainer|0.3259|6.7460|
> |EnergyDPS-FFHQ|0.4790|6.7421|
> |EnergyDPS|0.5791|6.7821|
>
> Compared with the ImageNet-pretrained prior, the explanation quality drops noticeably under the FFHQ prior. This suggests that even when the diffusion model only imperfectly captures the target image distribution, EnergyDPS can still produce meaningful samples, but the results become less natural and therefore less interpretable to humans. We will clarify this limitation more explicitly in the revised version.
>
> [1]Kazemi, V., & Sullivan, J. (2014). One millisecond face alignment with an ensemble of regression trees. In Proceedings of the IEEE conference on computer vision and pattern recognition (pp. 1867-1874).
>
> ***
>
> Thanks again for your positive comments and constructive suggestions. Should you require any further information, please do not hesitate to let us know. We would greatly appreciate your feedback.

---

> > ### Author Rebuttal · Reviewer_uVx3 · 2026-04-04
> >
> > The authors' sincere response has addressed my concerns. I have raised my score accordingly.

---

> > > ### Author Response · Authors · 2026-04-04
> > >
> > > We thank the reviewer again for the constructive comments and suggestions!

---

### Official Review · Reviewer_4w74 · 2026-03-08

**Soundness:** 3
**Presentation:** 3
**Significance:** 3
**Originality:** 3
**Overall Recommendation:** 5
**Confidence:** 4

**Summary:**

The submission studies visual mechanistic interpretability (MI). That is, the task of identifying which "concepts" (i.e., human-interpretable features) are encoded by particular neurons inside large, black-box predictors. In the broader machine learning literature, this is also known as "feature visualization", a task that goes back to the introduction of the first convolutional networks (e.g., the Inception network).

The submission sets forth guiding principles to generate example images that are representative of a sparse autoencoders (SAEs) features. The proposed method, EnergyDPS, boils down to conditional guidance of a pretrained diffusion model with the gradient of the SAE feature score.

Theoretical analyses and synthetic experiments present the properties of the proposed method, and experiments on DINOv3 compare with existing methods.

**Compliance With Llm Reviewing Policy:**

Affirmed.

**Final Justification:**

I have expanded on the justification of my final score in the rebuttal acknowledgement.

**Key Questions For Authors:**

**Interpretability vs Realism**

The notion of "interpretability" presented in the paper is that of realism of the samples. Could the authors expand on how the KLSC principle may be extended to cases where the distribution $p$ is not inherently interpretable by humans? For example, there exist image domains such as medical imaging, astronomical data, or earth sciences, where the input space is not inherently interpretable. There, producing realistic images does not provide interpretability gains.

**Strength of the Diffusion Prior**

To the point of realism, could the authors expand on how practitioners can distinguish between real signal in an SAE feature or the prior overpowering the results? For example, it would be interesting to show what samples look like for highly-activated SAE features vs lowly-activated ones. If the samples generated by EnergyDPS for those images also look realistic, that might be evidence that the method really is picking up on the prior rather than the SAE feature? Or if an SAE feature is weak, than the posterior distribution should match the prior $p$? In that sense, it might be better to report the distance of $p$ from the posterior $q$, with for example, LPIPS or FID.

Could the authors expand on how the learned image prior might bias the sample images generated by EnergyDPS? For example, how should one choose which diffusion model to use? The experiments use ImageNet, which contains it own biases.

These are important points, as visual interpretability methods are prone to over-interpretation by projecting our own biases and preconceptions onto the generated images.

**Questions DINOv3 Experiments**

Could the authors expand on how they assigned words like "heart" and "wing" to the SAE features? Are baseline methods dependent on this choice of word?

Could the authors expand on how practitioners should assign these words in practice? For example, looking at Fig. 4, I am not convinced that feature 9863 represents "wing" or "wings"? What are the images shown in Figure 4? Posterior means or individual samples out of EnergyDPS? What does the posterior distribution look like?

---

**Minor Comments**

- Typo on line 186 (left column)? Should be capital $X$ randomly sampled from P?
- The claim that "hard-constraint methods tend to pick just-barely activate examples" is misleading: the result shows that hard thresholding is more biased towards the boundary compared to KLSC, but the method itself precisely selects all images with score greater than $m$. The statement does not provide an absolute measure of bias, it could still be the case that the mass around the boundary is small, and it will be even smaller for KLSC.
- The fact that KL divergence can be infinity when the $p$, $q$ do not have the same support is well-known. A statement in terms of TV distance with be more interesting.
- Typo on line 335 (left column): a proxy optimization.
- Lines 372 - 376 (left column): this sentence contradicts the motivation that SAE features are monosemantic. I am not sure the argument about "nonverbal" features is convincing within the scope of the submission since the experiments still try and assign words to features.

**Limitations:**

yes

**Strengths And Weaknesses:**

**Strengths**
1. The paper is well-written
2. Theoretical analysis of the proposed method is rigorous and interesting
3. Results provide evidence in support of the proposed method

**Weaknesses**
1. The notion of interpretability mentioned in the paper is closer to realism
2. The proposed method falls within the broader category of guided diffusion, which limits novelty of the algorithm itself
3. The qualitative comparison of samples for individual features seems ad-hoc

I will expand on these points below and I am looking forward to discussing with the authors!

---

> ### Author Rebuttal · Authors · 2026-03-29
>
> We sincerely thank you for the positive comments and constructive suggestions. We reply to them below.
> ***
> **Interpretability vs. realism**
>
> Thank you for raising this distinction. In our paper, interpretability is not equivalent to realism; it asks whether model behavior is expressed in a form humans can understand. We use the natural image distribution as the reference space because natural images are relatively human-interpretable, and because the DINOv3 itself is an image representation model, so aligning back to the natural image distribution incurs the least information loss. By contrast, projecting features into a prompt space may discard intrinsic visual information, consistent with the information bottleneck discussed in Sec. 2.2.4.
>
> For domains where the raw data are not directly human-interpretable, the KLSC can be generalized by choosing a human-interpretable associated modality and defining the prior there. For example, in the fields of geoscience, geophysicists typically interpret reconstructed subsurface parameters (like velocity models) rather than the uninterpretable raw observed seismic data, so the prior could be defined over velocity models instead.
> ***
> **Strength of the diffusion prior**
>
> We agree that this question concerns whether the explanation is genuinely driven by the model behavior rather than dominated by the learned image prior. We also agree with the reviewer’s intuition that weak guidance should keep the posterior close to the prior, while stronger guidance should move it farther away. To examine this, we conducted three additional analyses.
>
> 1) Prior-posterior distance. We generated 2k unguided and guided samples (feature 81) and computed KID (a robust metric similar to FID) against the unguided prior. With very weak guidance (Act.), KID remains close to the prior; as guidance increases, KID rises monotonically, showing that the posterior moves progressively farther away.
>
> |Sets|Act.|KID|
> |-|-|-|
> |Unguided set|1.23e-3|0.32|
> |EnergyDPS|6.17|29.70|
> ||10.28|49.47|
> ||16.17|64.66|
>
> 2) Instance-level deviation. Following Appd. K.4.2, we fixed the reference guidance and varied the guidance strength (Act.) for feature 81. LPIPS increases with guidance, indicating larger image-level deviations.
>
> |Act.|LPIPS|
> |-|-|
> |4.84|0.23|
> |6.21|0.26|
> |12.75|0.31|
> |19.71|0.32|
> |26.46|0.35|
>
> 3) Influence of different diffusion priors. We additionally tested a DDPM pretrained on FFHQ (EnergyDPS-FFHQ), which is much narrower than ImageNet. The quality drops noticeably, confirming that narrower or more biased priors can degrade explanations.
>
> |Methods|QualiCLIP|NRQM|
> |-|-|-|
> |MACO|0.42|4.48|
> |DiffExplainer|0.32|6.74|
> |EnergyDPS|0.57|6.78|
> |EnergyDPS-FFHQ|0.47|6.74|
>
> Overall, these results support two points: (i) very weak guidance keeps the posterior close to the prior, while stronger guidance moves it farther away; and (ii) explanation quality depends on the quality and domain coverage of the diffusion prior. We will include these results in the revision.
> ***
> **Questions on DINOv3 experiments**
>
> We thank the reviewer for pointing this out. Labels such as “heart” or “wing” are post-hoc informal descriptions added only to convey the rough semantic tendency of a feature. They are not used by our method or by any baseline during generation, so the comparisons do not depend on this wording.
>
> We also agree that such labels should not be interpreted as the precise semantics of an SAE feature. In our view, the object of interpretation is the posterior distribution induced by the feature, rather than a single word. Accordingly, the images shown in the manuscript are individual posterior samples from EnergyDPS, not posterior means.
>
> To assess whether these sampled distributions admit stable human interpretation, we conducted a small study on 100 random features (details in response to Reviewer 1KPt). Six raters were asked to provide short descriptions, and EnergyDPS achieved the lowest weighted variance of CLIP embeddings, suggesting more consistent interpretations.
>
> |Methods|Weighted Variance|
> |-|-|
> |MACO|0.2917|
> |DiffExplainer|0.2643|
> |EnergyDPS|0.2505|
>
> ***
> **Additional discussion about hard threshold**
>
> We agree that the phrase “hard-constraint methods tend to pick just-barely activate examples” is too strong. The intended claim is comparative rather than absolute: compared with KLSC, hard-threshold methods are more biased toward near-threshold examples, and this relative boundary bias is what leads to the KL-based instability in Appd. D.3. We will revise it accordingly.
>
> We also appreciate the suggestion to consider TV. A TV-based characterization is possible and complementary: the local TV sensitivity of hard-threshold conditioning is governed by the hazard rate. Moreover, in canonical settings such as Gaussian, we find that at matched faithfulness levels, the local TV of KLSC is more stable. We will discuss it in the revision.
> ***
> Thanks again for your comments. Also, we will revise the typos.

---

> > ### Author Rebuttal · Reviewer_4w74 · 2026-04-02
> >
> > I sincerely thank the authors for addressing my questions.
> >
> > I raised my score accordingly. I ask that, in the revised version of the paper, the authors make sure to include:
> > - Discussion of "interpretability vs lying on the natural image manifold", and how this might need to be addressed in other domains, which is nontrivial.
> > - Discussion of importance of guidance strength.
> > - Discussion of importance of prior strength.
> > - Include more examples of the samples generated by EnergyDPS for the particular neurons.
> > - Potentially remove the associated word labels, which might cause confusion rather than strenghten presentation.
> > - Include the reader study mentioned in response to Reviewer 1Kpt

---

> > > ### Author Response · Authors · 2026-04-03
> > >
> > > We sincerely appreciate your constructive comments and helpful suggestions. We will make sure to incorporate all of the suggested revisions in the camera-ready version. Thank you again for your support!

---

### Official Review · Reviewer_K4yc · 2026-03-09

**Soundness:** 3
**Presentation:** 4
**Significance:** 3
**Originality:** 3
**Overall Recommendation:** 5
**Confidence:** 3

**Summary:**

This paper provides a distributional view for visual mechanistic interpretability analysis, summarizing current visual mechanistic interpretability methods from three perspectives: Object of study, Constraint formulation, and Sampling methodology. Based on this framework, it analyzes the biases of existing methods. Therefore, this paper proposes a brand new paradigm called SAE energy-guided DPS, which resolves the biases present in current works.

**Compliance With Llm Reviewing Policy:**

Affirmed.

**Final Justification:**

I would remain my positive assessment.

**Key Questions For Authors:**

1. Will the proposed interpretability method fail when applied to SAE feature dimensions that lack clear semantics? If so, what proportion of the total SAE dimensions actually correspond to clear semantic concepts (e.g., Feature 81, which clearly models the "heart" concept)?

2. The authors train a transcoder on stage-2, layer-20 of DINOv3 ResNeXt Large. How was this specific intermediate layer selected? Furthermore, how significantly would the interpretation results be affected if a different intermediate layer were chosen?

3. To what extent does the prior distribution $p_\theta(\cdot)$ modeled by the Diffusion Model influence the final interpretation results of the proposed method?

**Limitations:**

yes

**Strengths And Weaknesses:**

Strengths:
1. Presentation. This paper is well-written. The narrative flow is highly coherent: it first clearly identifies the potential flaws of existing methods before proposing the new approach based on these findings.

2. Rigorous Argumentation. The paper provides detailed mathematical proofs for its key theorems. Furthermore, the experimental section offers clear settings, making the paper easy to follow.

3. Novel Theoretical Perspective. This paper examines existing visual mechanistic interpretability methods, offering a theoretical perspective that has not been clearly studied previously. In the field of mechanistic interpretability, it is often difficult to conduct direct comparisons across different interpretability methods, and the distributional view proposed here provides a unified theoretical framework to help address this issue.

Weakness:
1. The newly proposed method, SAE energy-guided DPS, requires performing interpretation on features decoded by a SAE. Consequently, the interpretability performance of this new method heavily relies on the quality of the chosen SAE.

2. The proposed method relies on the prior distribution $p_\theta(\cdot)$ modeled by the chosen diffusion ,odel, as stated in Section 2.2.4 (Line 331). This dependency would causes the interpretation results to be influenced by the diffusion model's own biases.

---

> ### Author Rebuttal · Authors · 2026-03-29
>
> We sincerely thank your positive comments and constructive suggestions. We reply to the comments below.
>
> ***
>
> **Dependence on SAE quality.**
>
> Our method does depend on the quality of the underlying SAE features. In practice, we do observe a nontrivial number of features whose semantics remain weak or difficult to interpret. This is precisely why we focus on SAE features rather than individual neurons: although imperfect, SAE features are substantially more interpretable on average. To support this point, we conducted a small pilot automated evaluation on generated visualization sets for 100 randomly sampled SAE features and 100 randomly sampled neurons. For each object (feature or neuron), GPT-5.4 was shown a set of four generated images via EnergyDPS and asked to assign an interpretability score on a 1–4 scale  (details in response to Reviewer 1KPt), where higher means that the shared concept is clearer and the images are more visually informative. We use 2.5 as a conservative threshold to calculate the success rate below. The results (shown below) indicate that SAE features are substantially more interpretable than neurons under this protocol. We view this as preliminary evidence to validate the interpretability of SAE features, and we will include this study and clarify its details in the camera-ready version. The resulting success rate also provides a preliminary estimate of the proportion of SAE dimensions that correspond to relatively clear semantic concepts.
>
> |Object|Mean score|Success rate|
> |-|-|-|
> |Neurons|2.24|28%|
> |SAE feature|2.90|74%|
>
>
>
> ***
>
> **Choice of the transcoder layer.**
>
> For the main study, we selected stage-2 layer-20 as a representative intermediate layer in DINOv3 ResNeXt Large and used it as the primary testbed to study EnergyDPS. To examine robustness to layer choice, we additionally trained transcoders on other layers and evaluated the corresponding features. The additional results (shown below, following Table 3 of the main text) show that EnergyDPS remains competitive on both a lower layer (e.g., layer 5) and a higher layer (e.g., layer 25), suggesting that the main conclusions are not tied to the specific choice of layer 20.
>
> |Layer|Methods|QualiCLIP|NRQM|
> |-|-|-|-|
> |5|MACO|0.4440|5.4322|
> |5|DiffExplainer|0.3527|68261|
> |5|EnergyDPS|0.7187|8.0481|
> |20|MACO|0.4228|4.4869|
> |20|DiffExplainer|0.3259|6.7460|
> |20|EnergyDPS|0.5791|6.7821|
> |25|MACO|0.4274|4.4635|
> |25|DiffExplainer|0.3170|5.8579|
> |25|EnergyDPS|0.4283|7.8664|
>
> More broadly, differences among SAE features across layers are interesting research directions. In preliminary analysis, following prior work [1], we examined feature biases such as texture/shape/color preference and observed clear layer-dependent trends in DINOv3. We view this as an important future direction and will report a more systematic study in our future work.
>
> [1] Burgert, T., Stoll, O., Rota, P., & Demir, B. ImageNet-trained CNNs are not biased towards texture: Revisiting feature reliance through controlled suppression. In The Thirty-ninth Annual Conference on Neural Information Processing Systems.
>
> ***
>
> **Dependence on the diffusion prior.**
>
> The diffusion prior does affect the interpretation results, and the generated explanations should be understood relative to the chosen prior. To examine this dependence, we additionally evaluated a DDPM pretrained on FFHQ [2] (denoted EnergyDPS-FFHQ), which is trained only on 70k face images, as an alternative prior for the experiment corresponding to Table 3 of the main paper. Compared with the ImageNet-pretrained prior, the explanation quality drops noticeably (shown below). This suggests that a narrower or more biased prior can indeed distort the posterior samples and degrade explanatory quality.
>
> |Methods|QualiCLIP|NRQM|
> |-|-|-|
> |MACO|0.4228|4.4869|
> |DiffExplainer|0.3259|6.7460|
> |EnergyDPS-FFHQ|0.4790|6.7421|
> |EnergyDPS|0.5791|6.7821|
>
> In our experiments, we use an ImageNet-pretrained DDPM because it provides a broader and better-matched prior than FFHQ for interpreting general natural-image features. Compared with the narrow-domain FFHQ prior, ImageNet covers a much wider range of object categories and visual patterns, making it a more appropriate reference prior for the DINOv3 setting studied here. It also provides strong generation quality in practice and enables interpretable visualizations for a large fraction of features (over 70% in our preliminary study; see the success-rate table above). For these reasons, we believe this is a reasonable and representative setup here. Future work will explore stronger diffusion priors that may capture natural-image statistics even better.
>
> [2] Kazemi, V., & Sullivan, J. (2014). One millisecond face alignment with an ensemble of regression trees. In Proceedings of the IEEE conference on computer vision and pattern recognition (pp. 1867-1874).
>
> ***
>
> Thanks again for your positive comments and constructive suggestions.

---

> > ### Author Rebuttal · Reviewer_K4yc · 2026-04-03
> >
> > The authors have provided a clear rebuttal and fully resolve my initial concerns. Therefore I would remain my positive assessment.

---

> > > ### Author Response · Authors · 2026-04-03
> > >
> > > We thank the reviewer again for the constructive comments and suggestions!

---

### Official Review · Reviewer_1KPt · 2026-03-12

**Soundness:** 3
**Presentation:** 2
**Significance:** 3
**Originality:** 4
**Overall Recommendation:** 5
**Confidence:** 2

**Summary:**

This paper aims to models the influence of feature activation on natural image distribution. The focus is on whether concepts (discovered in mech-interp settings) are uninterpretable to humans (i.e., far from natural image distribution) and how to make them more aligned with human understanding. This is realized via diffusion posterior sampling, they propose EnergyDPS, which aims to solve the problems of boundary bias, threshold instability, and interpretability gap for visual interpretations (e.g., maximally activating examples). Experiments on Dinov3 demonstrate qualitative and quantitative improvements over recent baselines

**Compliance With Llm Reviewing Policy:**

Affirmed.

**Final Justification:**

My initial review stands and the rebuttal resolved my main question about the relationship between the data manifold and interpretability to humans

**Key Questions For Authors:**

- Is staying on the image manifold really the same as human interpretability? For example, models can use concepts from the dataset that humans cannot understand. To truly determine if this is true, a human study would need to be done that confirms that more closely matching the natural image distribution increases human interpretability. I wonder if you could define a diffusion prior based on interpretable features (concepts) rather than just images.

**Limitations:**

No, there is no discussion of limitations of the work. There should be a section (at a minimum in the appendix) on realistic limitations of the method.

**Strengths And Weaknesses:**

Overall, I think this is a super interesting paper with strong motivation and nice results (especially Figures 3 and 4). The related work and experiments are generally good and I think the paper should be accepted. The main weakness being that it is extremely dense and most of the crucial details are left for the appendix, with a whopping 148(!) equations in total.

Strengths
- Turning the qualitative interpretation into a quantifiable math problem is an awesome idea and a great thing to strive for
- I like the taxonomy presented in Table 1, across objects, constraints, and samplings, is very interesting, intuitive, and useful
- The idea of turn a hard constraint into a soft one (Defn. 2.2) is an nice idea and well executed (diffusion models are great at this!)

Weaknesses
- Overall, while the grammar and prose were clean, the paper is extremely dense with the majority of details in the appendix. One way to improve this would be to add one sentence describing the experimental setting to the beginning of each figure, which are currently pretty dense and difficult to parse without looking at the appendix.

small stuff
- zfaithfulness in line 412

---

> ### Author Rebuttal · Authors · 2026-03-29
>
> We sincerely thank your positive comments and constructive suggestions. We reply to the comments below.
>
> ***
>
> **Presentation, organization, and limitations**
>
> We sincerely appreciate the reviewer’s comments on the presentation of the paper. Limited by the page limit of the conference paper, some details are pushed to the appendix. In the camera-ready version, we will make full use of the additional page to improve the exposition by moving several central intuitions into the main text, adding one-sentence summaries of the experimental setting at the start of figure captions, and correcting the typos in the current version.
>
> We also appreciate the suggestion to discuss the limitations more explicitly. In the revised version, we will include a dedicated discussion of the limitations of our current approach. In particular, there still remains a gap between the natural image distribution and truly human-understandable concepts. How to further reduce this gap is an important open question. We believe that incorporating human-preference alignment, potentially through RLHF-style strategies, may be a promising direction for future work.
>
> ***
>
> **Natural image manifold vs. human interpretability**
>
> We agree that staying on the natural image manifold isn't equivalent to human interpretability. A concept-level prior would indeed be appealing, but if it is instantiated through texts, labels, or prompts, it reintroduces the prompt bottleneck discussed in our paper (Section 2.2.4 Methodology III). If it is viewed as a more latent underlying concept space, then images and texts are at best different projections of it rather than direct access to it [1], so such a prior is difficult to observe and optimize directly. Under this perspective, our claim is operational rather than absolute: aligning with the natural image prior brings explanations back into a human-perceptible space and therefore tends to improve interpretability on average, even though it is not a complete definition of human semantic understanding.
>
> As suggested, to better support this point, we additionally conducted a small blinded pilot human study on our generated visualization image sets for the 100 random features demonstrated in Figure 10 of the manuscript. Independent human and AI (GPT-5.4, Gemini-3-flash-lite) raters were shown, for each feature and each method, a set of four generated images and asked to provide (i) a short description of the shared semantic meaning, and (ii) an interpretability score on a 1–4 scale, where higher means that the shared concept is clearer and the images are more visually informative. The standard of interpretability score is: 1: cannot really tell the concept, or the images are too unclear/uninformative; 2: only a weak or vague shared concept, or the visual evidence is poor; 3: the shared concept is reasonably visible and the images provide usable evidence; 4: the shared concept is very clear and the images are visually clear and informative. In this pilot study (shown below), EnergyDPS obtained the highest average interpretability score among the compared methods. We view this as preliminary evidence that better alignment with the natural image prior tends to improve human interpretability.
>
> As an auxiliary consistency analysis, we also measured the weighted variance (weighted by the interpretability score) of textual descriptions in CLIP embedding space. Lower variance suggests that annotators converged to more similar semantic descriptions. From the below table, EnergyDPS also showed the lowest variance under this metric, which is consistent with the human ratings. We will include this pilot study and clarify its details in the camera-ready version.
>
> |Methods|Human 1|Human 2|Human 3|Human 4|GPT-5.4|Gemini-3-flash-lite|Weighted variance of description $\downarrow$|
> |-|-|-|-|-|-|-|-|
> |MACO|2.77|1.97|1.02|2.31|2.09|2.57|0.2917|
> |DiffExplainer|1.77|1.85|2.24|1.31|2.14|1.90|0.2643|
> |EnergyDPS|2.90|2.31|3.57|2.73|2.90|3.09|0.2505|
>
> [1] Minyoung Huh, Brian Cheung, Tongzhou Wang, Phillip Isola. Position: The Platonic Representation Hypothesis. Proceedings of the 41st International Conference on Machine Learning, PMLR 235:20617-20642, 2024.
>
>
> ***
>
> Thanks again for your positive comments and constructive suggestions. Should you require any further information, please do not hesitate to let us know. We would greatly appreciate your feedback.

---

> > ### Author Rebuttal · Reviewer_1KPt · 2026-04-03
> >
> > All my concerns are resolved and I think the deeper exploration into the 'manifold vs interpretability' experiment (something the other reviewer brought up too) makes the papers motivation much stronger! I have raised my score accordingly.

---

> > > ### Author Response · Authors · 2026-04-04
> > >
> > > We thank the reviewer again for the constructive comments and suggestions!

---

### Decision · Program_Chairs · 2026-04-30

**Decision:**

Accept (spotlight)

**Comment:**

The paper introduces a principled distributional view for visual mechanistic interpretability, combining strong theoretical grounding with solid empirical validation, and reviewers consistently appreciated its novelty, clarity of motivation, and unifying perspective over prior methods. Minor concerns regarding presentation and dependence on the diffusion prior were effectively addressed in the rebuttal with additional analyses and planned clarifications, leading all reviewers to maintain or raise their scores to strong accept. Overall, this is a clear, well-supported contribution and merits acceptance.